# TOWARDS FEDERATED LEARNING ON TIME-EVOLVING HETEROGENEOUS DATA

## ABSTRACT

Federated Learning (FL) is an emerging learning paradigm that preserves privacy by ensuring client data locality on edge devices. The optimization of FL is challenging in practice due to the diversity and heterogeneity of the learning system. Despite recent research efforts on improving the optimization of heterogeneous data, the impact of time-varying heterogeneous data in real-world scenarios, such as changing client data or intermittent clients joining or leaving during training, has not been well studied. In this work, we propose Continual Federated Learning (CFL), a flexible framework, to capture the time-varying heterogeneity of FL. CFL covers complex and realistic scenarios—which are challenging to evaluate in previous FL formulations—by extracting the information of past local datasets and approximating the local objective functions. Theoretically, we demonstrate that CFL methods achieve a faster convergence rate than FedAvg in time-varying scenarios, with the benefit being dependent on approximation quality. In a series of experiments, we show that the numerical findings match the convergence analysis, and CFL methods significantly outperform the other SOTA FL baselines.

## 1 INTRODUCTION

Federated Learning (FL) has recently emerged as a critical distributed machine learning paradigm to preserve user/client privacy. Clients engaged in the training process of FL only communicate their local model parameters, rather than their private local data, with the central server.

As the workhorse algorithm in FL, FedAvg (McMahan et al., 2017) performs multiple local stochastic gradient descent (SGD) updates on the available clients before communicating with the server. Despite its success, FedAvg suffers from the large heterogeneity (non-iid-ness) in the data presented on the different clients, causing drift in each client's updates and resulting in slow and unstable convergence (Karimireddy et al., 2020b). To address this issue, a new line of study has been suggested lately that either simulates the distribution of the whole dataset using preassigned weights of clients (Wang et al., 2020; Reisizadeh et al., 2020; Mohri et al., 2019; Li et al., 2020a) or adopts variance reduction methods (Karimireddy et al., 2020b;a; Das et al., 2020; Haddadpour et al., 2021). However, the FL formulation in these approaches always assumes a fixed data distribution among clients throughout all training rounds, while in practice this assumption does not always hold because of the complex, uncontrolled, and unpredictable client behaviors. Local datasets, for example, often vary over time, and intermittent clients can join or depart during training without prior notice.

The difficulty in addressing the time-varying heterogeneity of FL lies in the stateless nature of the local datasets, which includes the unpredictable future datasets and the impossibility of retaining all prior local datasets. To this end, we first provide a novel Continual Federated Learning (CFL) formulation and then propose a unified CFL framework as our solution. This framework encompasses a variety of design choices for approximating the local objective functions in the previous training rounds, with the difference between actual and estimated local object functions described as *information loss* for further analysis.

To capture the time-varying data heterogeneity in the CFL framework, we expand the theoretical assumption on the client drift—which has been extensively utilized recently in Karimireddy et al. (2020b); Khaled et al. (2020); Li et al. (2020b)—to include both client drift and *time drift*. This allows us to quantify the difference between local client objective function and global objective function for the considered time-varying heterogeneous clients.

We provide convergence rates for our unified CFL framework in conjunction with information loss and new models of both client and time drifts. Our analysis reveals a faster convergence of CFL framework than FedAvg, with the benefit dependent on approximation quality. The rate of FedAvg obtained from

our framework is consistent with previous work, while a similarly simplified rate on Continual Learning (CL) (French, 1999; Kirkpatrick et al., 2017) for the convex and strongly convex cases is novel. Finally, in extensive empirical results, we demonstrate that CFL methods—stemmed from the CFL framework with different approximation techniques—significantly outperform the SOTA FL competitors on various realistic datasets. These numerical observations corroborate our theoretical findings.

**Contribution.** We summarize our key contributions below.
- We present a unified framework, termed Continual Federated Learning (CFL), together with a novel client and time drift modeling approach, to capture complex FL scenarios involving time-varying heterogeneous data. This is the first theoretical study, to our knowledge, that describes the time-varying nature of FL.
- We provide rigorous convergence analysis for the CFL methods. Our theoretical analysis explains the faster and more stabilized optimization of the CFL methods over that of FedAvg, and conjecture their variance reduction effect. In addition, we provide tight convergence rates for standalone CL methods on convex and strongly-convex problems: to the best of our knowledge, we are the first to provide such guarantees for CL on SGD.
- We examine several approximation techniques for CFL framework: the insights therein offer a valuable practical guideline. We demonstrate the efficacy and necessity of CFL methods over the SOTA FL baselines across a range of time-varying heterogeneous data scenarios and datasets.

## 2 RELATED WORK

**Federated Learning.** FedAvg (McMahan et al., 2017; Lin et al., 2020b) is the de facto standard FL algorithm, in which multiple local SGD steps are executed on the available clients to alleviate the communication bottleneck. While communication efficient, heterogeneity, such as system heterogeneity (Li et al., 2018; Wang et al., 2020; Mitra et al., 2021; Diao et al., 2021) and statistical/objective heterogeneity (Li et al., 2018; Wang et al., 2020; Mitra et al., 2021; Lin et al., 2020a; Karimireddy et al., 2020b;a), results in inconsistent optimization objectives and drifted clients models, impeding federated optimization considerably.

A line of work has been proposed to address the heterogeneity in FL. FedProx (Li et al., 2018) adds the regularization term on the distance of local and global models when performing local training—similar formulations can be found in other recent FL works (Hanzely & Richtárik, 2020; Dinh et al., 2020; Li et al., 2021) for various purposes. To address the issue of objective heterogeneity e.g. caused by heterogeneous data, works like SCAFFOLD (Karimireddy et al., 2020b; Mitra et al., 2021) introduce the idea of variance reduction on the client local update steps. FedNova (Wang et al., 2020) further proposes a general framework for unifying FedAvg and FedProx, and argues that while averaging, local updates should be normalized to minimize heterogeneity induced by different number of local update steps. However, most of these prior works focus on the fixed heterogeneity across clients and throughout the entire optimization procedure; we instead consider the novel scenario with time-varying data heterogeneity.

The theoretical study on the convergence of FedAvg can date back to the parallel SGD analysis on the identical functions Zinkevich et al. (2010) and recently is improved by Stich (2019); Stich & Karimireddy (2020); Patel & Dieuleveut (2019); Khaled et al. (2020); Woodworth et al. (2020b). For the analysis of heterogeneous data, Li et al. (2020b) first give the convergence rate of FedAvg on non-iid datasets with random selection, assuming that the client optimum are $\epsilon$-close. Woodworth et al. (2020a); Khaled et al. (2020) give tighter convergence rates under the assumption of bounded gradient drift. All above works give a $\mathcal{O}\left(1/T\right)$ convergence rate for convex local objective functions. More recently, Karimireddy et al. (2020b); Koloskova et al. (2020) give the convergence analysis of local SGD for non-convex objective functions under bounded gradient noise assumptions, and obtain a $\mathcal{O}(1/\sqrt{T})$ convergence rate. Our theoretical analysis framework covers more challenging time-varying data heterogeneity in FL, which has not been considered in the community yet—our rate can be simplified to the standard FL scenario, matching the tight analysis in Karimireddy et al. (2020b).

**Continual Learning.** Continual learning, also known as incremental learning or lifelong learning, aims to learn from (time-varying) sequential data while avoiding the problem of *catastrophic forgetting* (French, 1999; Kirkpatrick et al., 2017). There exists a large amount of works, from the perspectives of regularization (Kirkpatrick et al., 2017; Li & Hoiem, 2017; Zenke et al., 2017), experience replay (Castro et al., 2018; Rebuffi et al., 2017), and dynamic architectures (Maltoni & Lomonaco, 2019; Rusu et al., 2016). In this paper, we examine both regularization and experience replay based methods, and compare their empirical performance in depth.

Despite the empirical success, the theoretical analysis of CL is limited: only a very recent preprint (Yin et al., 2020b) provides a viewpoint of regularization-based continual learning, and only for the single worker scenario. In this work, we provide tight convergence analysis for both CL and CFL on SGD.

**Continual Federated Learning.**    To our knowledge, the scenario of CFL was originally described in Bui et al. (2018) in order to federated train Bayesian Neural Network and continually learn for Gaussian Process models—it is orthogonal to our optimization aspect in this paper.

FedCurv (Shoham et al., 2019) blends EWC (Kirkpatrick et al., 2017) regularization with FedAvg, and empirically shows a faster convergence. FedWeIT (Yoon et al., 2021) extends the regularization idea, with a focus on selective inter-client knowledge transfer for task-adaptive parameters. In a very recent parallel (and empirical) work, CDA-FedAvg (Casado et al., 2021) uses a locally maintained long-term data samples, with asynchronous communication; however, this design choice lacks sufficient numerical results to justify the performance gain (even over FedAvg) and it cannot be applied to more general scenarios. Our theoretically sound CFL framework covers the regularization component of FedCurv and the core set part of CDA-FedAvg, and is orthogonal to the neural architecture manipulation idea in FedWeIT.

## 3 CONTINUAL FEDERATED LEARNING FRAMEWORK

### 3.1 FORMULATION

**Conventional FL Formulation.**    The standard FL typically considers a sum-structured distributed optimization problem as below:

$$f^\star = \min_{\boldsymbol{\omega} \in \mathbb{R}^d} \left[ f(\boldsymbol{\omega}) := \sum_{i=1}^N p_i f_i(\boldsymbol{\omega}) \right] , \tag{1}$$

where the objective function $f(\boldsymbol{\omega}) : \mathbb{R}^d \to \mathbb{R}$ is the weighted sum of the local objective functions $f_i(\boldsymbol{\omega}) := \mathbb{E}_{\mathcal{D}_i} [F_i(\boldsymbol{\omega})]$ of $N$ nodes/clients, and $p_i$ is the weight of client $i$.

In practice, it may be infeasible to select all clients each round, especially for cross-device setup (McMahan et al., 2017; Kairouz et al., 2019). The standard FedAvg then randomly selects $S$ clients to receive the model parameters from the server ($S \le N$) in each communication round and performs $K$ local SGD update steps in the form of $\boldsymbol{\omega}_{t,i,k} = \boldsymbol{\omega}_{t,i,k-1} - \eta_l(\nabla f_i(\boldsymbol{\omega}_{t,i,k-1}) + \boldsymbol{\nu}_{t,i,k-1})$, with local step-size $\eta_l$ and gradient noise $\boldsymbol{\nu}_{t,i,k-1}$. The selected clients then communicate the updates $\Delta\boldsymbol{\omega}_{t,i} = \boldsymbol{\omega}_{t,i,K} - \boldsymbol{\omega}_t$ with the server for the model aggregation: $\boldsymbol{\omega}_{t+1} = \boldsymbol{\omega}_t - \frac{\eta_g}{S} \sum_{i=1}^S \Delta\boldsymbol{\omega}_{t,i}$.

**Continual FL Formulation.**    Despite its wide usage, the standard FL formulation given in Equation (1) cannot properly reflect actual time-varying scenarios, such as local datasets changing over time or intermittent clients joining or leaving during training. To address this problem, we propose the Continual Federated Learning (CFL) formulation:

$$f^\star = \min_{\boldsymbol{\omega} \in \mathbb{R}^d} \left[ f(\boldsymbol{\omega}) := \sum_{t=1}^T \sum_{i \in \mathcal{S}_t} p_{t,i} f_{t,i}(\boldsymbol{\omega}) \right] , \tag{CFL}$$

where $f_{t,i}(\boldsymbol{\omega})$ represents the local objective function of client $i$ at time $t$. $\mathcal{S}_t$ is a subset of clients sampled from all clients set $\Omega$, where $|\mathcal{S}_t| = S$. For the time-varying scenarios, client $i$ could have different local objective functions $f_{t,i}(\boldsymbol{\omega})$ due to the changing local datasets on different $t$.

### 3.2 APPROXIMATION OF CFL

The challenge of addressing (CFL) formulation stems from the stateless nature of the local datasets, which include unpredictable future datasets, as well as the difficulty of keeping all the previous local datasets. The original definition of (CFL) is theoretically and empirically infeasible. To handle the second case (while ignoring the intractable former), a straightforward approach is to approximate the prior local objective functions (Kirkpatrick et al., 2017; Zenke et al., 2017; Li & Hoiem, 2017), which may be accomplished by retraining information from earlier rounds in accordance with privacy protection standards. Thus, we can express the approximated CFL formulation as:

$$\tilde{f}_t^\star = \min_{\boldsymbol{\omega} \in \mathbb{R}^d} \left[ \tilde{f}_t(\boldsymbol{\omega}) := \sum_{i \in \mathcal{S}_t} p_{t,i} f_{t,i}(\boldsymbol{\omega}) + \sum_{\tau=1}^{t-1} \sum_{i \in \mathcal{S}_t} p_{\tau,i} \tilde{f}_{\tau,i}(\boldsymbol{\omega}) \right] , \tag{2}$$

where $\tilde{f}_{t,i}(\boldsymbol{\omega})$ denotes the approximated local objective function of client $i$ at time $t$. In practice, different approximation methods can be used to calculate $\tilde{f}_{\tau,i}(\boldsymbol{\omega})$, and we refer the detailed illustration and discussions of these approximation algorithms in Section 5.1, Section 5.2, and Appendix C.2.2

We give the formal definition of CFL framework in Algorithm 1. CFL methods are a collection of methods that make use of different approximation techniques and are based on Algorithm 1. We retrieve the objective function of conventional Continual Learning (CL), by setting $S = 1$ in (2).

Take note that in most cases, the previous local object functions cannot be properly approximated. Due to the fact that such approximation impairs optimization, we define the information loss below.

---

**Algorithm 1** Approximated CFL Framework

---

**Require:** initial weights $\boldsymbol{\omega}_0$, global learning rate $\eta_g$, local learning rate $\eta_l$, number of training rounds $T$
**Ensure:** trained weights $\boldsymbol{\omega}_T$
1: **for** round $t = 1, \ldots, T$ **do**
2:      **communicate** $\boldsymbol{\omega}_t$ to the chosen clients.
3:      **for** client $i \in \mathcal{S}_t$ **in parallel do**
4:          initialize local model $\boldsymbol{\omega}_{t,i,0} = \boldsymbol{\omega}_t$.
5:          **for** $k = 1, \ldots, K$ **do**
6:              $\tilde{g}_{t,i,k}(\boldsymbol{\omega}_{t,i,k-1}) = \left( p_{t,i} \nabla f_{t,i}(\boldsymbol{\omega}_{t,i,k-1}) + \sum_{\tau=1}^{t-1} p_{\tau,i} \nabla \tilde{f}_{\tau,i}(\boldsymbol{\omega}_{t,i,k-1}) \right) + \boldsymbol{\nu}_{t,i,k}.$
7:              $\boldsymbol{\omega}_{t,i,k} \leftarrow \boldsymbol{\omega}_{t,i,k-1} - \eta_l \tilde{g}_{t,i,k}(\boldsymbol{\omega}_{t,i,k-1}).$
8:          **communicate** $\Delta\boldsymbol{\omega}_{t,i} \leftarrow \boldsymbol{\omega}_{t,i,K} - \boldsymbol{\omega}_t.$
9:      $\Delta\boldsymbol{\omega}_t \leftarrow \frac{\eta_g}{S} \sum_{i \in \mathcal{S}_t} \Delta\boldsymbol{\omega}_{t,i}.$
10:      $\boldsymbol{\omega}_{t+1} \leftarrow \boldsymbol{\omega}_t + \Delta\boldsymbol{\omega}_t.$

---

**Definition 3.1** (Information Loss). *We define the information loss as the difference between the approximated local objective function $\tilde{f}_{t,i}(\boldsymbol{\omega})$ and the real local objective function $f_{t,i}(\boldsymbol{\omega})$, i.e.*

$$\Delta_{t,i}(\boldsymbol{\omega}) = \nabla f_{t,i}(\boldsymbol{\omega}) - \nabla \tilde{f}_{t,i}(\boldsymbol{\omega}). \tag{3}$$

In practice, we use $\|\Delta_{t,i}(\boldsymbol{\omega})\|_2$ to measure the information loss of different approximation methods. We show large information loss can impede the convergence theoretically (c.f. Theorem 4.1) and empirically (e.g. Figure 1 in Section 5.2).

### 3.3 Gradient Noise Model

To analyze Equation (2) and Algorithm 1 in-depth, we propose the Gradient Noise Model (Equation (5)) to capture the dynamics of performing SGD with data heterogeneity. We first recap the standard definition of gradient noise in Sammut & Webb (2010); Gower et al. (2019).

**Gradient noise in SGD.** For objective function $f(\boldsymbol{\omega})$, the gradient with stochastic noise can be defined as $\nabla f(\boldsymbol{\omega}) = g(\boldsymbol{\omega}) + \boldsymbol{\nu}$, where $g(\boldsymbol{\omega})$ is the stochastic gradient, and $\boldsymbol{\nu}$ is a zero-mean noise.

**Client drift in FL.** To analyze the impact of data heterogeneity, recent works (Karimireddy et al., 2020b; Khaled et al., 2020; Li et al., 2020b) similarly use the gradient noise to capture the distribution drift between local objective function and global objective function:

$$\nabla f(\boldsymbol{\omega}) = \nabla f_i(\boldsymbol{\omega}) + \boldsymbol{\delta}_i, \tag{4}$$

where $\boldsymbol{\delta}_i$ is a zero-mean random variable which measures the gradient noise of client $i$.

**Client drift and time drift in CFL framework.** In (CFL), we use $\{t, i\}$ pair instead of $i$ to represent client $i$ at time $t$. Considering the distribution drift in the dimension of client and time, we further modify the gradient noise model in FL as,

$$\nabla f(\boldsymbol{\omega}) = \nabla f_{t,i}(\boldsymbol{\omega}) + \boldsymbol{\delta}_{t,i} + \boldsymbol{\xi}_{t,i}. \tag{5}$$

Note that we extend the gradient noise to two terms: $\boldsymbol{\delta}_{t,i}$ and $\boldsymbol{\xi}_{t,i}$. $\boldsymbol{\delta}_{t,i}$ is a time independent (of $t$), zero-mean random variable that measures the drift of client $i$. The zero-mean $\boldsymbol{\xi}_{t,i}$ measures the drift of client $i$ at time $t$: this value of client $i$ may change over time $t$, due to the time-varying local datasets.

**Remark 3.2.** *We assume different $\boldsymbol{\xi}_{t,i}$ are independent to simplify the analysis. We also empirically examine the performance of CFL methods under overlapped time-varying local data in Section 5.2.*

### 3.4 Assumptions

To ease the theoretical analysis of (CFL) framework, we use the following widely used assumptions.

**Assumption 1** (Smoothness and convexity). *Assume local objective functions $f_{t,i}(\boldsymbol{\omega})$ are $L$-smooth and $\mu$-convex, i.e. $\frac{\mu}{2} \|\boldsymbol{\omega}_1 - \boldsymbol{\omega}_2\|^2 \leq f_{t,i}(\boldsymbol{\omega}_1) - f_{t,i}(\boldsymbol{\omega}_2) - \langle \nabla f_{t,i}(\boldsymbol{\omega}_2), \boldsymbol{\omega}_1 - \boldsymbol{\omega}_2 \rangle \leq \frac{L}{2} \|\boldsymbol{\omega}_1 - \boldsymbol{\omega}_2\|^2$.* A widely used corollary is that if a function $f_{t,i}$ is both $L$-smooth and $\mu$-convex, then it satisfies $\frac{1}{2L} \|\nabla f_{t,i}(\mathbf{x}) - \nabla f_{t,i}(\mathbf{y})\|^2 \leq f_{t,i}(\mathbf{x}) - f_{t,i}(\mathbf{y}) - \nabla f_{t,i}(\mathbf{x})^T (\mathbf{x} - \mathbf{y})$, and $L \geq \mu$.

**Assumption 2** (Bounded noise in stochastic gradient). *Let $g_{t,i,k}(\boldsymbol{\omega}) = \nabla f_{t,i,k}(\boldsymbol{\omega}) + \boldsymbol{\nu}_{t,i,k}$, where $\boldsymbol{\nu}_{t,i,k}$ is the stochastic noise of client $i$ on round $t$ at $k$-th local update step. We assume that $\mathbb{E}[\boldsymbol{\nu}|\boldsymbol{\omega}] = 0$, and $\mathbb{E}\left[\|\boldsymbol{\nu}\|^2 |\boldsymbol{\omega}\right] \leq \sigma^2$.*

Table 1: **Number of communication rounds required to reach $\varepsilon + \varphi$ accuracy** for $\mu$ strongly convex and general convex functions under time-varying FL scenarios (Assumption 1–4). We can recover the rates in conventional FL setups by setting $D = 0$ and $A = 0$. Note that $\varphi = 0$ in FedAvg. Our convergence rate of FedAvg matches the results in Karimireddy et al. (2020b). Our SGD rate of CL on strongly-convex case is novel.

| Algorithm | Strongly Convex | General Convex |
|---|---|---|
| **SGD** | | |
| | $\frac{\sigma^2}{\mu NK\varepsilon} + \frac{1}{\mu}$ | $\frac{\sigma^2}{NK\varepsilon^2} + \frac{1}{\varepsilon}$ |
| **FedAvg** | | |
| Li et al. (2020b) | $\frac{\sigma^2}{\mu^2 NK\varepsilon} + \frac{(G^2+D^2)K}{\mu^2\varepsilon}$ | - |
| Khaled et al. (2020) | $\frac{\sigma^2+G^2+D^2}{\mu NK\varepsilon} + \frac{\sigma+G+D}{\mu\sqrt{\varepsilon}} + \frac{N(A^2+B^2)}{\mu}$ | $\frac{\sigma^2+G^2+D^2}{NK\varepsilon^2} + \frac{\sigma+G+D}{\mu\varepsilon^{\frac{3}{2}}} + \frac{N(A^2+B^2)}{\varepsilon}$ |
| Karimireddy et al. (2020b) | $\frac{\sigma^2}{\mu NK\varepsilon} + \frac{G+D}{\mu\sqrt{\varepsilon}} + \frac{c_{p_B}}{\mu}$ | $\frac{\sigma^2}{KN\varepsilon^2} + \frac{G+D}{\varepsilon^{\frac{3}{2}}} + \frac{c_{p_B}(D^2+G^2)}{\varepsilon}$ |
| Woodworth et al. (2020a) | $\frac{\sqrt{c_{p_B}}}{K\sqrt{\mu\varepsilon}} + \frac{\sigma^2 c_{p_B}}{NK\varepsilon^2} + \frac{G+D}{\mu\sqrt{\varepsilon}} + \frac{\sigma}{\mu\sqrt{K\varepsilon}}$ | $\frac{c_{p_B}}{K\varepsilon} + \frac{\sigma^2 c_{p_B}}{NK\varepsilon^2} + \frac{(G+D)c_{p_B}}{\varepsilon^{\frac{3}{2}}} + \frac{\sigma c_{p_B}}{\sqrt{K}\varepsilon^{\frac{3}{2}}}$ |
| Ours | $\frac{\sigma^2}{\mu NK\varepsilon} + \frac{G^2+D^2}{\mu\varepsilon} + \frac{\frac{\sigma}{\sqrt{K}}+G+D}{\mu\sqrt{\varepsilon}} + \frac{c_{p_B}}{\mu}$ | $\frac{c_{p_B}}{\varepsilon} + \frac{\sigma^2}{NK\varepsilon^2} + \frac{G^2+D^2}{\varepsilon^2} + \frac{\frac{\sigma}{\sqrt{K}}+G+D}{\varepsilon^{\frac{3}{2}}}$ |
| **CL** | | |
| Yin et al. (2020b) | $\frac{\mu}{\varepsilon}$ (GD) | - |
| Ours | $\frac{1+A^2}{\mu} + \frac{\sigma^2}{\mu K\epsilon} + \frac{c_A}{\mu\epsilon} + \sqrt{\frac{c_B}{\mu\epsilon}} + \sqrt{\frac{\frac{\sigma^2}{K}+c_A}{\mu^2\epsilon}} + \sqrt[3]{\frac{c_B}{\mu^2\epsilon}}$ | $\frac{c_{p_B}+\sqrt{c_B}+\sqrt[3]{c_B}}{\varepsilon} + \frac{\sigma^2}{K\varepsilon^2} + \frac{c_A}{\varepsilon^2} + \sqrt{\frac{\frac{\sigma^2}{K}+c_A}{\varepsilon^3}}$ |
| **CFL** | | |
| Ours | $\frac{c_{p_B}}{\mu} + \frac{\sigma^2}{\mu NK\epsilon} + \frac{c_A+G^2}{\mu\epsilon} + \sqrt{\frac{c_B}{\mu\epsilon}} + \sqrt{\frac{\frac{\sigma^2}{K}+c_A+G^2}{\mu^2\epsilon}} + \sqrt[3]{\frac{c_B}{\mu^2\epsilon}}$ | $\frac{c_{p_B}+\sqrt{c_B}+\sqrt[3]{c_B}}{\varepsilon} + \frac{\sigma^2}{NK\varepsilon^2} + \frac{c_A+G^2}{\varepsilon^2} + \sqrt{\frac{\frac{\sigma^2}{K}+c_A+G^2}{\varepsilon^3}}$ |

$c_{p_B} = 1 + B^2 + A^2, c_A = \frac{D^2 R^2}{R^2+D^2}, c_B = \frac{D^6}{(R^2+D^2)^2}$. $N$ is the number of chosen clients in each round, and $K$ is the number of local iterations.

Assumption 1 and 2 are common assumptions in FL (Karimireddy et al., 2020b; Li et al., 2020b). Considering a relaxed assumption like the bounded noise at optimum in Khaled et al. (2020) will be an interesting future research direction. Besides, we introduce the following assumptions to better characterize the client and time drifts in CFL framework (i.e. (5)).

**Assumption 3** (Bounded gradient drift of CFL framework). *Let $f(\boldsymbol{\omega}) = \nabla f_{t,i}(\boldsymbol{\omega}) + \boldsymbol{\delta}_{t,i} + \boldsymbol{\xi}_{t,i}$, where $\boldsymbol{\delta}_{t,i}$ measures client drift and $\boldsymbol{\xi}_{t,i}$ indicates the time drift of $\{t,i\}$. We assume $\mathbb{E}[\boldsymbol{\delta}|\boldsymbol{\omega}] = 0$, $\mathbb{E}[\boldsymbol{\xi}|\boldsymbol{\omega}] = 0$, and thus $\mathbb{E}\left[\|\boldsymbol{\delta}\|^2 |\boldsymbol{\omega}\right] \leq G^2 + B^2\mathbb{E}\|\nabla f(\boldsymbol{\omega})\|^2$ and $\mathbb{E}\left[\|\boldsymbol{\xi}\|^2 |\boldsymbol{\omega}\right] \leq \hat{D}^2 + \hat{A}^2\mathbb{E}\|\nabla f_i(\boldsymbol{\omega})\|^2$.*

We can derive the corollary from Assumption 3 that $\mathbb{E}\left[\|\boldsymbol{\xi}\|^2 |\boldsymbol{\omega}\right] \leq \hat{D}^2 + \hat{A}^2 G^2 + \hat{A}^2 B^2\mathbb{E}\|\nabla f(\boldsymbol{\omega})\|^2$, which can simplified to $\mathbb{E}\left[\|\boldsymbol{\xi}\|^2 |\boldsymbol{\omega}\right] \leq D^2 + A^2\mathbb{E}\|\nabla f(\boldsymbol{\omega})\|^2$, by assuming $A^2 = \hat{A}^2 B^2$, $D^2 = \hat{D}^2 + \hat{A}^2 G^2$.

Assumption 3 assumes the bounded $\mathbb{E}\left[\|\boldsymbol{\delta}\|^2 |\boldsymbol{\omega}\right]$ and $\mathbb{E}\left[\|\boldsymbol{\xi}\|^2 |\boldsymbol{\omega}\right]$ in CFL framework, which is equivalent to the widely used $(G, B)$ gradient drift assumption[1] (Gower et al., 2019; Karimireddy et al., 2020b) on the bounded $\mathbb{E}\left[\|\nabla f_i(\boldsymbol{\omega})\|^2\right]$. We prove this claim in Appendix B.1.

**Assumption 4** (Bounded information loss). *We assume the information loss $\Delta_{t,i}(\boldsymbol{\omega})$ can be bounded by an arbitrary non-negative value R, i.e. $\|\Delta_{t,i}(\boldsymbol{\omega})\| \leq R$.*

**Remark 3.3.** *The exact information loss in Assumption 4 may be intractable for some approximation methods. In Lemma B.3 of Appendix B.3, we showcase a tight analysis of information loss for Taylor extension based regularization methods.*

# 4 THEORETICAL RESULTS

In this section, we analyze the theoretical performance of Algorithm 1 under Assumption 1–4. We prove that CFL methods converges faster than FedAvg, despite that the term (refer to $\varphi_t$ in Theorem 4.1) induced by information loss concurrently trades off the optimization. Additionally, our results include the FedAvg rate (and match the prior work) and a novel convergence rate for SGD on CL. In Table 1, we summarize the convergence rate of different algorithms[2]. The proof details refer to Appendix B.4 and Appendix B.5.

## 4.1 CONVERGENCE RATE OF CFL METHODS

We start our analysis of Algorithm 1 from the one round progress of CFL methods in the Theorem stated below.

---

[1]The $(G, B)$ gradient drift: $\mathbb{E}\left[\|\nabla f_i(\boldsymbol{\omega})\|^2\right] \leq G^2\mathbb{E}\left[\|\nabla f(\boldsymbol{\omega})\|^2\right] + B^2$, where $G^2 \geq 1$ and $B^2 \geq 0$.

[2]To match the notations of prior works that include all clients in all training rounds, we use the abbreviation $N$ to denote the number of selected clients in each round in this section.

**Theorem 4.1** (One round Progress of CFL methods). *When $\{f_{t,i}(\boldsymbol{\omega})\}$ satisfy Assumption 1–4, and let $\eta = K\eta_g\eta_l$, we have the one round progress of CFL methods as,*

$$f(\boldsymbol{\omega}_t) - f(\boldsymbol{\omega}^*) \leq \underbrace{\frac{1}{\eta}(1 - \frac{\mu\eta}{2})\mathbb{E}\|\boldsymbol{\omega}_t - \boldsymbol{\omega}^*\|^2 - \frac{1}{\eta}\mathbb{E}\|\boldsymbol{\omega}_{t+1} - \boldsymbol{\omega}^*\|^2}_{A_1} + \underbrace{c_1\eta + c_2\eta^2}_{A_2} + \varphi_t \,, \tag{6}$$

*where $c_1 = 2c_{p_G} + \frac{\sigma^2}{NK} + 2c_R$, $c_2 = \frac{Lc_{p_G}}{3\eta_g^2} + \frac{L\sigma^2}{6\eta_g^2 K} + \frac{Lc_R}{3\eta_g^2}$, $c_{p_G} = \frac{1}{N}\sum_{i=1}^N G^2 + D^2(\sum_{\tau=1}^t p_{\tau,i}^2)$, $c_R = \frac{1}{N}\sum_{i=1}^N(\sum_{\tau=1}^{t-1} p_{\tau,i})^2 R^2$. $\varphi_t$ is a constant satisfying $\varphi_t = \mathcal{O}\left(\frac{1}{N}\sum_{i=1}^N\sum_{\tau=1}^{t-1} p_{t,i}R \|\boldsymbol{\omega}_0 - \boldsymbol{\omega}^*\|\right)$. $N$ is the number of chosen clients in each round, and $K$ is the number of local iterations.*

Theorem 4.1 shows the one round progress of CFL methods. The $A_1$ part of (6) indicates the linear convergence rate, while the $A_2$ part illustrates the worsened convergence induced by gradient noise and information loss. $\varphi_t$ is a constant related to the information loss that causes the drift of the optimum. When the upper limit on information loss $R$ exceeds 0, we cannot reach the exact optimum due to the approximation error: the iterates instead reach the neighborhood of $\epsilon + \varphi$, where $\varphi = \frac{\sum_{t=1}^T q_t\varphi_t}{\sum_{t=1}^T q_t}$ for some sequence $q_t$. We show in Section 5 that in practice, the approximation methods with small information loss converge much better than methods with larger information loss.

**Theorem 4.2** (Convergence rate of CFL methods). *Assume $\{f_{t,i}(\boldsymbol{\omega})\}$ satisfy Assumption 1–4, the output of Algorithm 1 has expected error smaller than $\epsilon + \varphi$, for $\eta_g = 1$, $\eta_l \leq \frac{\sqrt{3+4(1+B^2+A^2)} - \sqrt{4(1+B^2+A^2)}}{6KL\sqrt{1+B^2+A^2}}$, $p_{\tau,i} = \frac{D^2}{tD^2+(t-1)R^2}$ $(\tau < t)$, and $p_{t,i} = \frac{(t-1)R^2+D^2}{tD^2+(t-1)R^2}$ on round $t$. When $\{f_{t,i}(\boldsymbol{\omega})\}$ are $\mu$-strongly convex functions, we have,*

$$T = \mathcal{O}\left(\frac{Lc_O}{\mu} + \frac{\sigma^2}{\mu NK\epsilon} + \frac{c_A}{\mu\epsilon} + \sqrt{\frac{c_B}{\mu\epsilon}} + \sqrt{\frac{c_C}{\mu^2\epsilon}} + \sqrt[3]{\frac{c_D}{\mu^2\epsilon}}\right) \,, \tag{7}$$

*and when $\{f_{t,i}(\boldsymbol{\omega})\}$ are general convex functions ($\mu = 0$), we have*

$$T = \mathcal{O}\left(\frac{c_O F + \sqrt{c_B F} + \sqrt[3]{c_D F^2}}{\epsilon} + \frac{\sigma^2 F}{NK\epsilon^2} + \frac{c_A F}{\epsilon^2} + \sqrt{\frac{c_C F^2}{\epsilon^3}}\right) \,, \tag{8}$$

*and when $\{f_{t,i}(\boldsymbol{\omega})\}$ are non-convex, setting $\eta = K\eta_g\eta_l = \frac{\sqrt{KN}}{\sqrt{T}L}$, and when $\frac{1}{T}\sum_{t=1}^T \mathbb{E}\left[\|\nabla f(\boldsymbol{\omega}_t)\|^2\right]$ converge to $\epsilon$, we have*

$$T = \mathcal{O}\left(\frac{L^2(f_0 - f_*)^2}{NKc_m^2\epsilon^2} + \frac{1}{\epsilon^2}\left(\frac{\sqrt{KN}c_{R1}^2}{L} + \frac{\sigma^2}{\sqrt{NK}}\right)^2 + \frac{\sqrt{KN}c_{R2}}{c_m\epsilon L} + \frac{(KN)^{\frac{1}{3}}c_{R2}^{\frac{4}{3}}}{(\epsilon L)^{\frac{2}{3}}}\right) \,, \tag{9}$$

*where $c_O = 1 + A^2 + B^2$, $c_A = G^2 + \frac{D^2 R^2}{R^2 + D^2}$, $c_B = \frac{D^6}{(R^2+D^2)^2}$, $c_C = \frac{L\sigma^2}{\eta_g^2 K} + \frac{Lc_A}{\eta_g^2}$, $c_D = \frac{Lc_B}{\eta_g^2}$, $c_{R1} = \frac{R^2}{R^2+D^2}$, $c_{R2} = \frac{D^4}{(R^2+D^2)^2}$, $c_m$ is a constant related to A, B and R, and $F = \|\boldsymbol{\omega}_0 - \boldsymbol{\omega}^*\|^2$. $N$ is the number of chosen clients in each round, and $K$ is the number of local iterations. We elaborate the choice of $p_{t,i}$ in Appendix B.4.3.*

**Remark 4.3.** *When setting $N = 1$ in Theorem 4.2, we recover the convergence rate of standalone CL methods. To the best of our knowledge, we are the first to provide such theoretical guarantees for SGD: the recent work (Yin et al., 2020a) only gives the convergence rate of GD for regularization based CL methods on general convex case, under the constraint of $R = 0$.*

**Remark 4.4.** *For stateless FL scenarios (clients only appear once during training), we can derive tighter bounds than that of Theorem 4.2. Applying Lemma B.2 to the considered stateless scenario, $c_A$ and $c_B$ in Theorem 4.2 become $c_{A'} = \frac{(G^2+D^2)R^2}{G^2+D^2+R^2}$ and $c_{B'} = \frac{(D^2+G^2)^3}{(R^2+D^2+G^2)^2}$, where $\frac{c_A}{\mu\epsilon} + \sqrt{\frac{c_B}{\mu\epsilon}} \geq \frac{c_{A'}}{\mu\epsilon} + \sqrt{\frac{c_{B'}}{\mu\epsilon}}$ and $\frac{c_A}{\mu\epsilon} + \sqrt{\frac{c_B}{\mu\epsilon}}$ corresponds to the rate of strongly convex setting in Theorem 4.2.*

### 4.2 CONVERGENCE RATE OF FEDAVG UNDER TIME-VARYING SCENARIOS

To show the faster convergence of CFL methods than FedAvg, we provide the convergence rate of FedAvg under time-varying scenarios. Results are derived from Theorem 4.1, by setting $p_{t,i} = 1$. Note that when $p_{t,i} = 1$, we can set $\varphi_t$ to 0.

**Theorem 4.5** (Convergence rate of FedAvg under time-varying scenarios). *Assume $\{f_{t,i}(\boldsymbol{\omega})\}$ satisfy Assumption 1–4, the output of FedAvg has expected error smaller than $\epsilon$, for $\eta_g = 1$ and $\eta_l \leq \frac{\sqrt{3+4(1+B^2+A^2)}-\sqrt{4(1+B^2+A^2)}}{6KL\sqrt{1+B^2+A^2}}$. When $\{f_{t,i}(\boldsymbol{\omega})\}$ are $\mu$-strongly convex functions, we have,*

$$T = \mathcal{O}\left(\frac{Lc_O}{\mu} + \frac{\sigma^2}{\mu N K \epsilon} + \frac{c_A}{\mu \epsilon} + \sqrt{\frac{c_C}{\mu^2 \epsilon}}\right), \tag{10}$$

*and when $\{f_{t,i}(\boldsymbol{\omega})\}$ are general convex functions ($\mu = 0$), we have*

$$T = \mathcal{O}\left(\frac{c_O F}{\epsilon} + \frac{\sigma^2 F}{N K \epsilon^2} + \frac{c_A F}{\epsilon^2} + \sqrt{\frac{c_C F^2}{\epsilon^3}}\right), \tag{11}$$

*where $c_O = 1 + A^2 + B^2$, $c_A = G^2 + D^2$, $c_C = \frac{L\sigma^2}{\eta_g^2 K} + \frac{Lc_A}{\eta_g^2}$, and $F = \|\boldsymbol{\omega}_0 - \boldsymbol{\omega}^*\|^2$. $N$ is the number of chosen clients in each round, and $K$ is the number of local iterations.*

**Remark 4.6.** *CFL methods accelerates the convergence by reducing the variance term. As stated in Theorem 4.2 and Theorem 4.5, FedAvg (under time-varying scenarios) is a special case of CFL methods by setting $p_{t,i} = 1$, and we can show by proof (in Appendix B.4) that it is equivalent to setting the upper bound of information loss $R \to \infty$ (we can also observe this result by setting $R \to \infty$ in $p_{t,i} = \frac{(t-1)R^2 + D^2}{tD^2 + (t-1)R^2}$ in Theorem 4.2). The upper bound $R$ is linearly correlated to gradient noise, where $c_A = G^2 + \frac{D^2 R^2}{R^2 + D^2}$ in Theorem 4.2 increases to $c_A = G^2 + D^2$ in Theorem 4.5.*

## 5 EXPERIMENTS

In our numerical investigation, we first use the Noisy Quadratic Model—a simple convex model—to verify our theoretical results stated in Section 4. The results (in Appendix C.1) demonstrate that CFL methods converge much faster than baselines like FedAvg and FedProx, with a smoother convergence curve. These findings are corroborate Remark 4.6 about the variance reduction effect of CFL methods. We further provide extensive empirical evaluations below by comparing CFL methods with various strong FL competitors on different realistic datasets. We explore several approximation techniques in CFL methods and shed light on achieving practical efficient federated learning with time-varying data heterogeneity.

### 5.1 SETUP

**Time-varying heterogeneous local datasets.** We consider federated learn an image classifier using ResNet18 on split-CIFAR10 and split-CIFAR100 datasets, as well as a two layer MLP on the split-Fashion-MNIST dataset. The "split" follows the idea introduced in Yurochkin et al. (2019); Hsu et al. (2019); Reddi et al. (2021), where we leverage the Latent Dirichlet Allocation (LDA) to control the distribution drift with parameter $\alpha$ (See Algorithm 4). Larger $\alpha$ indicates smaller drifts.
Unless specifically mentioned otherwise our studies use the following protocol. All datasets are partitioned to 210 subsets for 7 distinct clients: all clients are selected and trained for 500 communication rounds, and each client randomly samples one of the corresponding 30 subsets for the local training—this challenging time-varying scenario mimics the realistic client data sampling scheme (from some underlying distributions). Notably, the training strategy used in this section is applicable to all FL baselines and CFL methods, and we assess the model performance using a global test dataset[3]. We carefully tune the hyper-parameters in all algorithms (details refer to Appendix C.2.1), and report the optimal results (i.e. mean test accuracy across the past 5 best epochs) after repeated trials.

**Approximation techniques in CFL methods.** Below, we review three representative types of information approximation techniques in CL, and use them to investigate the effect of various types of information loss under the (CFL) formulation. We refer to Appendix C.2.2 for a more comprehensive introduction and discussion.
- **Regularization methods.** Instead of keeping datasets from previous rounds, we keep track the gradients and Hessian matrices of local objective functions, and use Taylor Extension to approximate the local objective functions in previous rounds. Note that the trade-off between Hessian estimation and computational overhead constrains the practical feasible of such approach.
- **Core set methods.** Another simple yet effective treatment in CL lines in the category of Exemplar Replay (Rebuffi et al., 2017; Castro et al., 2018). This method selects and regularly saves previous core-set samples (a.k.a. exemplars), and replays them with the current local datasets.

---

[3]The training loss/accuracy on (CFL) formulation are closely aligned with that of global test data, as shown in Figure 5 of Appendix C.2.3. We here only present the global test results for the common interests in practice.

Table 2: **Top-1 accuracy for different choices of approximation techniques in CFL.** We train ResNet18 on split-CIFAR10 dataset (w/ $\alpha = 0.2$) for 300 communication rounds, and the dataset is partitioned to 300 subsets for 10 different clients. All examined algorithms use FedAvg as the backbone.

| Algorithm | Regularization Methods | | Core Set Methods | | | | Generative Methods |
|---|---|---|---|---|---|---|---|
| | PyHessian | Fisher Information Matrix | Naive (Small Set) | Naive (Large Set) | iCaRL (Small Set) | iCaRL (Large Set) | MCMC |
| Accuracy | $74.15 \pm 0.66$ | $73.70 \pm 0.39$ | $77.84 \pm 0.06$ | $\mathbf{78.90 \pm 0.09}$ | $76.97 \pm 0.16$ | $77.09 \pm 0.09$ | $74.18 \pm 0.08$ |

Table 3: **Top-1 accuracy of various CFL methods on diverse datasets** for training ResNet18 with 500 communication rounds. In order to observe a noticeable performance difference on Fashion-MNIST, we use $\alpha = 0.1$ instead. All examined algorithms use FedAvg as the backbone. Both CFL-Regularization and CFL-Regularization-Full are regularization based method, and the difference lies on where the regularization is applied: the full version applies regularization to all layers while the other only considers the top layers. $R_r$ indicates the accuracy of CFL-Regularization, while $R_f$ denotes the accuracy of FedAvg.

| Algorithm | Accuracy on Fashion-MNIST (%) | Accuracy on CIFAR10 (%) | Accuracy on CIFAR100 (%) |
|---|---|---|---|
| FedAvg | $86.75 \pm 0.14$ | $70.51 \pm 0.19$ | $49.97 \pm 0.19$ |
| CFL-Regularization | $87.02 \pm 0.21$ | $70.86 \pm 0.31$ | $50.69 \pm 0.06$ |
| CFL-Core-Set | $\mathbf{88.32 \pm 0.12}$ | $\mathbf{81.48 \pm 0.24}$ | $\mathbf{53.17 \pm 0.08}$ |
| CFL-Regularization-Full | $77.37 \pm 0.63$ | $33.09 \pm 1.45$ | $14.84 \pm 0.12$ |

| Metric | Improvement on Fashion-MNIST (%) | Improvement on CIFAR10 (%) | Improvement on CIFAR100 (%) |
|---|---|---|---|
| Absolute $(R_r - R_f)$ | 0.27 | 0.35 | 0.72 |
| Ratio $((R_r - R_f)/R_f)$ | 0.31 | 0.50 | 1.44 |

- **Generative methods.** Maintaining a core set for each client may become impractical when learning scales to millions of clients. To ensure a privacy-preserved federated learning, the generative models (Goodfellow et al., 2014) could be used locally to capture the local data distribution: fresh data samples will be generated on the fly and combined with the current local dataset. For the sake of simplicity, we use Markov Chain Monte Carlo (Nori et al., 2014) in our assessment.

## 5.2 RESULTS

**Comments on different CFL approximation techniques.** In Table 2, we examine the performance of several approximation techniques using the split-CIFAR10 dataset. We can conclude that (1) *Core set methods outperform other methods by a significant margin.* The simple choice of "naive core set", i.e. randomly and uniformly sample data from the local dataset, surpasses the sampling technique described in iCaRL (Rebuffi et al., 2017)[4] for CL, despite their faster convergence in the initial training phase. (2) *The quality of Hessian estimation matters for regularization based methods.* PyHessian (Yao et al., 2020), as a method to approximate diagonal Hessian matrix, is slightly preferable than Fisher Information Matrix, though the latter one involves less computation. (3) *The performance of generative methods is restricted*, and we hypothesize that the poor quality of generated samples contributes to the constraint.

In the subsequent evaluation, we consider naive core set sampling for CFL-Core-Set method, and use PyHessian for CFL-Regularization. We exclude the results of generative methods, due to the trivial performance gain and significant computational overhead.

**Understanding various approximated CFL implementations on different datasets.** Table 3 experimentally studies the impacts of various information approximation techniques in CFL methods—as discussed in Section 5.1—and compares them with the backbone algorithm of CFL methods, i.e. FedAvg. We have the following consistent findings on different datasets.

1. *The improvement of CFL methods over FedAvg becomes larger for more challenging tasks*, as shown in the bottom of the Table 3 for regularization based methods. This might reflect the fact that the precision of the information approximation is more crucial for complicated tasks. Similarly, we demonstrate in Figure 6 (in Appendix C.2.3) that *CFL methods offer better resistance to time-varying non-iid data* and the significance of such finding is depending on the task difficulty.

2. *For CFL-Regularization method, applying regularization terms on top layers works better than that on all layers.* This observation matches the recent work on decoupling feature extractor and classifier (Collins et al., 2021; Chen & Chao, 2021; Luo et al., 2021): the bottom layers are more generic across tasks and can serve as a global feature extractor, while the top layers are subject to task-specific information.

3. *The core-set method consistently outperforms FedAvg and other CFL approximation techniques by a large margin.*

We further investigate the connection between the information loss and the learning performance. We examine CFL methods with naive core sets, where the degree of information loss[5] can be changed

---

[4] The algorithmic details of the sampling method in iCaRL refer to Algorithm 3 in Appendix C.2.2.

[5] We estimate the information loss by using $\frac{1}{tS} \sum_{i=1}^{S} \sum_{\tau=1}^{t} \|\Delta_{\tau,i}(\boldsymbol{\omega})\|$, where $\Delta_{\tau,i}(\boldsymbol{\omega})$ is defined in Equation (3), $S$ is the number of clients chosen in each round, and $t$ is the number of communication rounds.

by altering the core set size from 20 to 150, with the same random seed and optimizers. Figure 1 depicts that the *performance of models is highly linked to the value of information loss: the lesser the information loss, the higher the performance.*

**Superior performance of CFL methods over other strong FL baselines.** Alongside the comparison between CFL methods and FedAvg on various datasets (Table 3), in Table 4 *we verify the efficacy of CFL methods over other strong FL baselines*, on split-CIFAR10 dataset. We also examine MimeLite (Karimireddy et al., 2020a), a method suggested for stateless FL scenarios. Note that we exclude the results illustration for methods like SCAFFOLD (Karimireddy et al., 2020b) and Ditto (Li et al., 2021), due to their infeasibility to be applied in our continual scenarios[6].

We further relax the difficulty of federated continual learning, from challenging non-overlapping time-varying heterogeneous data (e.g. in Table 2 and Table 3) to a moderate time-varying case (i.e. the local data evolves with the overlapping, while the size of local datasets stay unchanged). Table 5 illustrates the performance of FL baselines and CFL methods, under different degrees of overlapping (the construction details refers to Appendix C.2.1): *the improvement of CFL methods is consistent to our previous results, while the overlap parameter has no obvious connection with the final global test performance.* We believe that both the overlap degree and the new arriving data influence final performance, and we leave future work on realistic time-varying FL datasets to gain a better understanding.

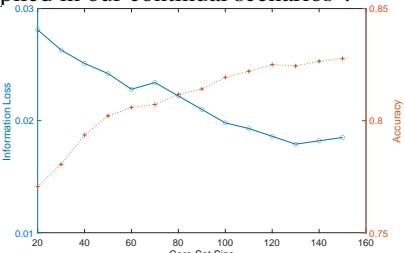

Figure 1: Information loss (left y-axis) and accuracy (right y-axis) for training ResNet18 on split-CIFAR10 with different core set sizes (x-axis) and $\alpha = 0.2$.

Table 4: **Comparing the SOTA FL baselines with several CFL methods**, for training ResNet18 on split-CIFAR10 dataset with different degrees of non-iid-ness $\alpha$ (and with total 500 communication rounds).

| Accuracy | FL baselines | | | CFL methods | | |
|---|---|---|---|---|---|---|
| | FedAvg | FedProx | MimeLite | CFL-Core-Set | CFL-Regularization | CFL-Regularization + FedProx |
| $\alpha = 0.1$ | $70.51 \pm 0.19$ | $71.57 \pm 0.17$ | $69.55 \pm 0.36$ | $\mathbf{81.48 \pm 0.24}$ | $70.86 \pm 0.31$ | $71.50 \pm 0.07$ |
| $\alpha = 0.2$ | $77.98 \pm 0.36$ | $78.40 \pm 0.22$ | $78.65 \pm 0.26$ | $\mathbf{84.41 \pm 0.11}$ | $78.76 \pm 0.26$ | $79.04 \pm 0.01$ |

Table 5: **Benchmarking FL baselines and CFL methods on different degrees of local dataset overlapping**, for training ResNet18 on split-CIFAR10 dataset. The overlap reduces the degree of non-iid-ness. In order to observe a noticeable performance difference, we use a larger data distribution gap between rounds (i.e. $\alpha = 0.1$).

| Overlap | FL baselines | | CFL methods | |
|---|---|---|---|---|
| | FedAvg | MimeLite | CFL-Core-Set | CFL-Regularization |
| 0% | $80.66 \pm 0.40$ | $80.78 \pm 0.07$ | $\mathbf{84.87 \pm 0.11}$ | $81.27 \pm 0.38$ |
| 25% | $80.16 \pm 0.01$ | $80.17 \pm 0.11$ | $\mathbf{84.66 \pm 0.11}$ | $80.67 \pm 0.27$ |
| 50% | $79.97 \pm 0.23$ | $80.35 \pm 0.39$ | $\mathbf{84.59 \pm 0.07}$ | $80.91 \pm 0.25$ |

**Investigating other time-varying scenarios.** In previous numerical investigations, we examine the *stateful clients* and assume that these clients would learn and retain their client states throughout the training process. The remainder of this section will discuss how to learn with *stateless clients* (i.e. each client only appear once). To do this, we change the data partition strategy such that fresh data partitions are sampled on the fly for each client and communication round.

Table 6 evaluates the above-mentioned scenario. We find that *CFL-Regularization methods consistently outperform alternative FL baselines*, similar to the observations in Table 4. However, due to the nature of stateless under the privacy concern, the advancements of core set method in CFL algorithm cannot be seen in this scenario. Additionally, we discuss the applicability of different algorithms in a variety of time-varying scenarios in Table 8 of Appendix C.2.3.

Table 6: **Learning with stateless clients**, regarding training ResNet18 on split-CIFAR10 with $\alpha = 0.2$. All examined algorithms use FedAvg as the backbone. The details w.r.t. CFL-Regularization refer to Appendix C.2.2.

| Algorithm | FL baselines | | | CFL methods | |
|---|---|---|---|---|---|
| | FedAvg | MimeLite | FedProx | CFL-Regularization | CFL-Regularization + FedProx |
| Accuracy | $78.07 \pm 0.06$ | $78.47 \pm 0.19$ | $78.48 \pm 0.05$ | $79.07 \pm 0.25$ | $\mathbf{79.32 \pm 0.07}$ |

---

[6]We also naively adapted and examined these methods, but we cannot observe significant performance gains.

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

CONTENTS OF APPENDIX

## A TECHNIQUES

Here we show some technical lemmas which are helpful in the theoretical proof.

**Lemma A.1** (Linear convergence rate (Karimireddy et al., 2020b, Lemma 1)). *For every non-negative sequence $\{d_{r-1}\}_{r\geq 1}$, and any parameter $\mu > 0$, $T \geq \frac{1}{2\eta_{max}\mu}$, there exists a constant step-size $\eta \leq \eta_{max}$ and weights $\omega_t = (1-\mu\eta)^{1-t}$ such that for $W_T = \sum_{t=1}^{T+1}\omega_t$*

$$\Phi_T = \frac{1}{W_T}\sum_{t=1}^{T+1}\left(\frac{\omega_{t-1}}{\eta}(1-\mu\eta)d_{t-1} - \frac{\omega_t}{\eta}d_t\right) = \tilde{\mathcal{O}}\left(\mu d_0 exp(-\mu\eta_{max}T)\right). \quad (12)$$

**Lemma A.2** (Sub-linear convergence rate (Karimireddy et al., 2020b, Lemma 2)). *For every non-negative sequence $\{d_{r-1}\}_{r\geq 1}$ and any parameters $\eta_{max} > 0$, $c_1 \geq 0$, $c_2 \geq 0$, $T \geq 0$, there exists a constant step-size $\eta \leq \eta_{max}$ such that*

$$\Phi_T = \frac{1}{T+1}\sum_{t=1}^{T+1}\left(\frac{d_{t-1}}{\eta} - \frac{d_t}{\eta} + c_1\eta + c_2\eta^2\right) \leq \frac{d_0}{\eta_{max}(T+1)} + \frac{2\sqrt{c_1 d_0}}{\sqrt{T+1}} + 2\left(\frac{d_0}{T+1}\right)^{\frac{2}{3}}c_2^{\frac{1}{3}}. \quad (13)$$

**Lemma A.3** (Relaxed triangle inequality (Karimireddy et al., 2020b, Lemma 3)). *Let $\{\mathbf{v}_1, ..., \mathbf{v}_\tau\}$ be $\tau$ vectors in $\mathcal{R}^d$. Then the following is true:*

$$\|\frac{1}{\tau}\sum_{i=1}^{\tau}\mathbf{v}_i\|^2 \leq \frac{1}{\tau}\sum_{i=1}^{\tau}\|\mathbf{v}_i\|^2. \quad (14)$$

**Lemma A.4** (Separating mean and variance (Stich & Karimireddy, 2020, Lemma 14)). *Let $\Xi_1, \Xi_2, ..., \Xi_\tau$ be $\tau$ random variables in $\mathcal{R}^d$ which are not necessarily independent. First suppose that their mean is $E[\Xi_i] = \xi_i$, and variance is bounded as $\mathbb{E}\|\Xi_i - \xi_i\|^2 \leq M\|\xi_i\|^2 + \sigma^2$, then the following holds*

$$\mathbb{E}\left\|\sum_{i=1}^{\tau}\Xi_i\right\|^2 \leq (\tau + M)\sum_{i=1}^{\tau}\|\xi_i\|^2 + \tau\sigma^2. \quad (15)$$

**Lemma A.5** (Perturbed strongly convexity (Karimireddy et al., 2020b, Lemma 5)). *The following holds for any $L$-smooth and $\mu$-strongly convex function $h$, and any $\mathbf{x}$, $\mathbf{y}$, $\mathbf{z}$ in the domain of $h$:*

$$\langle \nabla h(\mathbf{x}), \mathbf{z} - \mathbf{y} \rangle \geq h(\mathbf{z}) - h(\mathbf{y}) + \frac{\mu}{4}\|\mathbf{y} - \mathbf{z}\|^2 - L\|\mathbf{z} - \mathbf{x}\|^2. \quad (16)$$

**Lemma A.6.** *Define $S = \sum_{t=1}^{T}\frac{p_t}{t}$ where $\sum_{t=1}^{T}p_t = 1$, and $p_t \leq p_{t+1}$ then $S \leq \frac{1}{T}\sum_{t=1}^{T}\frac{1}{t} = \tilde{\mathcal{O}}\left(\frac{lnT}{T}\right)$.*

# B CONVERGENCE RATE OF CFL

## B.1 BOUND OF GRADIENT NOISE

**Lemma B.1** (Bound of Gradient Noise). *Suppose $f_i(\boldsymbol{\omega})$ is the local objective function, and $f(\boldsymbol{\omega}) = \mathbb{E}\left[f_i(\boldsymbol{\omega})\right]$ is the global objective function. Define $\nabla f(\boldsymbol{\omega}) = \nabla f_i(\boldsymbol{\omega}) + \boldsymbol{\varsigma}_i$, $\mathbb{E}[\boldsymbol{\varsigma}_i] = 0$, and assume $\mathbb{E}\left[\|\boldsymbol{\varsigma}\|^2 \,|\, \boldsymbol{\omega}\right] \le A^2 \mathbb{E}\left[\|\nabla f(\boldsymbol{\omega})\|^2\right] + B^2$, we have*

$$\mathbb{E}\left[\|\nabla f_i(\boldsymbol{\omega})\|^2\right] \le (A^2 + 1)\mathbb{E}\left[\|\nabla f(\boldsymbol{\omega})\|^2\right] + B^2. \tag{17}$$

*Proof.*

$$\mathbb{E}\left[\|\nabla f_i(\boldsymbol{\omega})\|^2\right] = \mathbb{E}\left[\|\nabla f(\boldsymbol{\omega}) + \boldsymbol{\varsigma}_i\|^2\right] \tag{18}$$

$$= \mathbb{E}\left[\|\nabla f(\boldsymbol{\omega})\|^2\right] + \mathbb{E}\left[\|\boldsymbol{\varsigma}_i\|^2\right] \tag{19}$$

$$\le (A + 1)\mathbb{E}\left[\|\nabla f(\boldsymbol{\omega})\|^2\right] + B. \tag{20}$$

$\square$

## B.2 BOUNDED GRADIENT NOISE OF CFL

**Lemma B.2** (Bounded Gradient Noise of CFL). *For Formulation* (CFL)*, consider CFL and FedAvg, we can bound the gradient drift as*
*(1) FedAvg only optimize current local objective functions, thus,*

$$\mathbb{E}\|\nabla f_{t,i}(\boldsymbol{\omega})\|^2 \le (1 + A^2 + B^2)\mathbb{E}\|\nabla f(\boldsymbol{\omega})\|^2 + G^2 + D^2.$$

*(2) For CFL, in general (same clients can appear in different rounds), we have*

$$\mathbb{E}\left\|\sum_{\tau=1}^t p_{\tau,i}\nabla f_{\tau,i}(\boldsymbol{\omega})\right\|^2 \le \left(1 + B^2 + A^2\sum_{\tau=1}^t p_{\tau,i}^2\right)\mathbb{E}\|\nabla f(\boldsymbol{\omega})\|^2$$
$$+ G^2 + D^2\sum_{\tau=1}^t p_{\tau,i}^2.$$

*(3) For CFL, when clients only participate in training once($\boldsymbol{\delta}_{t,i}$ are independent for all $t$ and $i$), we have*

$$\mathbb{E}\left\|\sum_{\tau=1}^t p_{\tau,i}\nabla f_{\tau,i}(\boldsymbol{\omega})\right\|^2 \le \left(1 + (B^2 + A^2)\sum_{\tau=1}^t p_{\tau,i}^2\right)\mathbb{E}\|\nabla f(\boldsymbol{\omega})\|^2$$
$$+ (G^2 + D^2)\sum_{\tau=1}^t p_{\tau,i}^2.$$

*Proof.* Using Assumption 3, together with the fact that $\nabla f_{t,i}(\boldsymbol{\omega}) = \nabla f(\boldsymbol{\omega}) + \boldsymbol{\delta}_{t,i} + \boldsymbol{\xi}_{t,i}$, we have

$$\mathbb{E}\|\nabla f_{t,i}(\boldsymbol{\omega})\|^2 = \mathbb{E}\|\nabla f(\boldsymbol{\omega}) + \boldsymbol{\delta}_{t,i} + \boldsymbol{\xi}_{t,i}\|^2 \tag{21}$$

$$= \mathbb{E}\|\nabla f(\boldsymbol{\omega})\|^2 + \mathbb{E}\|\boldsymbol{\delta}_{t,i}\|^2 + \mathbb{E}\|\boldsymbol{\xi}_{t,i}\|^2 \tag{22}$$

$$\le (1 + A^2 + B^2)\mathbb{E}\|\nabla f(\boldsymbol{\omega})\|^2 + G^2 + D^2. \tag{23}$$

The second inequality is based on the independence of noise, and directly use Assumption 3 we get the last inequality. Similarly, we have

$$\mathbb{E}\left\|\sum_{\tau=1}^t p_{\tau,i}\nabla f_{\tau,i}(\boldsymbol{\omega})\right\|^2 = \mathbb{E}\left\|\sum_{\tau=1}^t p_{\tau,i}(\nabla f_{\tau,i}(\boldsymbol{\omega}) + \boldsymbol{\delta}_i + \boldsymbol{\xi}_{\tau,i})\right\|^2 \tag{24}$$

$$= \mathbb{E}\|\nabla f(\boldsymbol{\omega})\|^2 + \mathbb{E}\|\boldsymbol{\delta}_i\|^2 + \mathbb{E}\left\|\sum_{\tau=1}^t p_{\tau,i}\boldsymbol{\xi}_{\tau,i}\right\|^2 \tag{25}$$

$$\le \left(1 + B^2 + A^2\sum_{\tau=1}^t p_{\tau,i}^2\right)\mathbb{E}\|\nabla f(\boldsymbol{\omega})\|^2 + G^2 + D^2\sum_{\tau=1}^t p_{\tau,i}^2. \tag{26}$$

For the last inequality, when $\boldsymbol{\delta}_{t_1,i}$ and $\boldsymbol{\delta}_{t_2,i}$ are independent for all $t_1$, $t_2$,

$$\mathbb{E}\left\|\sum_{\tau=1}^{t} p_{\tau,i}\nabla f_{\tau,i}(\boldsymbol{\omega})\right\|^2 = \mathbb{E}\left\|\sum_{\tau=1}^{t} p_{\tau,i}(\nabla f_{\tau,i}(\boldsymbol{\omega}) + \boldsymbol{\delta}_{t,i} + \boldsymbol{\xi}_{\tau,i})\right\|^2 \tag{27}$$

$$= \mathbb{E}\|\nabla f(\boldsymbol{\omega})\|^2 + \mathbb{E}\left\|\sum_{\tau=1}^{t} p_{\tau,i}\boldsymbol{\delta}_{t,i}\right\|^2 + \mathbb{E}\left\|\sum_{\tau=1}^{t} p_{\tau,i}\boldsymbol{\xi}_{\tau,i}\right\|^2 \tag{28}$$

$$\leq \left(1 + (A^2 + B^2)\sum_{\tau=1}^{t} p_{\tau,i}^2\right)\mathbb{E}\|\nabla f(\boldsymbol{\omega})\|^2 + (G^2 + D^2)\sum_{\tau=1}^{t} p_{\tau,i}^2. \tag{29}$$

$\square$

### B.3 BOUNDED APPROXIMATION ERROR

**Lemma B.3** (Bounded Approximation Error). *For $\Delta_{t,i}(\boldsymbol{\omega}) = \nabla f_{t,i}(\boldsymbol{\omega}) - \nabla \tilde{f}_{t,i}(\boldsymbol{\omega})$ (c.f. Definition 3), when use Taylor Extension,*

$$\nabla \tilde{f}_{t,i}(\boldsymbol{\omega}) = \nabla f_{t,i}(\boldsymbol{\omega}_t) + \nabla^2 f_{t,i}(\boldsymbol{\omega}_t)(\boldsymbol{\omega} - \boldsymbol{\omega}_t),$$

*we have*

$$\|\Delta_{t,i}(\boldsymbol{\omega})\| \leq \epsilon \|\boldsymbol{\omega} - \hat{\boldsymbol{\omega}}_{t,i}\|.$$

*Proof.*

$$\|\Delta_{t,i}(\boldsymbol{\omega})\| = \left\|\nabla f_{t,i}(\boldsymbol{\omega}) - \nabla \tilde{f}_{t,i}(\boldsymbol{\omega})\right\| \tag{30}$$

$$= \|\nabla f_{t,i}(\boldsymbol{\omega}) - \nabla f_{t,i}(\boldsymbol{\omega}_{t,i,K}) - \mathbf{H}_{tjK}(\boldsymbol{\omega} - \boldsymbol{\omega}_{t,i,K})\| \tag{31}$$

$$= \left\|\nabla^2 f_{t,i}(\boldsymbol{\omega}_{\boldsymbol{\xi},t})(\boldsymbol{\omega} - \boldsymbol{\omega}_{t,i,K}) - \mathbf{H}_{tjK}(\boldsymbol{\omega} - \boldsymbol{\omega}_{t,i,K})\right\| \tag{32}$$

$$= \left\|(\nabla^2 f_{t,i}(\boldsymbol{\omega}_{\boldsymbol{\xi},t}) - \mathbf{H}_{tjK})(\boldsymbol{\omega} - \boldsymbol{\omega}_{t,i,K})\right\| \tag{33}$$

$$\leq \epsilon \|\boldsymbol{\omega} - \boldsymbol{\omega}_{t,i,K}\|. \tag{34}$$

The first two equations come from the mean value theorem which says for continuous function $f$ in closed intervals $[a, b]$ and differentiable on open interval $(a, b)$, there exists a point $c \subseteq (a, b)$ such that

$$f'(c) = \frac{f(b) - f(a)}{b - a}. \tag{35}$$

The last inequality is based on Assumption 4. $\square$

### B.4 PROOF OF THEOREM 4.2

In this section we will give the complete proof of convergence rate of algorithm 2 when the local objective functions are convex.

---

**Algorithm 2** CFL Framework

---

**Require:** initial weights $\boldsymbol{\omega}_0$, global learning rate $\eta_g$, local learning rate $\eta_l$

1: **for** round $t = 1, \ldots, T$ **do**
2:      **communicate** $\boldsymbol{\omega}_t$ to the chosen clients.
3:      **for client** $i \in \mathcal{S}_t$ **in parallel do**
4:          initialize local model $\boldsymbol{\omega}_{t,i,0} = \boldsymbol{\omega}_t$.
5:          **for** $k = 1, \ldots, K$ **do**
6:              $\tilde{g}_{t,i,k}(\boldsymbol{\omega}_{t,i,k-1}) = \left(p_{t,i}\nabla f_{t,i}(\boldsymbol{\omega}_{t,i,k-1}) + \sum_{\tau=1}^{t-1} p_{\tau,i}\nabla \tilde{f}_{t,i}(\boldsymbol{\omega}_{t,i,k-1})\right) + \boldsymbol{\nu}_{t,i,k}$.
7:              $\boldsymbol{\omega}_{t,i,k} \leftarrow \boldsymbol{\omega}_{t,i,k-1} - \eta_l \tilde{g}_{t,i,k}(\boldsymbol{\omega}_{t,i,k-1})$.
8:          **communicate** $\Delta\boldsymbol{\omega}_{t,i} \leftarrow \boldsymbol{\omega}_{t,i,K} - \boldsymbol{\omega}_t$.
9:      $\Delta\boldsymbol{\omega}_t \leftarrow \frac{\eta_g}{S}\sum_{i \in \mathcal{S}_t} \Delta\boldsymbol{\omega}_{t,i}$.
10:     $\boldsymbol{\omega}_{t+1} \leftarrow \boldsymbol{\omega}_t + \Delta\boldsymbol{\omega}_t$.

---

### B.4.1 START UP.

The local objective function of round $t$ on client $i$ is

$$\bar{f}_{t,i}(\boldsymbol{\omega}) = (p_{t,i}f_{t,i}(\boldsymbol{\omega}) + \sum_{\tau=1}^{t-1} p_{\tau,i}\tilde{f}_{\tau,i}(\boldsymbol{\omega})\,. \tag{36}$$

Without loss of generality, assume $\sum_{\tau=1}^{t} p_{\tau,i} = 1$. Then follow the steps in algorithm 2, we have

$$\nabla\bar{f}_{t,i}(\boldsymbol{\omega}) = p_{t,i}\nabla f_{t,i}(\boldsymbol{\omega}) + \sum_{\tau=1}^{t-1} p_{\tau,i}\nabla\tilde{f}_{\tau,i}(\boldsymbol{\omega})\,. \tag{37}$$

Then the noisy gradient in mini-batch SGD can be described as

$$\bar{g}_{t,i,k}(\boldsymbol{\omega}_{t,i,k-1}) = \nabla\bar{f}_{t,i}(\boldsymbol{\omega}_{t,i,k-1}) + \boldsymbol{\nu}_{t,i,k} \tag{38}$$

Then follow the steps in algorithm 2, we have

$$\Delta\boldsymbol{\omega}_t = \frac{-\eta_l\eta_g}{N} \sum_{i=1}^{N}\sum_{k=1}^{K} \bar{g}_{t,i}(\boldsymbol{\omega}_{t,i,k-1})\,, \tag{39}$$

$$E[\Delta\boldsymbol{\omega}_t] = \frac{-\eta_l\eta_g}{N} \sum_{i=1}^{N}\sum_{k=1}^{K} \nabla\bar{f}_{t,i}(\boldsymbol{\omega}_{t,i,k-1})\,. \tag{40}$$

Then let $\eta = K\eta_l\eta_g$, we have

$$\Delta\boldsymbol{\omega}_t = \frac{-\eta}{NK} \sum_{i=1}^{N}\sum_{k=1}^{K} \bar{g}_{t,i}(\boldsymbol{\omega}_{t,i,k-1})\,, \tag{41}$$

$$E[\Delta\boldsymbol{\omega}_t] = \frac{-\eta}{NK} \sum_{i=1}^{N}\sum_{k=1}^{K} \nabla\bar{f}_{t,i}(\boldsymbol{\omega}_{t,i,k-1})\,. \tag{42}$$

### B.4.2 ONE STEP PROGRESS

Consider the one-step progress we have

$$\mathbb{E}\|\boldsymbol{\omega}_t + \Delta\boldsymbol{\omega}_t - \boldsymbol{\omega}^*\|^2 = \|\boldsymbol{\omega}_t - \boldsymbol{\omega}^*\|^2 - \underbrace{2E\langle\boldsymbol{\omega}_t - \boldsymbol{\omega}^*, \Delta\boldsymbol{\omega}_t\rangle}_{A_1} + \underbrace{\mathbb{E}\|\Delta\boldsymbol{\omega}_t\|^2}_{A_2}\,. \tag{43}$$

Firstly we consider $A_1$ part in equation (43),

$$- 2E\langle\boldsymbol{\omega}_t - \boldsymbol{\omega}^*, \Delta\boldsymbol{\omega}_t\rangle$$

$$= \frac{2\eta}{NK} \sum_{i=1}^{N}\sum_{k=1}^{K} \langle\nabla\bar{f}_{t,i}(\boldsymbol{\omega}_{t,i,k-1}), \boldsymbol{\omega}^* - \boldsymbol{\omega}_t\rangle \tag{44}$$

$$= \frac{2\eta}{NK} \sum_{i=1}^{N}\sum_{k=1}^{K} \langle p_{t,i}\nabla f_{t,i}(\boldsymbol{\omega}_{t,i,k}) + \sum_{\tau=1}^{t-1} p_{\tau,i}\nabla\tilde{f}_{\tau,i}(\boldsymbol{\omega}_{t,i,k}), \boldsymbol{\omega}^* - \boldsymbol{\omega}_t\rangle \tag{45}$$

$$= \frac{2\eta}{NK} \sum_{i=1}^{N}\sum_{k=1}^{K}\sum_{\tau=1}^{t} \langle p_{\tau,i}\nabla f_{\tau,i}(\boldsymbol{\omega}_{t,i,k}), \boldsymbol{\omega}^* - \boldsymbol{\omega}_t\rangle - \frac{2\eta}{NK} \sum_{i=1}^{N}\sum_{k=1}^{K}\sum_{\tau=1}^{t-1} p_{\tau,i}\langle\Delta_{\tau,i}(\boldsymbol{\omega}_{t,i,k}), \boldsymbol{\omega}^* - \boldsymbol{\omega}_t\rangle \tag{46}$$

$$\leq \frac{2\eta}{NK} \sum_{i=1}^{N}\sum_{k=1}^{K}\sum_{\tau=1}^{t} p_{\tau,i}\langle\nabla f_{\tau,i}(\boldsymbol{\omega}_{t,i,k}), \boldsymbol{\omega}^* - \boldsymbol{\omega}_t\rangle + \frac{2\eta}{N} \sum_{i=1}^{N}\sum_{\tau=1}^{t-1} p_{\tau,i}R\|\boldsymbol{\omega}^* - \boldsymbol{\omega}_t\|\,. \tag{47}$$

We use equality (42) for first inequality, and directly use Assumption 4 and Cauchy–Schwarz inequality for last inequality.

Then we can consider the first term of inequality (47). Firstly, by Lemma A.5 we see that

$$\langle\nabla f_{\tau,i}(\boldsymbol{\omega}_{t,i,k-1}), \boldsymbol{\omega}^* - \boldsymbol{\omega}_t\rangle \leq f_{\tau,i}(\boldsymbol{\omega}^*) - f_{\tau,i}(\boldsymbol{\omega}_t) - \frac{\mu}{4}\|\boldsymbol{\omega}_t - \boldsymbol{\omega}^*\|^2 + L\|\boldsymbol{\omega}_{t,i,k-1} - \boldsymbol{\omega}_t\|^2\,. \tag{48}$$

That is, based on (47) and (48), we have,

$$
\begin{aligned}
&- 2E\langle \boldsymbol{\omega}_t - \boldsymbol{\omega}^*, \Delta\boldsymbol{\omega}_t\rangle \\
&\leq -2\eta(f_t(\boldsymbol{\omega}_t) - f_t(\boldsymbol{\omega}^*) + \frac{\mu}{4}\|\boldsymbol{\omega}_t - \boldsymbol{\omega}^*\|^2) + \frac{2\eta L}{NK}\sum_{i=1}^{N}\sum_{k=1}^{K}\|\boldsymbol{\omega}_{t,i,k-1} - \boldsymbol{\omega}_t\|^2 \\
&+ 2\eta\sum_{\tau=1}^{t-1}p_{\tau,i}R\|\boldsymbol{\omega}^* - \boldsymbol{\omega}_t\| .
\end{aligned}
\tag{49}
$$

Then consider $A_2$ in (43), and define $c_{p_B,i} = 1+B^2+A^2(\sum_{\tau=1}^{t}p_{\tau,i}^2)$, $c_{p_G,i} = G^2+D^2(\sum_{\tau=1}^{t}p_{\tau,i}^2)$, $c_{p_B} = \frac{1}{N}\sum_{i=1}^{N}c_{p_B,i}$, and $c_{p_G} = \frac{1}{N}\sum_{i=1}^{N}c_{p_G,i}$ we have

$$
\mathbb{E}\|\Delta\boldsymbol{\omega}_t\|^2
$$
$$
= \frac{\eta^2}{N^2K^2}\mathbb{E}\left\|\sum_{i=1}^{N}\sum_{k=1}^{K}\left(p_{t,i}\nabla f_{t,i}(\boldsymbol{\omega}_{t,i,k-1}) + \sum_{\tau=1}^{t-1}p_{\tau,i}\nabla\tilde{f}_{\tau,i}(\boldsymbol{\omega}_{t,i,k-1}) + \boldsymbol{\nu}_{t,i,k}\right)\right\|^2
\tag{50}
$$
$$
= \frac{\eta^2}{N^2K^2}\mathbb{E}\left\|\sum_{i=1}^{N}\sum_{k=1}^{K}\left(\sum_{\tau=1}^{t}p_{\tau,i}\nabla f_{\tau,i}(\boldsymbol{\omega}_{t,i,k-1}) - \sum_{\tau=1}^{t-1}p_{\tau,i}\Delta_{\tau,i}(\boldsymbol{\omega}_{t,i,k-1}) + \boldsymbol{\nu}_{t,i,k}\right)\right\|^2
\tag{51}
$$
$$
\overset{Lem\ A.4}{\leq} \frac{\eta^2}{NK}\sum_{i=1}^{N}\sum_{k=1}^{K}\mathbb{E}\left\|\sum_{\tau=1}^{t}p_{\tau,i}\nabla f_{\tau,i}(\boldsymbol{\omega}_{t,i,k-1}) - \sum_{\tau=1}^{t-1}p_{\tau,i}\Delta_{\tau,i}(\boldsymbol{\omega}_{t,i,k-1})\right\|^2 + \frac{\eta^2\sigma^2}{NK}
\tag{52}
$$
$$
\overset{Lem\ A.3}{\leq} \frac{2\eta^2}{NK}\sum_{i=1}^{N}\sum_{k=1}^{K}\left(\mathbb{E}\left\|\sum_{\tau=1}^{t}p_{\tau,i}\nabla f_{\tau,i}(\boldsymbol{\omega}_{t,i,k-1})\right\|^2 + \mathbb{E}\left\|\sum_{\tau=1}^{t-1}p_{\tau,i}\Delta_{\tau,i}(\boldsymbol{\omega}_{t,i,k-1})\right\|^2\right) + \frac{\eta^2\sigma^2}{NK}
\tag{53}
$$
$$
\overset{Lem\ ??}{\leq} \frac{2\eta^2}{NK}\sum_{i=1}^{N}\sum_{k=1}^{K}\left(\left(1+B^2+A^2(\sum_{\tau=1}^{t}p_{\tau,i}^2)\right)\mathbb{E}\|\nabla f(\boldsymbol{\omega}_{t,i,k-1})\|^2 + \left(G^2+D^2(\sum_{\tau=1}^{t}p_{\tau,i}^2)\right)\right) + \frac{2\eta^2}{N}\sum_{i=1}^{N}(\sum_{\tau=1}^{t-1}p_{\tau,i})^2R^2 + \frac{\eta^2\sigma^2}{NK}
\tag{54}
$$
$$
\overset{Lem\ A.3}{\leq} \frac{4\eta^2}{NK}\sum_{i=1}^{N}\sum_{k=1}^{K}c_{p_B,i}\left(\mathbb{E}\|\nabla f(\boldsymbol{\omega}_{t,i,k-1}) - \nabla f(\boldsymbol{\omega}_t)\|^2 + \mathbb{E}\|\nabla f(\boldsymbol{\omega}_t)\|^2\right) + 2\eta^2 c_{p_G} + \frac{2\eta^2}{N}\sum_{i=1}^{N}(\sum_{\tau=1}^{t-1}p_{\tau,i})^2R^2 + \frac{\eta^2\sigma^2}{NK}
\tag{55}
$$
$$
\overset{Ass\ 1}{\leq} 16\eta^2 L c_{p_B}(f(\boldsymbol{\omega}_t) - f(\boldsymbol{\omega}^*)) + \frac{8\eta^2 L^2}{NK}\sum_{i=1}^{N}\sum_{k=1}^{K}c_{p_B,i}\mathbb{E}\|\boldsymbol{\omega}_{t,i,k-1} - \boldsymbol{\omega}_t\|^2 + 2\eta^2 c_{p_G} + \frac{2\eta^2}{N}\sum_{i=1}^{N}(\sum_{\tau=1}^{t-1}p_{\tau,i})^2R^2 + \frac{\eta^2\sigma^2}{NK} .
\tag{56}
$$

Then consider equation (49) and (56), to solve $\mathbb{E}\|\boldsymbol{\omega}_{t,i,k-1} - \boldsymbol{\omega}_t\|^2$, we have

$$
\mathbb{E}\|\boldsymbol{\omega}_{t,i,k} - \boldsymbol{\omega}_t\|^2 = \eta_l^2\mathbb{E}\left\|\sum_{\tau=1}^{k}\bar{g}_{t,i}(\boldsymbol{\omega}_{t,i,\tau-1})\right\|^2
\tag{57}
$$
$$
\overset{Lem\ A.4}{\leq} k\eta_l^2\sum_{\tau=1}^{k}\mathbb{E}\left\|\nabla\bar{f}_{t,i}(\boldsymbol{\omega}_{t,i,\tau-1})\right\|^2 + k\eta_l^2\sigma^2 .
\tag{58}
$$

Here we directly use Lemma A.4 to get inequality (58).

Then for $\mathbb{E}\big\|\nabla\bar{f}_{t,i}(\boldsymbol{\omega})\big\|^2$, we have

$$\mathbb{E}\big\|\nabla\bar{f}_{t,i}(\boldsymbol{\omega})\big\|^2$$

$$= \mathbb{E}\left\|p_{t,i}\nabla f_{t,i}(\boldsymbol{\omega}) + \sum_{\tau=1}^{t-1} p_{\tau,i}\nabla\tilde{f}_{\tau,i}(\boldsymbol{\omega})\right\|^2 \tag{59}$$

$$= \mathbb{E}\left\|\sum_{\tau=1}^{t} p_{\tau,i}\nabla f_{\tau,i}(\boldsymbol{\omega}) - \sum_{\tau=1}^{t-1} p_{\tau,i}\Delta_{\tau,i}(\boldsymbol{\omega})\right\|^2 \tag{60}$$

$$\overset{Lem\ A.3}{\leq} 2\mathbb{E}\left\|\sum_{\tau=1}^{t} p_{\tau,i}\nabla f_{\tau,i}(\boldsymbol{\omega})\right\|^2 + 2\mathbb{E}\left\|\sum_{\tau=1}^{t-1} p_{\tau,i}\Delta_{\tau,i}(\boldsymbol{\omega})\right\|^2 \tag{61}$$

$$\overset{Lem\ ??}{\leq} 2c_{p_B,i}\mathbb{E}\|\nabla f(\boldsymbol{\omega})\|^2 + 2c_{p_G,i} + 2(\sum_{\tau=1}^{t-1} p_{\tau,i})^2 R^2 \tag{62}$$

$$\overset{Ass\ 1}{\leq} 4L^2 c_{p_B,i}\mathbb{E}\|\boldsymbol{\omega} - \boldsymbol{\omega}_t\|^2 + 8Lc_{p_B,i}\left(f(\boldsymbol{\omega}_t) - f(\boldsymbol{\omega}^*)\right) + 2c_{p_G,i} + 2(\sum_{\tau=1}^{t-1} p_{\tau,i})^2 R^2 . \tag{63}$$

Combine equation (58) and (63) we get,

$$\mathbb{E}\|\boldsymbol{\omega}_{t,i,k} - \boldsymbol{\omega}_t\|^2$$

$$\leq 4k^2\eta_l^2 L^2 c_{p_B,i}\mathbb{E}\|\boldsymbol{\omega}_{t,i,k-1} - \boldsymbol{\omega}_t\|^2 + 8k^2\eta_l^2 Lc_{p_B,i}(f(\boldsymbol{\omega}_t) - f(\boldsymbol{\omega}^*)) + 2k^2\eta_l^2 c_{p_G,i} + 2k^2\eta_l^2(\sum_{\tau=1}^{t-1} p_{\tau,i})^2 R^2 + k\eta_l^2\sigma^2 . \tag{64}$$

That is,

$$\mathbb{E}\|\boldsymbol{\omega}_{t,i,k} - \boldsymbol{\omega}_t\|^2$$

$$\leq \sum_{\tau=0}^{k-1} (8K^2\eta_l^2 Lc_{p_B,i}(f(\boldsymbol{\omega}_t) - f(\boldsymbol{\omega}^*)) + 2K^2\eta_l^2 c_{p_G,i} + 2K^2\eta_l^2(\sum_{\tau=1}^{t-1} p_{\tau,i})^2 R^2 + K\eta_l^2\sigma^2)(4K^2\eta_l^2 L^2 c_{p_B,i})^\tau \tag{65}$$

$$\leq \frac{8K^2\eta_l^2 Lc_{p_B,i}(f(\boldsymbol{\omega}_t) - f(\boldsymbol{\omega}^*)) + 2K^2\eta_l^2 c_{p_G,i} + 2K^2\eta_l^2(\sum_{\tau=1}^{t-1} p_{\tau,i})^2 R^2 + K\eta_l^2\sigma^2}{1 - 4K^2\eta_l^2 L^2 c_{p_B,i}} . \tag{66}$$

Combine the equation (43), (49), (56), and (66), we get

$$\mathbb{E}\|\boldsymbol{\omega}_t + \Delta\boldsymbol{\omega}_t - \boldsymbol{\omega}^*\|^2$$

$$\leq (1 - \frac{\mu\eta}{2})\mathbb{E}\|\boldsymbol{\omega}_t - \boldsymbol{\omega}^*\|^2 + (-2\eta + 16\eta^2 Lc_{p_B} + \frac{1}{N}\sum_{i=1}^{N} \frac{16K^2\eta\eta_l^2 Lc_{p_B,i}(1 + 4\eta Lc_{p_B,i})}{1 - 4K^2\eta_l^2 L^2 c_{p_B,i}})(f(\boldsymbol{\omega}_t) - f^*) + 2\eta^2 c_{p_G}$$

$$+ \frac{\eta^2\sigma^2}{NK} + \frac{1}{N}\sum_{i=1}^{N}\left(2\eta^2(\sum_{\tau=1}^{t-1} p_{\tau,i})^2 + \frac{4K^2\eta\eta_l^2 L(\sum_{\tau=1}^{t-1} p_{\tau,i})^2(1 + 4\eta Lc_{p_B,i})}{1 - 4K^2\eta_l^2 L^2 c_{p_B,i}}\right) R^2$$

$$+ \frac{1}{N}\sum_{i=1}^{N} \frac{2K\eta\eta_l^2 L(2Kc_{p_G,i} + \sigma^2)(1 + 4\eta Lc_{p_B,i})}{1 - 4K^2\eta_l^2 L^2 c_{p_B,i}} + \frac{2\eta}{N}\sum_{i=1}^{N}\sum_{\tau=1}^{t-1} p_{\tau,i}R\|\boldsymbol{\omega}_t - \boldsymbol{\omega}^*\| . \tag{67}$$

Then we notice that

$$-2\eta + 16\eta^2 Lc_{p_B} + \frac{1}{N}\sum_{i=1}^{N} \frac{16K^2\eta\eta_l^2 L^2 c_{p_B,i}(1 + 4\eta Lc_{p_B,i})}{1 - 4K^2\eta_l^2 L^2 c_{p_B,i}} \leq 0 . \tag{68}$$

Use the fact that $\eta = K\eta_l\eta_g$, and solve above inequality, we notice that if $\eta$ satisfy

$$\eta \leq \frac{\sqrt{3\eta_g^2 + 4c_{p_B,i}\eta_g^4} - \sqrt{4c_{p_B,i}\eta_g^4}}{6L\sqrt{c_{p_B,i}}} , \tag{69}$$

for any $c_{p_B,i}$, convergence can be promised. Define $\hat{c}_{p_B}$ as the largest $c_{p_B,i}$, let $\eta \leq \frac{\sqrt{3\eta_g^2 + 4\hat{c}_{p_B}\eta_g^4} - \sqrt{4\hat{c}_{p_B}\eta_g^4}}{12L\sqrt{\hat{c}_{p_B}}}$, we have $1 - 4K^2\eta_l^2 L^2 c_{p_B,i} \geq \frac{33 - 8\eta_g^2\hat{c}_{p_B} + 4\sqrt{3\eta_g^2\hat{c}_{p_B} + 4\eta_g^4\hat{c}_{p_B}^2}}{36}$, and then $\frac{1 + 4\eta L c_{p_B,i}}{1 - 4K^2\eta_l^2 L^2 c_{p_B,i}} \leq \frac{1}{12} - \frac{7}{33 + \sqrt{3\eta_g^2\hat{c}_{p_B} + 4\eta_g^4\hat{c}_{p_B}^2} - 2\eta_g^2\hat{c}_{p_B}} \leq \frac{1}{12}$.

Define $c_{R,i} = (\sum_{\tau=1}^{t-1} p_{\tau,i})^2 R^2$, and $c_R = \frac{1}{N}\sum_{i=1}^N c_{R,i}$. Assume $c_1 = 2c_{p_G} + \frac{\sigma^2}{NK} + 2c_R$, $c_2 = \frac{Lc_{p_G}}{3\eta_g^2} + \frac{L\sigma^2}{6\eta_g^2 K} + \frac{Lc_R}{3\eta_g^2}$, and $c_3 = \frac{2}{N}\sum_{i=1}^N\sum_{\tau=1}^{t-1} p_{\tau,i} R$, inequality (67) become

$$\mathbb{E}\|\boldsymbol{\omega}_t + \Delta\boldsymbol{\omega}_t - \boldsymbol{\omega}^*\|^2 \leq (1 - \frac{\mu\eta}{2})\mathbb{E}\|\boldsymbol{\omega}_t - \boldsymbol{\omega}^*\|^2 - \eta(f(\boldsymbol{\omega}_t) - f^*) + c_1\eta^2 + c_2\eta^3 + c_3\eta\|\boldsymbol{\omega}_t - \boldsymbol{\omega}^*\|. \tag{70}$$

Move $f(\boldsymbol{\omega}_t) - f^*$ to the left side, we get

$$f(\boldsymbol{\omega}_t) - f(\boldsymbol{\omega}^*) \leq \frac{1}{\eta}(1 - \frac{\mu\eta}{2})\mathbb{E}\|\boldsymbol{\omega}_t - \boldsymbol{\omega}^*\|^2 - \frac{1}{\eta}\mathbb{E}\|\boldsymbol{\omega}_{t+1} - \boldsymbol{\omega}^*\|^2 + c_1\eta + c_2\eta^2 + \varphi_t. \tag{71}$$

Here $\varphi_t$ is a constant bounded by $c_3\|\boldsymbol{\omega}_0 - \boldsymbol{\omega}^*\|$. Because of $\varphi_t$, the model cannot converge to an arbitrary $\epsilon$; instead, it can only converge to the neighborhood of $\epsilon + \varphi$, where $\varphi$ is the weighted sum of the sequence $\varphi_t$. The following parts give the convergence rates when models converge to $\epsilon + \varphi$ for convenience.

### B.4.3 WEIGHTS OF ROUNDS.

We can carefully tune the $p_{\tau,i}$ to minimize the noise while speeding up the convergence. Here we give the lemma of the best choice of $p_{\tau,i}$.

**Lemma B.4.** *On round $t$, when setting $p_{\tau,i} = \frac{D^2}{tD^2 + (t-1)R^2}$ for any $\tau < t$, and $p_{t,i} = \frac{(t-1)R^2 + D^2}{tD^2 + (t-1)R^2}$, we can minimize $c_1$ and $c_2$ to get best convergence.*

*Proof.* Minimizing $c_1$ and $c_2$ is equal to minimize $c_{p_G} + c_R$, that is,

$$\min_p \frac{1}{N}\sum_{i=1}^N (1 - p_{t,i})^2 R^2 + G^2 + D^2(\sum_{\tau=1}^t p_{\tau,i}^2),$$

$$s.t. \sum_{\tau=1}^t p_{\tau,i} = 1, \forall i = 1, \dots, N. \tag{72}$$

Solving above optimization problem, we get the results. $\square$

Using lemma B.4, we have $\min\{c_{p_G} + c_R\} = G^2 + D^2\left(\frac{D^2 + (t-1)R^2}{tD^2 + (t-1)R^2}\right)$. Notice that the value of $c_1$, $c_2$, and $c_3$ are changing over $t$. We use $c_1^t$, $c_2^t$, and $c_3^t$ to represent $c_1$, $c_2$, and $c_3$ on round $t$.

### B.4.4 CONVERGENCE RESULTS

**Strongly convex functions.** By equation (80), when $f_{t,i}(\boldsymbol{\omega})$ are strongly convex functions, using Lemma A.1, and define $q_t = (1 - \frac{\mu\eta}{2})^{1-t}$, we have

$$f(\boldsymbol{\omega}_F) - f(\boldsymbol{\omega}^*) \leq \mathcal{O}\left(\mu\|\boldsymbol{\omega}_0 - \boldsymbol{\omega}^*\|^2 \exp(-\frac{\mu\eta T}{2})\right) + \frac{1}{\sum_{t=1}^T q_t} q_t\left(c_1\eta + c_2\eta^2 + \varphi_t\right). \tag{73}$$

By Lemma A.6, we have,

$$\frac{1}{\sum_{t=1}^T q_t}\sum_{t=1}^T q_t(c_R + c_{p_G}) = \frac{1}{\sum_{t=1}^T q_t}\sum_{t=1}^T q_t\left(G^2 + D^2\left(\frac{D^2 + (t-1)R^2}{tD^2 + (t-1)R^2}\right)\right)$$

$$= \frac{1}{\sum_{t=1}^T q_t}\sum_{t=1}^T q_t\left(G^2 + D^2\left(\frac{R^2}{R^2 + D^2} + \frac{D^4}{(R^2 + D^2)^2}\frac{1}{t - \frac{R^2}{R^2 + D^2}}\right)\right)$$

$$= \mathcal{O}\left(G^2 + D^2\left(\frac{R^2}{R^2 + D^2} + \frac{D^4}{(R^2 + D^2)^2}\frac{\ln T}{T}\right)\right). \tag{74}$$

For $\varphi_t$, by Lemma B.4, we have,

$$\varphi = \frac{1}{\sum_{t=1}^{T} q_t} \sum_{t=1}^{T} q_t \varphi_t \leq \mathcal{O}\left( R \left( \frac{D^2}{D^2 + R^2} - \frac{D^4}{(D^2 + R^2)^2} \frac{\ln T}{T} \right) \|\boldsymbol{\omega}_0 - \boldsymbol{\omega}^*\| \right). \tag{75}$$

Then define $c_1^o = \frac{1}{\sum_{t=1}^{T} q_t} \sum_{t=1}^{T} q_t c_1^t, c_2^o = \frac{1}{\sum_{t=1}^{T} q_t} \sum_{t=1}^{T} q_t c_2^t$, we have,

$$f(\boldsymbol{\omega}_F) - f(\boldsymbol{\omega}^*) - \varphi \leq \mathcal{O}\left( \mu \|\boldsymbol{\omega}_0 - \boldsymbol{\omega}^*\|^2 \exp(-\frac{\mu \eta T}{2}) + c_1^o \eta + c_2^o \eta^2 \right) \tag{76}$$

$$= \mathcal{O}\left( \mu \|\boldsymbol{\omega}_0 - \boldsymbol{\omega}^*\|^2 \exp(-\frac{\mu T}{L c_O}) + \frac{\sigma^2}{\mu N K T} + \frac{c_A}{\mu T} + \frac{c_B}{\mu T^2} + \frac{c_C}{\mu^2 T^2} + \frac{c_D}{\mu^2 T^3} \right), \tag{77}$$

where $c_O = 1 + A^2 + B^2, c_A = G^2 + \frac{D^2 R^2}{R^2 + D^2}, c_B = \frac{D^6}{(R^2 + D^2)^2}, c_C = \frac{L\sigma^2}{\eta_g^2 K} + \frac{Lc_A}{\eta_g^2}, c_D = \frac{Lc_B}{\eta_g^2}$.

**General convex functions.** When $f_{t,i}(\boldsymbol{\omega})$ are general convex functions ($\mu = 0$), directly use Lemma A.2 we get,

$$f(\boldsymbol{\omega}_F) - f(\boldsymbol{\omega}^*) - \varphi \leq \mathcal{O}\left( \frac{c_O F}{T} + \sqrt{\frac{\sigma^2 F}{N K T}} + \sqrt{\frac{F c_A}{T}} + \frac{\sqrt{c_B F}}{T} + \sqrt[3]{\frac{c_C F^2}{T^2}} + \frac{\sqrt[3]{c_D F^2}}{T} \right), \tag{78}$$

where $c_O = 1 + A^2 + B^2, c_A = G^2 + \frac{D^2 R^2}{R^2 + D^2}, c_B = \frac{D^6}{(R^2 + D^2)^2}, c_C = \frac{L\sigma^2}{\eta_g^2 K} + \frac{Lc_A}{\eta_g^2}, c_D = \frac{Lc_B}{\eta_g^2}$, and $F = \|\boldsymbol{\omega}_0 - \boldsymbol{\omega}^*\|^2$.

**The drift of optimal point.** Because of information loss, we can't get the true optimal point when considering previous rounds' information. Instead, the drift can be described as

$$\varphi = c_3^o \|\boldsymbol{\omega}_0 - \boldsymbol{\omega}^*\| = \mathcal{O}\left( R \left( \frac{D^2}{D^2 + R^2} - \frac{D^4}{(D^2 + R^2)^2} \frac{\ln T}{T} \right) \|\boldsymbol{\omega}_0 - \boldsymbol{\omega}^*\| \right). \tag{79}$$

**Convergence of FedAvg.** Notice that FedAvg is a special case of CFL by setting $p_{t,i} = 1$ on round $t$. Having this, we derive the one round convergence of FedAvg under our formulation as,

$$f(\boldsymbol{\omega}_t) - f(\boldsymbol{\omega}^*) \leq \frac{1}{\eta}(1 - \frac{\mu \eta}{2}) \mathbb{E}\|\boldsymbol{\omega}_t - \boldsymbol{\omega}^*\|^2 - \frac{1}{\eta} \mathbb{E}\|\boldsymbol{\omega}_{t+1} - \boldsymbol{\omega}^*\|^2 + c_1 \eta + c_2 \eta^2, \tag{80}$$

where $c_1 = 2(G^2 + D^2) + \frac{\sigma^2}{NK}, c_2 = \frac{L(G^2 + D^2)}{3\eta_g^2} + \frac{L\sigma^2}{6\eta_g^2 K}$. Then using Lemma A.1, we get the convergence rate when $f_{t,i}(\boldsymbol{\omega})$ are $\mu$ strongly convex.

$$f(\boldsymbol{\omega}_t) - f(\boldsymbol{\omega}^*) \leq \mathcal{O}\left( \mu \|\boldsymbol{\omega}_0 - \boldsymbol{\omega}^*\|^2 \exp(\frac{\mu T}{L c_O}) + \frac{\hat{c}_A}{\mu T} + \frac{\hat{c}_C}{\mu^2 T^2} \right), \tag{81}$$

where $c_O = 1 + A^2 + B^2, c_A = G^2 + D^2, c_C = \frac{L\sigma^2}{\eta_g^2 K} + \frac{Lc_A}{\eta_g^2}$. Besides, when $f_{t,i}(\boldsymbol{\omega})$ are general convex, we have,

$$f(\boldsymbol{\omega}_F) - f(\boldsymbol{\omega}^*) \leq \mathcal{O}\left( \frac{c_O F}{T} + \sqrt{\frac{\sigma^2 F}{N K T}} + \sqrt{\frac{F c_A}{T}} + \sqrt[3]{\frac{c_C F^2}{T^2}} \right), \tag{82}$$

where $c_O = 1 + A^2 + B^2, c_A = G^2 + D^2, c_C = \frac{L\sigma^2}{\eta_g^2 K} + \frac{Lc_A}{\eta_g^2}$, and $F = \|\boldsymbol{\omega}_0 - \boldsymbol{\omega}^*\|^2$.

## B.5 PROOF OF CONVERGENCE RATE FOR NON-CONVEX SETTING

$$\mathbb{E}\left[ f(\boldsymbol{\omega}_{t+1}) \right] \leq \mathbb{E}\left[ f(\boldsymbol{\omega}_t) \right] + \underbrace{\mathbb{E}\left[ \langle \nabla f(\boldsymbol{\omega}_t), \Delta \boldsymbol{\omega}_t \rangle \right]}_{A_1} + \frac{L}{2} \underbrace{\mathbb{E}\left[ \|\Delta \boldsymbol{\omega}_t\|^2 \right]}_{A_2} \tag{83}$$

For $A_1$ part, we have,

$$\mathbb{E}\left[\langle \nabla f(\boldsymbol{\omega}_t), \Delta \boldsymbol{\omega}_t \rangle\right]$$

$$= \frac{-\eta}{NK} \sum_{i=1}^{N} \sum_{k=1}^{K} \mathbb{E}\left[\langle \nabla f(\boldsymbol{\omega}_t), \nabla \bar{f}_{t,i}(\boldsymbol{\omega}_{t,i,k}) \rangle\right] \tag{84}$$

$$= \frac{-\eta}{NK} \sum_{i=1}^{N} \sum_{k=1}^{K} \mathbb{E}\left[\langle \nabla f(\boldsymbol{\omega}_t), \sum_{\tau=1}^{t} p_{\tau,i} \nabla f_{\tau,i}(\boldsymbol{\omega}_{t,i,k}) - \sum_{\tau=1}^{t-1} p_{\tau,i} \Delta_{\tau,i}(\boldsymbol{\omega}_{t,i,k}) \rangle\right] \tag{85}$$

$$\leq \frac{-\eta}{NK} \sum_{i=1}^{N} \sum_{k=1}^{K} \mathbb{E}\left[\langle \nabla f(\boldsymbol{\omega}_t), \sum_{\tau=1}^{t} p_{\tau,i} \nabla f_{\tau,i}(\boldsymbol{\omega}_{t,i,k}) \rangle\right] + \frac{\eta}{2N} \sum_{i=1}^{N}(1-p_{t,i})\left(\mathbb{E}\left[\|\nabla f(\boldsymbol{\omega}_t)\|^2\right] + R^2\right) \tag{86}$$

$$= \frac{\eta}{2NK} \sum_{i=1}^{N} \sum_{k=1}^{K} \left( \mathbb{E}\left[\left\|\nabla f(\boldsymbol{\omega}_t) - \sum_{\tau=1}^{t} p_{\tau,i} \nabla f_{\tau,i}(\boldsymbol{\omega}_{t,i,k})\right\|^2\right] - \mathbb{E}\left[\|\nabla f(\boldsymbol{\omega}_t)\|^2\right] - \mathbb{E}\left[\left\|\sum_{\tau=1}^{t} p_{\tau,i} \nabla f_{\tau,i}(\boldsymbol{\omega}_{t,i,k})\right\|^2\right] \right)$$
$$+ \frac{\eta}{2N} \sum_{i=1}^{N}(1-p_{t,i})\left(\mathbb{E}\left[\|\nabla f(\boldsymbol{\omega}_t)\|^2\right] + R^2\right) \tag{87}$$

From equation (53) we have

$$\mathbb{E}\left[\|\Delta \boldsymbol{\omega}_t\|^2\right] \leq \frac{2\eta^2}{NK} \sum_{i=1}^{N} \sum_{k=1}^{K} \left( \mathbb{E}\left\|\sum_{\tau=1}^{t} p_{\tau,i} \nabla f_{\tau,i}(\boldsymbol{\omega}_{t,i,k})\right\|^2 + \mathbb{E}\left\|\sum_{\tau=1}^{t-1} p_{\tau,i} \Delta_{\tau,i}(\boldsymbol{\omega}_{t,i,k})\right\|^2 \right) + \frac{\eta^2 \sigma^2}{NK} \tag{88}$$

Then when $\eta \leq \frac{1}{2L}$, the $\mathbb{E}\left\|\sum_{\tau=1}^{t} p_{\tau,i} \nabla f_{\tau,i}(\boldsymbol{\omega}_{t,i,k})\right\|^2$ term can be ignored, since the coefficient number will less than 0. Then the remaining term in $A_2$ is $\frac{2\eta^2}{N} \sum_{i=1}^{N}(1-p_{t,i})^2 R^2 + \frac{\eta^2 \sigma^2}{NK}$. Then we are willing to consider the $\left\|\nabla f(\boldsymbol{\omega}_t) - \sum_{\tau=1}^{t} p_{\tau,i} \nabla f_{\tau,i}(\boldsymbol{\omega}_{t,i,k})\right\|^2$. Then we have

$$\mathbb{E}\left[\left\|\nabla f(\boldsymbol{\omega}_t) - \sum_{\tau=1}^{t} p_{\tau,i} \nabla f_{\tau,i}(\boldsymbol{\omega}_{t,i,k})\right\|^2\right]$$

$$= \mathbb{E}\left[\left\|\nabla f(\boldsymbol{\omega}_t) - \nabla f(\boldsymbol{\omega}_{t,i,k}) + \nabla f(\boldsymbol{\omega}_{t,i,k}) - \sum_{\tau=1}^{t} p_{\tau,i} \nabla f_{\tau,i}(\boldsymbol{\omega}_{t,i,k})\right\|^2\right] \tag{89}$$

$$\leq 2\mathbb{E}\left[\|\nabla f(\boldsymbol{\omega}_t) - \nabla f(\boldsymbol{\omega}_{t,i,k})\|^2\right] + 2\mathbb{E}\left[\left\|\nabla f(\boldsymbol{\omega}_{t,i,k}) - \sum_{\tau=1}^{t} p_{\tau,i} \nabla f_{\tau,i}(\boldsymbol{\omega}_{t,i,k})\right\|^2\right] \tag{90}$$

$$\leq 2L^2 \mathbb{E}\left[\|\boldsymbol{\omega}_t - \boldsymbol{\omega}_{t,i,k}\|^2\right] + \left(B^2 + A^2 \sum_{\tau=1}^{t} p_{\tau,i}^2\right) \mathbb{E}\left[\|\nabla f(\boldsymbol{\omega}_{t,i,k})\|^2\right] + \left(G^2 + D^2 \sum_{\tau=1}^{t} p_{\tau,i}^2\right) \tag{91}$$

$$\leq 2L^2 \mathbb{E}\left[\|\boldsymbol{\omega}_t - \boldsymbol{\omega}_{t,i,k}\|^2\right] + 2L^2\left(B^2 + A^2 \sum_{\tau=1}^{t} p_{\tau,i}^2\right) \mathbb{E}\left[\|\boldsymbol{\omega}_{t,i,k} - \boldsymbol{\omega}_t\|^2\right]$$
$$+ 2\left(B^2 + A^2 \sum_{\tau=1}^{t} p_{\tau,i}^2\right) \mathbb{E}\left[\|\nabla f(\boldsymbol{\omega}_t)\|^2\right] + \left(G^2 + D^2 \sum_{\tau=1}^{t} p_{\tau,i}^2\right) \tag{92}$$

Combine equation (87) and (92) we have

$$\mathbb{E}\left[\langle \nabla f(\boldsymbol{\omega}_t), \Delta \boldsymbol{\omega}_t \rangle\right]$$

$$\leq \frac{\eta L^2}{NK} \sum_{i=1}^{N} \sum_{k=1}^{K} c_{p_B,i} \mathbb{E}\left[\|\boldsymbol{\omega}_{t,i,k} - \boldsymbol{\omega}_t\|^2\right] + \eta(c_{p_B} - 1 - \frac{\sum_{i=1}^{N} p_{t,i}}{2N})\mathbb{E}\left[\|\nabla f(\boldsymbol{\omega}_t)\|^2\right]$$

$$+ \frac{\eta}{2}\left(c_{p_G} + \frac{1}{N}\sum_{i=1}^{N}(1 - p_{t,i})R^2\right) \tag{93}$$

Then we have

$$\mathbb{E}\left[f(\boldsymbol{\omega}_{t+1})\right] \leq \mathbb{E}\left[f(\boldsymbol{\omega}_t)\right] + \frac{\eta L^2}{NK}\sum_{i=1}^{N}\sum_{k=1}^{K} c_{p_B,i}\mathbb{E}\left[\|\boldsymbol{\omega}_{t,i,k} - \boldsymbol{\omega}_t\|^2\right] + \eta(c_{p_B} - 1 - \frac{\sum_{i=1}^{N} p_{t,i}}{2N})\mathbb{E}\left[\|\nabla f(\boldsymbol{\omega}_t)\|^2\right]$$

$$+ \frac{\eta}{2}\left(c_{p_G} + \frac{1}{N}\sum_{i=1}^{N}(1 - p_{t,i})R^2\right) + \frac{L\eta^2}{N}\sum_{i=1}^{N}(1 - p_{t,i})^2 R^2 + \frac{L\eta^2\sigma^2}{2NK} \tag{94}$$

Because we know

$$\mathbb{E}\|\boldsymbol{\omega}_{t,i,k} - \boldsymbol{\omega}_t\|^2$$

$$\leq 4k^2\eta_l^2 L^2 c_{p_B,i}\mathbb{E}\|\boldsymbol{\omega}_{t,i,k-1} - \boldsymbol{\omega}_t\|^2 + 4k^2\eta_l^2 c_{p_B,i}(\mathbb{E}\left[\|\nabla f(\boldsymbol{\omega}_t)\|^2\right]) + 2k^2\eta_l^2 c_{p_G,i} + 2k^2\eta_l^2(\sum_{\tau=1}^{t-1} p_{\tau,i})^2 R^2 + k\eta_l^2\sigma^2. \tag{95}$$

That is,

$$\mathbb{E}\|\boldsymbol{\omega}_{t,i,k} - \boldsymbol{\omega}_t\|^2$$

$$\leq \sum_{r=0}^{k-1}\left(4k^2\eta_l^2 c_{p_B,i}(\mathbb{E}\left[\|\nabla f(\boldsymbol{\omega}_t)\|^2\right]) + 2k^2\eta_l^2 c_{p_G,i} + 2k^2\eta_l^2(\sum_{\tau=1}^{t-1} p_{\tau,i})^2 R^2 + k\eta_l^2\sigma^2\right)\left(4k^2\eta_l^2 L^2 c_{p_B,i}\right)^r \tag{96}$$

$$\leq \frac{\left(4k^2\eta_l^2 c_{p_B,i}(\mathbb{E}\left[\|\nabla f(\boldsymbol{\omega}_t)\|^2\right]) + 2k^2\eta_l^2 c_{p_G,i} + 2k^2\eta_l^2(\sum_{\tau=1}^{t-1} p_{\tau,i})^2 R^2 + k\eta_l^2\sigma^2\right)}{1 - 4k^2\eta_l^2 L^2 c_{p_B,i}} \tag{97}$$

Let $8K^2\eta_l^2 c_{p_B,i} \leq 1$, we have,

$$\frac{1}{NK}\sum_{i=1}^{N}\sum_{k=1}^{K} c_{p_B,i}\mathbb{E}\|\boldsymbol{\omega}_{t,i,k-1} - \boldsymbol{\omega}_t\|^2$$

$$\leq \frac{1}{N}\sum_{i=1}^{N} c_{p_B,i}\frac{\left(4K^2\eta_l^2 c_{p_B,i}(\mathbb{E}\left[\|\nabla f(\boldsymbol{\omega}_t)\|^2\right]) + 2K^2\eta_l^2 c_{p_G,i} + 2K^2\eta_l^2(\sum_{\tau=1}^{t-1} p_{\tau,i})^2 R^2 + K\eta_l^2\sigma^2\right)}{1 - 4K^2\eta_l^2 L^2 c_{p_B,i}} \tag{98}$$

$$\leq \frac{1}{N}\sum_{i=1}^{N} c_{p_B,i}\mathbb{E}\left[\|\nabla f(\boldsymbol{\omega}_t)\|^2\right] + \frac{1}{2}c_{p_G,i} + \frac{1}{2}(\sum_{\tau=1}^{t-1} p_{\tau,i})^2 R^2 + \frac{\sigma^2}{4K} \tag{99}$$

$$= c_{p_B}\mathbb{E}\left[\|\nabla f(\boldsymbol{\omega}_t)\|^2\right] + \frac{1}{2}c_{p_G} + \frac{1}{2N}\sum_{i=1}^{N}(\sum_{\tau=1}^{t-1} p_{\tau,i})^2 R^2 + \frac{\sigma^2}{4K} \tag{100}$$

Then we have

$$\mathbb{E}\left[f(\boldsymbol{\omega}_{t+1})\right] \leq \mathbb{E}\left[f(\boldsymbol{\omega}_t)\right] + \eta\left((L^2+1)c_{p_B} - 1 - \frac{\sum_{i=1}^N p_{t,i}}{2N}\right)\mathbb{E}\left[\|\nabla f(\boldsymbol{\omega}_t)\|^2\right]$$

$$+ \frac{\eta}{2}\left(c_{p_G} + \frac{1}{N}\sum_{i=1}^N (1-p_{t,i})R^2\right) + \frac{L\eta^2}{N}\sum_{i=1}^N (1-p_{t,i})^2 R^2 + \frac{L\eta^2\sigma^2}{2NK}$$

$$+ \eta L^2\left(\frac{1}{2}c_{p_G} + \frac{1}{2N}\sum_{i=1}^N (\sum_{\tau=1}^{t-1} p_{\tau,i})^2 R^2 + \frac{\sigma^2}{4K}\right) \tag{101}$$

Then we must to choose the value of $p_{t,i}$ for better convergence. Follow the idea that we want to train a better model when convergent, we choose $p_{t,i}$ that can minimize the constant term, which is by solving

$$\min_p \ G^2 + D^2\sum_{\tau=1}^t p_{\tau,i}^2 + (\sum_{\tau=1}^{t-1} p_{\tau,1})^2 R^2$$

$$s.t. \ \sum_{\tau=1}^t p_{\tau,1} = 1 \tag{102}$$

We get same results as when the objective function is convex, that is, $p_{\tau,i} = \frac{D^2}{tD^2+(t-1)R^2}$ for any $\tau < t$, and $p_{t,i} = \frac{(t-1)R^2+D^2}{tD^2+(t-1)R^2}$, and $\min\{G^2 + D^2\sum_{\tau=1}^t p_{\tau,i}^2 + (\sum_{\tau=1}^{t-1} p_{\tau,1})^2 R^2\} = G^2 + D^2\left(\frac{D^2+(t-1)R^2}{tD^2+(t-1)R^2}\right)$. Then we simplify the equation by using $c_0(t)$, $c_1(t)$, and $c_2(t)$ to denote the constant terms, we have

$$\mathbb{E}\left[f(\boldsymbol{\omega}_{t+1})\right] \leq \mathbb{E}\left[f(\boldsymbol{\omega}_t)\right] - \eta c_0(t)\mathbb{E}\left[\|\nabla f(\boldsymbol{\omega}_t)\|^2\right] + \eta c_1(t) + \eta^2 c_2(t) \tag{103}$$

Then we have

$$\frac{1}{T}\sum_{t=1}^T c_0(t)\mathbb{E}\left[\|\nabla f(\boldsymbol{\omega}_t)\|^2\right] \leq \frac{\mathbb{E}\left[f(\boldsymbol{\omega}_0)\right] - \mathbb{E}\left[f(\boldsymbol{\omega}^*)\right]}{\eta} + \eta\frac{1}{T}\sum_{t=1}^T c_2(t) + \varphi \tag{104}$$

Consider $c_0$, because the value of $B^2$ and $A$ can't be constrained, the algorithm can't converge for very large $A$ and $B$. Here we first consider when $c_0 < 0$, and use $c_m$ to denote the $\min c_0$. Then let $\eta = \frac{\sqrt{KN}}{\sqrt{T}L}$, we derive the convergence rate as

$$\frac{1}{T}\sum_{t=1}^T \mathbb{E}\left[\|\nabla f(\boldsymbol{\omega}_t)\|^2\right] = O(\frac{L(f_0 - f_*)}{\sqrt{TNK}c_m} + \frac{1}{\sqrt{T}}\left(\frac{\sqrt{KN}}{L}\left(\frac{R^2}{R^2+D^2}\right)^2 + \frac{\sigma^2}{\sqrt{NK}}\right)$$

$$+ \frac{\sqrt{KN}}{TL}\frac{D^4}{c_m(R^2+D^2)^2} + \frac{\sqrt{KN}}{T\sqrt{T}L}\left(\frac{D^4}{(R^2+D^2)^2}\right)^2) \tag{105}$$

Then we want to analyse when the algorithm won't converge. When the algorithm won't converge, that means $c_0 \geq 0$. Because $c_0 = (L^2+1)c_{p_B} - 1 - \frac{\sum_{i=1}^N p_{t,i}}{2N} = \frac{1}{N}\sum_{i=1}^N (L^2+1)(1+A^2+B^2\sum_{\tau=1}^t p_{\tau,i}^2) - 1 - \frac{p_{t,i}}{2}$. Then the value of $A^2, B^2, L, p$ decide if $c_0 \geq 0$.

Notice that in FedAvg, $p_{t,i} = 1$, and $c_{0,avg} = (L^2+1)(1+A^2+B^2) - \frac{3}{2}$, and in CFL, we can setting different $p$ to get better convergence. For example, when $B$ is large, setting $p_{\tau,i} = \frac{1}{t}$, we have $c_0 = (L^2+1)(1+A^2+\frac{B^2}{t}) - 1 - \frac{1}{2t}$. Thus, we draw the conclusion as

Then we have following observations:

- *CFL converge faster than FedAvg by reducing the variance terms.* For $c_0 = \frac{1}{N}\sum_{i=1}^N (L^2+1)(1+A^2+B^2\sum_{\tau=1}^t p_{\tau,i}^2) - 1 - \frac{p_{t,i}}{2}$, in FedAvg, we have $p_{t,i} = 1$, then $c_{0,avg} = (L^2+1)(1+A^2+B^2) - \frac{3}{2}$. However, in CFL, we can adjust $p$, such as when setting $p_{\tau,i} = \frac{1}{t}$, we have $c_0 = (L^2+1)(1+A^2+\frac{B^2}{t}) - 1 - \frac{1}{2t}$, then the variance term reduce from $B^2$ to $\frac{B^2}{t}$.

- *The convergence rate of CFL become better for larger $t$, since $c_{p_B}$ in $c_0$ become smaller for larger $t$.*
- Improve $N, K$ will speed up the convergence by reducing the terms about $f_0 - f_*$ and $\sigma^2$. However, larger $N$ and $K$ will cause larger information loss and round drift (terms with $R$ and $D$), thus, it should be a trade off in practice.

# C  EXPERIMENT DETAILS

## C.1  NOISY QUADRATIC MODEL

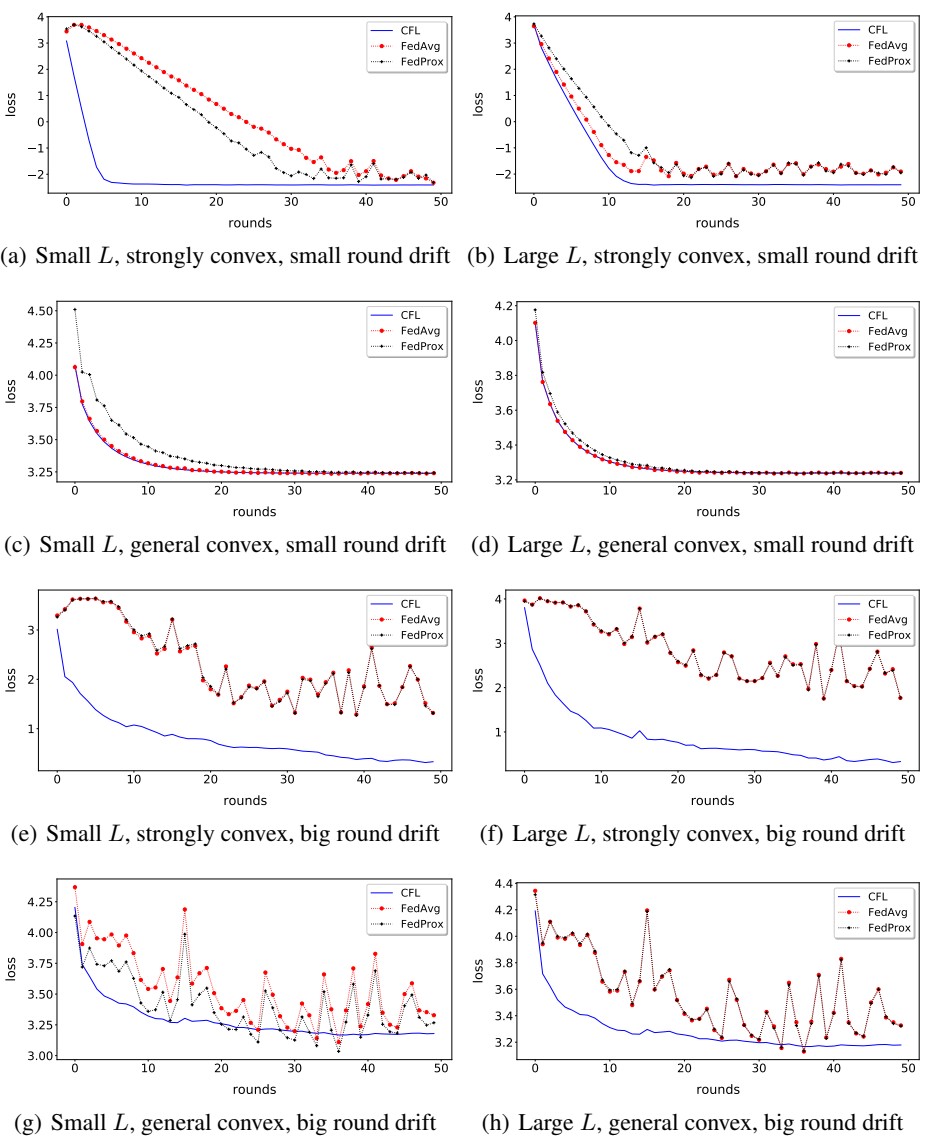

(a) Small $L$, strongly convex, small round drift    (b) Large $L$, strongly convex, small round drift

(c) Small $L$, general convex, small round drift    (d) Large $L$, general convex, small round drift

(e) Small $L$, strongly convex, big round drift    (f) Large $L$, strongly convex, big round drift

(g) Small $L$, general convex, big round drift    (h) Large $L$, general convex, big round drift

Figure 2: Performance of different algorithms on noisy quadratic model. The curves all evaluate the loss on global test datasets.

We use the noisy quadratic model introduced in Zhang et al. (2019) to simulate the assumptions we used in the proof. We use the model $f(\boldsymbol{\omega}) = \boldsymbol{\omega}^T \mathbf{A} \boldsymbol{\omega} + \mathbf{B}^T \boldsymbol{\omega} + \mathbf{C}$, where $\boldsymbol{\omega} \in \mathbb{R}^n$ is the parameter we want to optimize, and $\mathbf{A} \in \mathbb{R}^{n \times n}$, $\mathbf{A} \succeq 0$, $\mathbf{B} \in \mathbb{R}^n$, and $\mathbf{C} \in \mathbb{R}$. We construct $\mathbf{A}$ by firstly constructing a diagonal matrix $\Lambda$ and then constructing $\mathbf{A}$ by let $\mathbf{A} = \mathbf{U}^T \Lambda \mathbf{U}$, where $\mathbf{U}$ is a unitary matrix. Here we control the eigenvalues of $\mathbf{A}$ to control the convexity of model, that is,

| | SRD | | | | BRD | | | |
| Method | SL-SC | LL-SC | SL-GC | LL-GC | SL-SC | LL-SC | SL-GC | LL-GC |
|---|---|---|---|---|---|---|---|---|
| FAVG | 0.03 | 0.08 | 0.33 | 0.09 | 0.02 | 0.01 | 0.09 | 0.02 |
| FPROX | 0.04 | 0.07 | 0.35 | 0.09 | 0.02 | 0.01 | 0.26 | 0.02 |
| CFL | 0.2 | 0.09 | 0.35 | 0.09 | 0.25 | 0.08 | 0.3 | 0.08 |

Table 7: Best learning rate of different algorithms on noisy quadratic model

$\mu \leq \lambda_{min}(\mathbf{A}) \leq \lambda_{max}(\mathbf{A}) \leq L$. Because $\Lambda$ and $\mathbf{A}$ have same eigenvalues, it's simple to control the eigenvalues of $\mathbf{A}$ by control the eigenvalues of $\Lambda$.

To simulate Assumption 2, 3, we formulate the gradient noise model by

$$g_{t,i}(\boldsymbol{\omega}) = \mathbf{A}\boldsymbol{\omega} + \mathbf{B} + \boldsymbol{\delta}_i + \boldsymbol{\xi}_{t,i} + \boldsymbol{\nu}_{t,i,k}\,, \tag{106}$$

where $\boldsymbol{\delta}_i$ denotes the client drift of client $j$, $\boldsymbol{\xi}_{t,i}$ denotes the round drift of client $j$ on round $i$, and $\boldsymbol{\nu}_{t,i,k}$ denotes the noise of gradient of client $j$ on round $i$, iteration $k$. All of above drifts and noise are generated from Normal distribution with zero mean and different variance.

In practice, to show if the numerical results match our theoretical results, we tried eight settings: large and small $L$, large and small round drift, and general and strongly convex conditions. For different types of $L$, we set $L = 20$ for large $L$, and $L = 5$ for small $L$. For convexity, we set $\mu = 1$ for strongly convex, and $\mu = 0$ for general convex. For different round drift, we set $\mathbb{E}\|\boldsymbol{\xi}_{t,i}\|^2 = 100$ for big round drift, and $\mathbb{E}\|\boldsymbol{\xi}_{t,i}\|^2 = 0.01$ for small round drift. Besides, we set fixed variance of $\boldsymbol{\delta}_i$ and $\boldsymbol{\nu}_{t,i,k}$ for $\mathbb{E}\|\boldsymbol{\delta}_i\|^2 = 0.01$, and $\mathbb{E}\|\boldsymbol{\nu}_{t,i,k}\|^2 = 0.00001$.

We do the gradient descent by let $\boldsymbol{\omega}_{t,i,k+1} = \boldsymbol{\omega}_{t,i,k} - \eta_l g_{t,i}(\boldsymbol{\omega}_{t,i,k})$, and we use $\|A\boldsymbol{\omega} + B\|$ as loss because for convex functions, when it reaches the global optimal point, we should have $\|A\boldsymbol{\omega} + B\| = 0$, and if it is far away from global optimal, $\|A\boldsymbol{\omega} + B\|$ will be large.

Besides, we show that CFL algorithms can tolerate a larger learning rate here. Table 7 shows the best learning rates for different FL algorithms on noisy quadratic model. In Table 7, we denote small round drift as SRD, big round drift as BRD, small $L$ as SL, large $L$ as LL, strongly convex as SC, and general convex as GC. We show that the best learning rate of CFL is the largest. The difference between CFL methods and other FL baselines becomes large when models are strongly convex or with big round drift.

Figure 2 shows the performance of different algorithms, and CFL outperforms other FL baselines significantly in all settings. Besides, the loss curves of CFL are smoother than FL methods, which implies that CFL can reduce the variance of gradients.

## C.2 REALISTIC DATASETS

### C.2.1 SETUP

We consider federated learning an image classifier on split-CIFAR10 and split-CIFAR100 datasets with ResNet18, and split-Fashion-MNIST dataset with a two-layer MLP. The "split" follows the idea introduced in Yurochkin et al. (2019); Hsu et al. (2019); Reddi et al. (2021), where we leverage the Latent Dirichlet Allocation (LDA) to control the distribution drift with parameter $\alpha$ (See Algorithm 4). Larger $\alpha$ denotes smaller drifts here.

In our experiments, unless specifically mentioned otherwise all datasets are partitioned to 210 subsets for 7 different clients: all clients are selected and trained for 500 communication rounds, and each client samples one of the corresponding 30 subsets randomly for the local training. Note that unless mentioned otherwise the training strategy here applies to all FL baselines and CFL methods, and we evaluate the performance of models on global test datasets. We carefully tuned the hyper-parameters in all algorithms, and we report the results under the optimal settings after many trials. For CFL-Regularization, we set the weight of regularization term for $\beta = 1$ on fc layer, and $\beta = 0.1$ for last block (layer). For FedProx, we set the weight of proximal term $\mu = 0.1$, and for MimeLite, we set the global momentum weight for $\gamma = 0.01$. Besides, for all experiments, we set fixed learning rate $lr = 0.01$. We also naively examined warm-up tuning strategies but can not observe significant improvement on final results.

**Construct local datasets with overlap.** Follow the partition methods of disjoint local datasets; we first split the whole dataset to

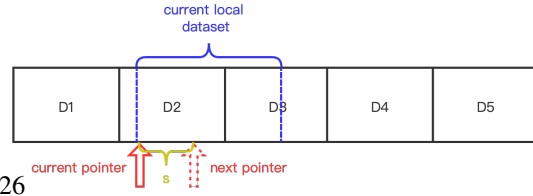

Figure 3: Process of constructing local datasets with overlap. Blue lines bound the current local datasets. Red

$M \times S$ subsets for $S$ different clients: each client has $M$ disjoint local subsets, and we put these $M$ subsets in sequence. For each client, we use a pointer to point to the start position of the current local dataset. Figure 3 show the case when $M = 5$: $D_1 - D_5$ are 5 disjoint local subsets of client $i$. The pointer point to the start point of a local dataset of the current round and the blue lines bound current subsets. At the beginning of training, pointers belong to each client will point to the beginning of the sequence. At the end of each round, the pointer moves $s$ steps, and if it moves to the end of all the sequence, it will come back to the start point.

In practice, the Cifar10 dataset is partitioned to 210 subsets for 7 clients, and set the size of local dataset $S_l = 285$. Then when we set $s = 285$, the local datasets are disjoint, which means the overlap is 0%. Otherwise, when set $s = 213$, the overlap is 25%, and when set $s = 142$, the overlap is 50%.

### C.2.2 APPROXIMATION METHODS

**Regularization Methods.** We use Taylor Extension to approximate the objective functions of previous rounds, that is,

$$\tilde{f}(\boldsymbol{\omega}) = f(\hat{\boldsymbol{\omega}}) + \nabla f(\hat{\boldsymbol{\omega}})^T (\boldsymbol{\omega} - \hat{\boldsymbol{\omega}}) + \frac{1}{2}(\boldsymbol{\omega} - \hat{\boldsymbol{\omega}})^T \nabla^2 f(\hat{\boldsymbol{\omega}})(\boldsymbol{\omega} - \hat{\boldsymbol{\omega}}), \tag{107}$$

where $\hat{\boldsymbol{\omega}}$ are the parameters of previous rounds. Then the gradient of $\tilde{f}(\boldsymbol{\omega})$ is,

$$\nabla \tilde{f}(\boldsymbol{\omega}) = \nabla f(\hat{\boldsymbol{\omega}}) + \nabla^2 f(\hat{\boldsymbol{\omega}})(\boldsymbol{\omega} - \hat{\boldsymbol{\omega}}). \tag{108}$$

Having this, we can approximate the gradients of objective functions of previous rounds. The problem is how to approximate $\nabla^2 f(\hat{\boldsymbol{\omega}})$. Here we use two kinds of methods to calculate the Hessian matrices. The first is to use the Fisher Information Matrix as introduced in EWC (Kirkpatrick et al., 2017). The Fisher Information Matrix is equal to the diagonal Hessian matrix when using cross entropy loss functions. The Fisher Information Matrix can be calculated by,

$$F_{ij} = [\nabla f(\hat{\boldsymbol{\omega}})]_{ij}^2, \tag{109}$$

where $F_{ij}$ are the entries of Fisher Information Matrix. Another method is to calculate the diagonal Hessian matrix. We use the PyHessian package (Yao et al., 2020) in this experiment.

In addition, if the parameters or the Hessian matrix changes too much, the performance of this method is constrained. To formulate this problem, assume $\left\| \nabla^2 f(\boldsymbol{\omega}) \right\| \leq \epsilon$ for some arbitrary $\epsilon$, we have

$$\|\Delta_{t,i}(\boldsymbol{\omega})\| \leq \epsilon \|\boldsymbol{\omega} - \hat{\boldsymbol{\omega}}\|^2 .$$

The corresponding proof refers to B.3.

In each round, the server will collect Hessian matrices from clients, and store them in a buffer. Then at the beginning of next round, server combine send latest 40 Hessian matrices, gradients, and parameters, and send them to chosen clients. Each clients use these to calculate regularization terms. In practice, we only add regularization terms to the top layers (last block and fc layer of ResNet18). This is based on the assumption that top layers contain more personal information while bottom layers contain more general information. We verified that this strategy perform better than adding regularization terms to all layers in experiments.

**Generation Methods.** We use MCMC to generate samples in each round. In practice, we initialize $\mathbf{x}$ from uniform distribution, and update $\mathbf{x}$ by,

$$\mathbf{x}^k = \mathbf{x}^{k-1} - \eta \nabla_{\mathbf{x}} E(\mathbf{x}^{k-1}) + \boldsymbol{\omega}, \tag{110}$$

where $E(\mathbf{x}^k - 1)$ is the function that can measure the distance between $\mathbf{x}^{k-1}$ and the real local distribution. $\boldsymbol{\omega} \sim \mathcal{N}(0, \sigma)$.

In practice, we generate 50-100 samples of each local dataset and add them to the current local datasets for training. However, because of the low quality of the generated data, the improvement is limited. Besides, the generated data will pollute the batch normalization(BN) layer, and we should use real data to refresh BN layer at the end of each round.

---

**Algorithm 3** iCaRL Construct Core Set

---

**Require:** Image set $\mathbf{X} = \{\mathbf{x}_1, \mathbf{x}_2, \mathbf{x}_3, ..., \mathbf{x}_n\}$ of class $y$, $m$: target number of samples, $\phi : \mathcal{X} \to \mathcal{R}^d$: current feature function

1: $\mu \leftarrow \frac{1}{n} \sum_{\mathbf{x} \in \mathbf{X}} \phi(\mathbf{x})$
2: **for** $k = 1, ..., m$ **do**
3: $\quad \Big| \quad p_k \leftarrow \arg\min_{\mathbf{x} \in \mathbf{X}} \left\| \mu - \frac{1}{k} \left( \phi(\mathbf{x}) + \sum_{j=1}^{k-1} \phi(p_i) \right) \right\|$
4: $P \leftarrow (p_1, p_2, ..., p_m)$

---

**Core Set Methods.** Another simple yet effective treatment in CL lines in the category of Exemplar Replay (Rebuffi et al., 2017; Castro et al., 2018). This approach stores past core-set samples (a.k.a. exemplars) selectively and periodically, and replays them together with the current local datasets. We tried two sample methods. First is so-called Naive method, in which the samples are uniformly chosen from local datasets. Except of naive select core sets, we also tried another core set sampling method introduced in iCaRL (Rebuffi et al., 2017). See Algorithm 3 for details.

In practice, we save 100 figures of each local dataset and combine them with the current local datasets. Saving core sets perform best compared with these three approximation methods; however, only valid when the number of clients is limited.

### C.2.3 ADDITIONAL EXPERIMENTS

**Applicability of different algorithms.** We list the applicability of different algorithms under various time-varying scenarios in Table 8.

Table 8: **The applicability of various algorithms under different time-varying scenarios.**

| Scenarios | FL baselines | | | | CFL methods | |
|---|---|---|---|---|---|---|
| | FedAvg | FedProx | SCAFFOLD | MimeLite | CFL-Regularization | CFL-Core-Set |
| Stateful clients | ✓ | ✓ | ✗ | ✓ | ✓ | ✓ |
| Stateless clients | ✓ | ✓ | ✗ | ✓ | ✓ | ✗ |

**Convergence curves for different settings.** Figure 4 show convergence curves of CFL and FedAvg on different datasets. The settings are the same as results in Table 3. Models are trained on partitioned datasets with $\alpha = 0.1$, and all datasets are partitioned to 210 subsets for 7 clients. To show the difference between different algorithms more clearly, all curves are smoothed by a 1D-Mean-Filter. Results show CFL can converge to a better optimum compare with FedAvg.

**Investigating resistance of CFL to time-evolving scenarios.** Figure 6 shows the loss curve of CFL-Regularization and FedAvg on different datasets. Models are trained on partitioned datasets with $\alpha = 0.1$, and all datasets are partitioned to 210 subsets for 7 clients. The first column shows the loss curve of the first 300 rounds, and the second column shows the loss of some chosen details. Because of the non-iidness of local datasets, the loss will suddenly rise. Notice that *CFL-Regularization has an apparent mitigation effect on this situation*.

**Difference between training and test loss.** Figure 5 show the loss on global test data and past appeared training data. We evaluate the model with stateful clients, set $\alpha = 0.2$, and use FedAvg algorithm. We show that *there is no significant difference between loss value on global test data and past appeared training data*.

### C.3 ALGORITHMS

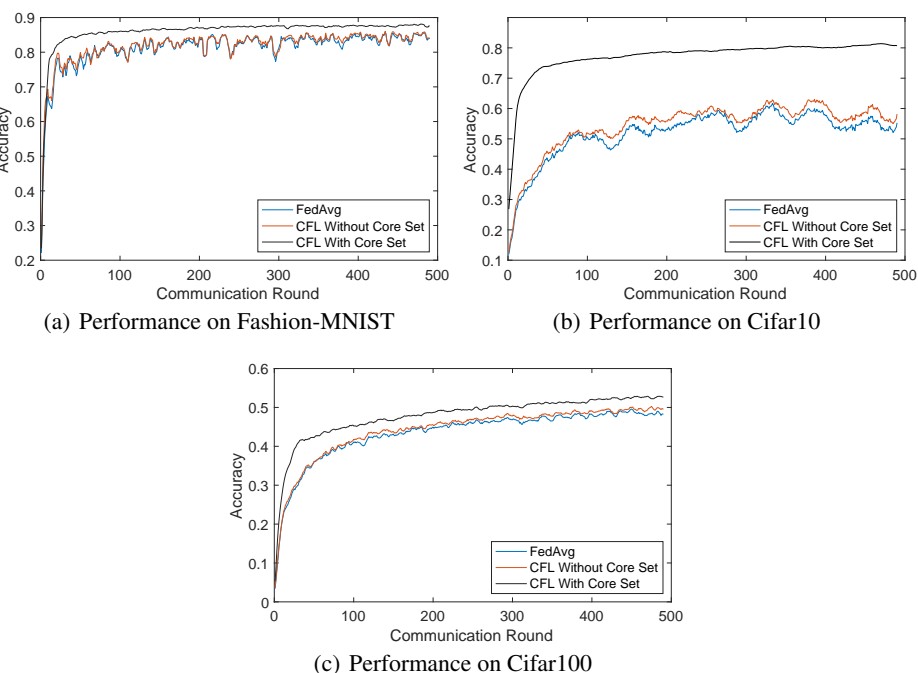

(a) Performance on Fashion-MNIST

(b) Performance on Cifar10

(c) Performance on Cifar100

Figure 4: Models are trained on various datasets with $\alpha = 0.1$. CFL Without Core Set method use regularization methods, and CFL With Core Set methods use core set methods. All these two CFL algorithms use FedAvg as backbone.

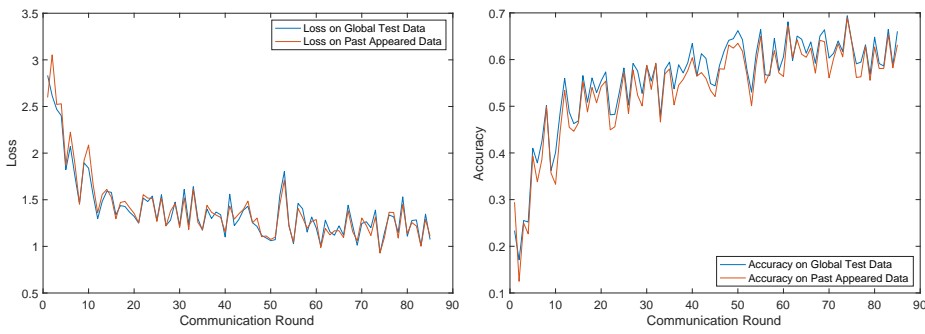

(a) Loss on global test data and past appeared train data

(b) Accuracy on global test data and past appeared train data

Figure 5: Evaluation on global test data and past appeared training data. We trained ResNet18 on split-Cifar10 dataset with $\alpha = 0.2$ for 85 rounds.

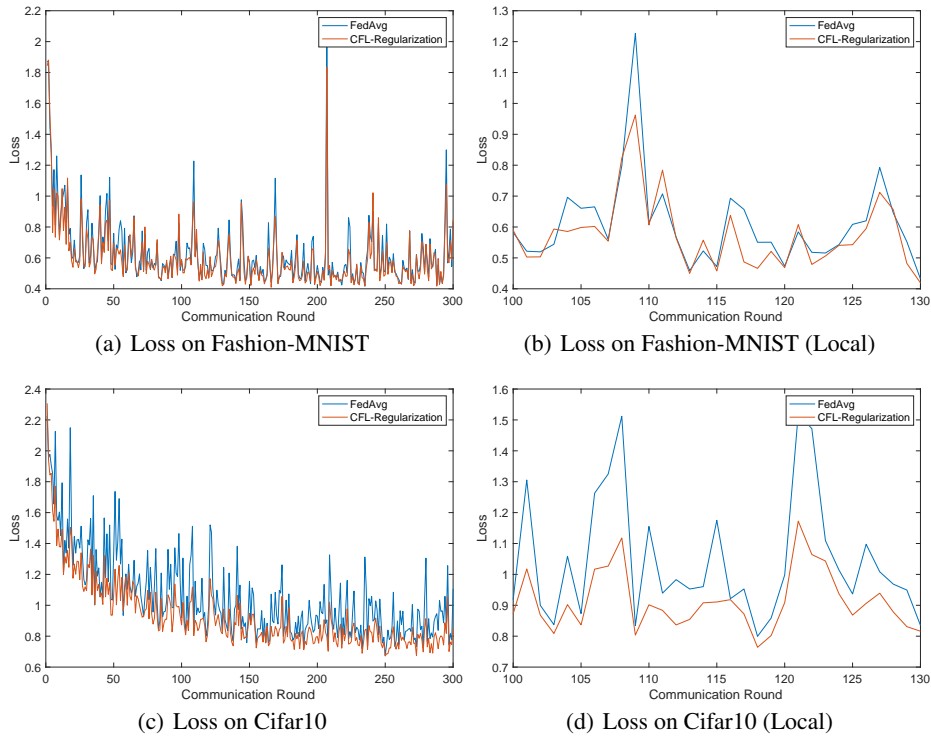

Figure 6: Models are trained on split-Fashion-MNIST and split-Cifar10 datasets with $\alpha = 0.1$. The loss is evaluated on global test datasets. Left column is the full curve of 300 rounds, and figures in right column are partially enlarged curves.

---

**Algorithm 4** Data splitting

**Require:** $S$: a list of datasets split by labels, $M$: the number of clients, $N$: the size of local datasets, $\alpha$: the Dirichlet distribution parameter

**Ensure:** $D$: split datasets

1: $D \leftarrow []$
2: $G \leftarrow [0, 1, 2, ..., len(S)]$
3: **for** $m = 1, 2, ..., M$ **do**
4:     $p = [p_1, p_2, ..., p_{len(S)}]$, where $p_{t,i}$ denotes the fraction of class $i$ in total dataset.
5:     $\theta \leftarrow Dirichlet(\alpha, p)$
6:     $D_m \leftarrow \emptyset$
7:     **while** $len(D_m) < N$ **do**
8:         $i \leftarrow multinomial(\theta, 1)$
9:         $y \leftarrow G[i]$
10:         $data \leftarrow uniform(S[y])$
11:         $D_m \leftarrow D_m \cup \{data\}$
12:         $S[y] \leftarrow S[y]/data$
13:         **if** len(S[y]) == 0 **then**
14:             $G \leftarrow G/y$
15:             $\theta \leftarrow renormalize(\theta, i)$
16:     $D.append(D_m)$

---

**Algorithm 5** renormalize

**Require:** $\theta$: weights for different classes, $i$: the class that should be deleted

**Ensure:** $\theta$: renormalized weights

1: **for** $j = 1, 2, ..., len(\theta), j \neq i$ **do**
2:     $\theta[j] \leftarrow \theta[j]/sum(theta/theta[i])$

**Algorithm 6** SplitdataMain

---

**Require:** $S$: total dataset split by class, $T$: rounds, $K$, $N$, $\alpha$, $\beta$
**Ensure:** $D_{final}$: split dataset
1:  $D \leftarrow splitData(S, K, N, \alpha)$
2:  $D_{final} \leftarrow []$
3:  **for** $i = 1, 2, ..., K$ **do**
4:     $S_i \leftarrow splitByClass(D[i])$
5:     $N_{local} \leftarrow len(D_m)/T$
6:     $D_i \leftarrow splitData(S_i, cluster\_num, N_{local}, \beta)$
7:     $D_{final} \leftarrow D_{final} \cup D_i$

---

