# OpenReview forum: "Towards Federated Learning on Time-Evolving Heterogeneous Data"
_ICLR.cc/2022/Conference — ICLR 2022 Submitted_

### Official Review · Reviewer_87pq · 2021-10-31

**Correctness:** 2
**Technical Novelty And Significance:** 2
**Empirical Novelty And Significance:** 2
**Recommendation:** 3
**Confidence:** 4

**Main Review:**

My main concern is about the novelty brought by the paper:

* As the first main contribution, authors list a novel CFL framework. I assume that authors refer to Eq. CFL as t
he "CFL framework". However, it seems that (CFL) is just a special case of the general FL objective (1).
Maybe authors mean by CFL framework the data access model, i.e., online gathering of Local datasets over time?
If (CFL) framework does not target an online data access model, I don't see the fundamental difference
between (CFL) and total variation type models that have already been proposed, e.g., in

A. Jung, "Networked Exponential Families for Big Data Over Networks," in IEEE Access, vol. 8, pp. 202897-202909, 2020, doi: 10.1109/ACCESS.2020.3033817.

SarcheshmehPour, Y., Tian, Y., Zhang, L., and Jung, A., “Networked Federated Multi-Task Learning”, <i>arXiv e-prints</i>, 2021. https://arxiv.org/abs/2105.12769


D. Hallac, J. Leskovec, and S. Boyd, Network Lasso: Clustering and Optimization in Large Graphs, Proceedings SIGKDD, pages 387-396, 2015.


These papers consider local models assigned to nodes of a graph or networks. These nodes could represent different time periods of the time-evolving datasets.

* As the second main contribution authors list a convergence analysis of a gradient method for (CFL). However, it is not clear how the resulting convergence rates are better than what can be obtained from existing analysis techniques for (online) SGD of strongly convex functions, see e.g.,

[1]Rakhlin, A., Shamir, O., and Sridharan, K., “Making Gradient Descent Optimal for Strongly Convex Stochastic Optimization”, <i>arXiv e-prints</i>, 2011. https://arxiv.org/abs/1109.5647

The convergence rates depicted in Table I need more discussion. It is not clear to me how Table I shows that the proposed convergence rates are substantially
better than rates of known FL methods.


* As the third main contribution authors mention approximation techniques. However, it is unclear what precise approximation
techniques are proposed for solving (CFL). The relevant Section 3.2 only introduces a measure (information loss) for the approximation quality but
does not discuss how precisely these approximations will be constructed for (CFL).

* The main theoretical results of the paper are bounds on the number of iterations/rounds required to achieve prescribed accuracy. These bounds involve several parameters used by Assumption 1 - 4. However, i do not see how the current numerical experiments verify these bounds on the required rounds for achieving prescribed accuracy.

The clarity of presentation and quality of language needs improvement:

* what is the output of Algorithm 1 ? Is Algorithm 1 meant to be run in an online fashion ? Otherwise the time horizon T should be listed as input parameter. How precisely is S_{t} constructed? The summation in step 6 seems to have flipped a \tau for a t in the subscript of \omega

* "...St is a subset of clients sampled from all clients set..." How precisely is this set sampled ?

* what is a "cross-device setup"?

* "We give the formal definition of CFL framework in Algorithm 1."  pls avoid the term "framework" when you actually mean a specific algorithm (your new algorithm 1)

* "...the previous local object functions cannot be properly approximated." ps provide more justification for this "non-approximhatability"

* Definition 3.1 talks about approximating loss functions but Eq. (3) is about gradients

* "To analyze Equation (2) and Algorithm 1 in-depth, we propose the Gradient Noise Model..." It seems a bit unusual to use a gradient noise model for analysing an  algorithm. The gradient noise model might be prescribed by the application setup (data access model)

* should the LHS of (4) and (5) involve the gradient of local loss functions instead of the global loss function ?

* "...and in some conditions, μ-convex."

* Assumption 2 and 3 involve conditional expectations given parameter vector \omega. Pls provide more details about the proability distribution of the random variable \omega.

* There should be more separation between the cells of the last row in Table I.

* "We prove this comment..." I would rather say to prove a theorem/lemma or claim.

* "..can be bounded by an arbitrary non-negative value..." So we can choose this value arbitrarily small ?

* "... we analyze the theoretical performance of Algorithm 1 ..."

* "..., despite that the term induced by information loss concurrently trades off the optimization...."  Pls refer to the term using \eqref{}

* what is a "...regular linear convergence term..."?

* "...the model instead reaches the neighborhood of..." Do you mean "iterates" by "model"?

* what is a "...smoother optimization phase."?

* "We consider to federated learn ..."

* "...this (most) challenging time-varying scenario mimics the realistic client data sampling scheme (from some underlying distributions). " in what precise sense is this the most challenging scenario ?

* "Note that the trade-off between Hessian estimation and computational overhead constrains the practical feasible of such approach." Unclear

* "... as a method to approximate diagonal Hessian matrix, is slightly preferable than Fisher Information Matrix, though the latter one involves less computation."

* what is "...challenging non-overlapping time-varying heterogeneous data" ?

* "...we are the first to provide such theoretical guarantees ..." what precisely do you mean by "such guarantees"?

**Summary Of The Paper:**

The authors study Federated Learning for time-evolving data. In particular, they study gradient based methods for learning parameter vectors and characterize their performance in terms of the approximation quality of a noisy gradient oracle. Numerical experiments compare the performance of the proposed method with existing FL methods.


**Summary Of The Review:**

My main concern is about the novelty brought by the paper:

* As the first main contribution, authors list a novel CFL framework. I assume that authors refer to Eq. CFL as t
he "CFL framework". However, it seems that (CFL) is just a special case of the general FL objective (1).
Maybe authors mean by CFL framework the data access model, i.e., online gathering of Local datasets over time?
If (CFL) framework does not target an online data access model, I don't see the fundamental difference
between (CFL) and total variation type models that have already been proposed, e.g., in

A. Jung, "Networked Exponential Families for Big Data Over Networks," in IEEE Access, vol. 8, pp. 202897-202909, 2020, doi: 10.1109/ACCESS.2020.3033817.

SarcheshmehPour, Y., Tian, Y., Zhang, L., and Jung, A., “Networked Federated Multi-Task Learning”, <i>arXiv e-prints</i>, 2021. https://arxiv.org/abs/2105.12769


D. Hallac, J. Leskovec, and S. Boyd, Network Lasso: Clustering and Optimization in Large Graphs, Proceedings SIGKDD, pages 387-396, 2015.


These papers consider local models assigned to nodes of a graph or networks. These nodes could represent different time periods of the time-evolving datasets.

* As the second main contribution authors list a convergence analysis of a gradient method for (CFL). However, it is not clear how the resulting convergence rates are better than what can be obtained from existing analysis techniques for (online) SGD of strongly convex functions, see e.g.,

[1]Rakhlin, A., Shamir, O., and Sridharan, K., “Making Gradient Descent Optimal for Strongly Convex Stochastic Optimization”, <i>arXiv e-prints</i>, 2011. https://arxiv.org/abs/1109.5647

The convergence rates depicted in Table I need more discussion. It is not clear to me how Table I shows that the proposed convergence rates are substantially
better than rates of known FL methods.


* As the third main contribution authors mention approximation techniques. However, it is unclear what precise approximation
techniques are proposed for solving (CFL). The relevant Section 3.2 only introduces a measure (information loss) for the approximation quality but
does not discuss how precisely these approximations will be constructed for (CFL).

* The main theoretical results of the paper are bounds on the number of iterations/rounds required to achieve prescribed accuracy. These bounds involve several parameters used by Assumption 1 - 4. However, i do not see how the current numerical experiments verify these bounds on the required rounds for achieving prescribed accuracy.

The clarity of presentation and quality of language needs improvement:

* what is the output of Algorithm 1 ? Is Algorithm 1 meant to be run in an online fashion ? Otherwise the time horizon T should be listed as input parameter. How precisely is S_{t} constructed? The summation in step 6 seems to have flipped a \tau for a t in the subscript of \omega

* "...St is a subset of clients sampled from all clients set..." How precisely is this set sampled ?

* what is a "cross-device setup"?

* "We give the formal definition of CFL framework in Algorithm 1."  pls avoid the term "framework" when you actually mean a specific algorithm (your new algorithm 1)

* "...the previous local object functions cannot be properly approximated." ps provide more justification for this "non-approximhatability"

* Definition 3.1 talks about approximating loss functions but Eq. (3) is about gradients

* "To analyze Equation (2) and Algorithm 1 in-depth, we propose the Gradient Noise Model..." It seems a bit unusual to use a gradient noise model for analysing an  algorithm. The gradient noise model might be prescribed by the application setup (data access model)

* should the LHS of (4) and (5) involve the gradient of local loss functions instead of the global loss function ?

* "...and in some conditions, μ-convex."

* Assumption 2 and 3 involve conditional expectations given parameter vector \omega. Pls provide more details about the proability distribution of the random variable \omega.

* There should be more separation between the cells of the last row in Table I.

* "We prove this comment..." I would rather say to prove a theorem/lemma or claim.

* "..can be bounded by an arbitrary non-negative value..." So we can choose this value arbitrarily small ?

* "... we analyze the theoretical performance of Algorithm 1 ..."

* "..., despite that the term induced by information loss concurrently trades off the optimization...."  Pls refer to the term using \eqref{}

* what is a "...regular linear convergence term..."?

* "...the model instead reaches the neighborhood of..." Do you mean "iterates" by "model"?

* what is a "...smoother optimization phase."?

* "We consider to federated learn ..."

* "...this (most) challenging time-varying scenario mimics the realistic client data sampling scheme (from some underlying distributions). " in what precise sense is this the most challenging scenario ?

* "Note that the trade-off between Hessian estimation and computational overhead constrains the practical feasible of such approach." Unclear

* "... as a method to approximate diagonal Hessian matrix, is slightly preferable than Fisher Information Matrix, though the latter one involves less computation."

* what is "...challenging non-overlapping time-varying heterogeneous data" ?

* "...we are the first to provide such theoretical guarantees ..." what precisely do you mean by "such guarantees"?

---

> ### Author Response · Authors · 2021-11-17
> **To Reviewer 87pq (1/3)**
>
> Dear Reviewer 87pq,
>
> We would like to thank you for your time spent reviewing our paper. We have polished the draft with blue color to avoid typos and unclear statements. Please find our responses to your raised main questions below:
>
> ----
>
> > Clarify: The definition of cross-device setup
>
> Sorry for the confusion. We have added reference [1] for clarification. As defined in the popular FL survey [1], cross-device and cross-silo are now two standard settings considered by the FL community.
> The cross-device setting indicates that (1) the clients are a very large number of mobile or IoT devices, (2) only a fraction of clients are available at any one time, and (3) clients may be stateless. At the same time, cross-silo setting refers to training a model on siloed data, where clients are different organizations (e.g., medical or financial) or geo-distributed datacenters.
>
> [1] Kairouz P, McMahan H B, Avent B, et al. Advances and open problems in federated learning[J]. Foundations and Trends in Machine Learning, 2020.
>
> ----
>
> > Clarify: "...the previous local object functions cannot be properly approximated." ps provide more justification for this "non-approximability"
>
> Under the considered time-varying CFL setup, the local client objective function determined by the local dataset will evolve over time. The objective of CFL formulation considers optimizing all these local objective functions over time and clients. However, due to the privacy-preserving rule of the training data, all previous local objective functions, determined by the client local dataset, are inaccessible. To this end, we need to approximate these local objective functions that occurred in previous rounds.
>
> We believe our formulation is sensible and crucial for the continual federated learning scenario to tackle the well-known catastrophic forgetting issue.
>
> ----
>
> > Clarification and novelty.
>
> **We believe the reviewer might have misinterpreted key aspects of our submission.**
> To the best of our knowledge,
> 1. **This is the first work that formally defines the CFL**. Please note that the traditional FL formulation is only a special case of our CFL formulation, while the opposite is wrong, as
> 	* CFL considers the time-varying local (client) objective functions, captured by time- and client- drift, while FL only considers the fixed client objective functions.
> 	* CFL formulation can be simplified to traditional FL formulation by setting $p_{t, i} = 1$, but it is impossible for traditional FL to cover the time-varying scenarios due to the missing modeling of the time-varying aspects. It is non-trivial to improve the traditional FL formulation to the time-varying setup.
> 2. **The proposed CFL framework is novel**, covering different approximation techniques for diverse time-varying FL scenarios.
> 	* In Section 3.1, we introduce the definition of information loss (Definition 3.1) to measure the approximation quality. This definition, together with Assumption 4, Remark 3.3, leads to the novel theoretical results for CFL in Section 4.
> 	* We introduce three types of approximation techniques in Section 5.1 and elaborate its details, strengths, and weaknesses in Appendix C.2.2. We further empirically justify its practical preciseness and insights using 2.5 pages of numerical results.
> 	* Due to intractable analysis of many considered function approximation methods, we use the core-set method in Figure 1 to empirically justify the correlation between information loss $R$ and learning performance. The observation in Figure 1 is aligned with our theoretical assumptions/results: *the lesser the information loss, the higher the performance.*
> 3. **Our setup is different from the total variation type models pointed out by the reviewer.**
> 	* [2, 3] consider the networked data, where the networked data is represented by an undirected empirical graph whose edges connect similar datasets. This strong assumption normally does not hold in the generic federated learning setup, where (1) the community uses a star network topology, (2) clients have their own participation pattern and will not connect with other clients with similar datasets (it also violates the key privacy concern).
> 	* The total variation type models mentioned in [2, 3] act only as a regularizer to enforce similar weights on different nodes, and we are unaware of how will it connect with our proposed setup.

---

> ### Author Response · Authors · 2021-11-17
> **To Reviewer 87pq (2/3)**
>
> 4. **Our setup is different from the idea of using different nodes to represent different time periods of the time-evolving datasets, mentioned by the reviewer.** The reasons come from the fact that
> 	* Different optimization objectives. The local objective function of clients in the considered CFL only gradually evolves over time. The local objective function from previous rounds cannot be accessed directly, while the simulation idea will consider all nodes (of different time steps) at once.
> 	* The simulation idea cannot model the time-drift of a given client, while our CFL can.
> 	* One potential workaround could be adding time-varying graph connectivity to the simulation idea, but it is a non-trivial orthogonal extension, and we are unaware of any prior work.
> 5. **We provide the first theoretical results for the time-varying CFL, which are novel and significant.**
> 	* Providing a tighter rate for traditional FL setup is beyond the scope of this work. The prior FL rates, developed on the traditional static FL scenario, are infeasible to compare with our new rates analyzed for the challenging time-varying CFL scenarios due to the completely different FL setups.
> 	* Our CFL framework and analysis can be simplified to traditional FL setup and traditional CL setup, but not the other way. Compared to the rate of Yin et al. 2020b under the traditional CL setup for GD, we can provide a tighter rate for CL on mini-batch SGD.
> 	* It is unfair to compare the results in [4], as it does not consider the distributed case and cannot capture/analyze the challenges like the data heterogeneity that emerged in FL.
>
> [2] Networked Federated Multi-Task Learning, https://arxiv.org/abs/2105.12769.
> [3] Networked Exponential Families for Big Data Over Networks, IEEE Access, https://ieeexplore.ieee.org/document/9239959.
> [4] Making Gradient Descent Optimal for Strongly Convex Stochastic Optimization, https://arxiv.org/abs/1109.5647.
>
> ----
>
> > Theory and experiments
>
> We did include the convex experiments to verify our theoretical results for convex and strongly-convex functions, as shown in Appendix C.1.
> We completely agree with the reviewer and we have added the non-convex proof in Theorem 4.2 (blue colored) in the revised draft to strengthen the draft; the proof details are deferred in Appendix B.5. These theoretical results---verify the faster convergence of CFL than that of FedAvg---are aligned with our convex and non-convex deep learning experiments.
>
> ----
>
> > Comment on the traditional online data access model
>
> The traditional online optimization is also different from the time-varying CFL scenario we considered here.
> * FL is motivated from the privacy concern aspect and data will be kept locally on each device, while traditional online optimization e.g. [5, 6, 7] normally does not consider these constraints.
> * The data heterogeneity challenge across different clients in FL cannot be captured by the analysis of traditional online optimization.
> * Different analysis techniques. Most of the works in FL analyze the convergence rate of local SGD under the gradient noise models [11, 12, 13, 17], while traditional online optimization considers regret bound.
> * We believe introducing traditional online learning to the community of FL is valuable, but it is beyond the scope of this work.
>
> [5] Online learning and online convex optimization. Foundations and Trends in Machine Learning, 2011.
> [6] Adaptive subgradient methods for online learning and stochastic optimization. Journal of machine learning research, 2011.
> [7] Online Passive-Aggressive Algorithms, JMLR 2006.
>
> ----
>
> > What is the output of Algorithm 1? Is Algorithm 1 meant to be run in an online fashion? Otherwise, the time horizon T should be listed as an input parameter. How precisely is S_{t} constructed? The summation in step 6 seems to have flipped a \tau for at in the subscript of \omega
>
> Sorry for the confusion. We have polished the confusion parts in the draft (blue colored), including the $T$ in our algorithm, the typo in step 6, and the client sampling choice of $S_t$.
>
> Regarding the construction of $S_t$, we follow the most common practice in FL, and uniformly sample a small fraction of clients per communication round. Note that enabling optimal client sampling is also an open research area in FL [8, 9, 10], and it is beyond the scope of our study.
>
> [8] Rizk E, Vlaski S, Sayed A H. Federated learning under importance sampling[J]. arXiv preprint arXiv:2012.07383, 2020.
> [9] Wang S, Lee M, Hosseinalipour S, et al. Device Sampling for Heterogeneous Federated Learning: Theory, Algorithms, and Implementation[J]. arXiv preprint arXiv:2101.00787, 2021.
> [10] Fraboni Y, Vidal R, Kameni L, et al. Clustered Sampling: Low-Variance and Improved Representativity for Clients Selection in Federated Learning[J]. arXiv preprint arXiv:2105.05883, 2021.
>
> ----

---

> ### Author Response · Authors · 2021-11-17
> **To Reviewer 87pq (3/3)**
>
> ----
>
> > “Definition 3.1 talks about approximating loss functions but Eq. (3) is about gradients”
>
> Similar to most FL works e.g. [11, 12, 13, 17], we consider the SGD-based method as the backbone algorithm. Thus, the difference between approximated function and original function only matters in its gradient space.
>
> ----
>
> > “It seems unusual to use a gradient noise model...”
>
> The gradient noise model is a widely used tool to analyze stochastic gradient descent problems [14, 15, 16]. Most theoretical works on FL/decentralized SGD also rely on this assumption, e.g., [11, 12, 13, 17]; please check a more complete list in our related work section.
>
> [11] Xiang Li, Kaixuan Huang, Wenhao Yang, Shusen Wang, and Zhihua Zhang. On the convergence of fedavg on non-iid data. In International Conference on Learning Representations, 2020b. URL https://openreview.net/forum?id=HJxNAnVtDS.
> [12] Ahmed Khaled, Konstantin Mishchenko, and Peter Richtárik. Tighter theory for local sgd on identical and heterogeneous data. In International Conference on Artificial Intelligence and Statistics, pp. 4519–4529. PMLR, 2020.
> [13] Sai Praneeth Karimireddy, Satyen Kale, Mehryar Mohri, Sashank Reddi, Sebastian Stich, and Ananda Theertha Suresh. Scaffold: Stochastic controlled averaging for federated learning. In International Conference on Machine Learning, pp. 5132–5143. PMLR, 2020b.
> [14] Sievert S. Improving the convergence of SGD through adaptive batch sizes[J]. arXiv preprint arXiv:1910.08222, 2019.
> [15] Shamir O, Zhang T. Stochastic gradient descent for non-smooth optimization: Convergence results and optimal averaging schemes[C]//International conference on machine learning. PMLR, 2013: 71-79.
> [16] Gower R M, Loizou N, Qian X, et al. SGD: General analysis and improved rates[C]//International Conference on Machine Learning. PMLR, 2019: 5200-5209.
> [17] Anastasia Koloskova, Nicolas Loizou, Sadra Boreiri, Martin Jaggi, and Sebastian Stich. A unified theory of decentralized sgd with changing topology and local updates. In International Conference on Machine Learning, pp. 5381–5393. PMLR, 2020.
>
> ----
>
> > “should the LHS of (4) and (5) involve the gradient of local loss functions….”
>
> Equation 4 is equivalent to the widely used bounded gradient dissimilarity, e.g., in [11, 12, 13, 17]. We showcase the equivalence between this common bounded gradient dissimilarity (in modeling client drift) and our zero-mean client drift formulation (i.e., Equation 4) in Lemma B.1. of Appendix B.
>
> ----
>
> > “Pls provide more details about the probability distribution of the random variable \omega.”
>
> $\omega$ is the model parameter, e.g., weights of neural networks, and its distribution is determined by its random initialization, training dataset, optimizer, learning rate schedule, etc. It is non-trivial to characterize its probability distribution in a principled way, and we are unaware of any prior work on this.
>
> ----
>
> > “Can we choose $R$ arbitrarily small?”
>
> A smaller R provides a better convergence rate; however, in practice, it cannot be arbitrarily small and will be upper bounded (please check our results in Figure 1).
>
> Please also check [our response to Reviewer s89S](https://openreview.net/forum?id=oxC2IBx8OuZ&noteId=yjQbvZrmQSR) (titled `About Assumption 4`) for a detailed explanation.
>
> ----
>
> > “The challenging non-overlapping time-evolving scenarios”
>
> The scenario refers to client stateless defined in the cross-device setting [1], where “each client will likely participate only once in training, so generally a fresh sample of never-before-seen clients in each round of computation is assumed.”
>
> This challenging scenario is equivalent to training with non-overlapping local datasets, and we report the results in Table 6, where we exclude the core-set method as it is infeasible for the stateless scenario. Regarding the theoretical feasibility of such extension, please also check [our response to Reviewer 2NMQ](https://openreview.net/forum?id=oxC2IBx8OuZ&noteId=AzUkyU7IfPY), under the title of `CFL extension to client-only-participate-once (stateless) FL setting`.
>
> ----

---

> ### Author Response · Authors · 2021-11-20
> **Kind reminder for our response**
>
> Dear Reviewer 87pq,
>
> We thank you again for providing timely feedback. As we believe we have addressed your main concerns in your review, we would therefore like to ask you if you could reconsider your score? We remain at your disposal for any further clarification which you might require.

---

### Official Review · Reviewer_s89S · 2021-11-02

**Correctness:** 3
**Technical Novelty And Significance:** 4
**Empirical Novelty And Significance:** 3
**Recommendation:** 8
**Confidence:** 3

**Main Review:**

This paper is well-written. It is easy to follow the narration, all assumptions are clearly described. The motivation of the problem is explained in detail. The notation might be confusing, for example $\Delta \boldsymbol{\omega}_{t, i}=\boldsymbol{\omega}_{t, i, K}-\boldsymbol{\omega}_{t}$ has three subscript indexes, which is not comfortable to read. It might be useful to use superscript.

Table 1 is illustrative, but it can be useful to add more paper with analysis of FedAvg, such as https://arxiv.org/pdf/2006.04735.pdf. It is not clear what is the CL rate of Yin et al. (2020b). Assumption 1 is not well-described.
Assumption 1 (Smoothness and convexity). Assume local objective functions $f_{t, i}(\boldsymbol{\omega})$ are $L$-smooth, and in some conditions, $\mu$-convex. If a function $f_{t, i}$ is both $L$-smooth and $\mu$-convex, then it satisfies $\frac{1}{2 L}\left\|\nabla f_{t, i}(\mathbf{x})-\nabla f_{t, i}(\mathbf{y})\right\|^{2} \leq f_{t, i}(\mathbf{x})-f_{t, i}(\mathbf{y})-\nabla f_{t, i}(\mathbf{x})^{T}(\mathbf{x}-\mathbf{y})$, and $L \geq \mu$

In the formula, there is no any $\mu$. It can be useful to have definitions of $L$-smooth function and $\mu$ strong convexity.

Assumption 2 is limited.

Assumption $\mathbf{2}$ (Bounded noise in stochastic gradient). Let $g_{t, i, k}(\boldsymbol{\omega})=\nabla f_{t, i, k}(\boldsymbol{\omega})+\boldsymbol{\nu}_{t, i, k}$, where $\boldsymbol{\nu}_{t, i, k}$ is the stochastic noise of client $i$ on round $t$ at $k$-th local update step. We assume that $\mathbb{E}[\boldsymbol{\nu} \mid \boldsymbol{\omega}]=0$, and $\mathbb{E}\left[\|\boldsymbol{\nu}\|^{2} \mid \boldsymbol{\omega}\right] \leq \sigma^{2}$

In the paper https://arxiv.org/pdf/1901.09401.pdf, it is shown that bounded variance assumption cannot capture all settings. In the paper https://arxiv.org/pdf/1909.04746.pdf, all results do not require a bounded variance assumption.

Assumption 4 is questionable.

Assumption 4 (Bounded information loss). We assume the information loss $\Delta_{t, i}(\boldsymbol{\omega})$ can be bounded by an arbitrary non-negative value $R$, i.e. $\left\|\Delta_{t, i}(\boldsymbol{\omega})\right\| \leq R$.

It seems not realistic to have a constant bound for information loss, can you please show when this assumption holds in practice?

Theoretical results seem to be sound. Unfortunately, this paper does not propose any non-convex analysis. This part is needed since experiments are done only for deep learning models, which are non-convex.

This paper has a wide set of different experiments. A broad comparison of different approximation approaches is done. Moreover, the authors compare a new method with other methods and results show that the new method outperforms other methods. It might be interesting to compare with a large number of federated learning algorithms.

The main issue of the experimental part is the lack of convex and strongly convex models. The theory is done only for convex and strongly convex settings, but experiments are done for deep learning models, so experiments cannot support the theory.


**Summary Of The Paper:**

This paper introduces the Continual Federated Learning framework that allows capturing time-evolving heterogeneity of FL. The authors introduce a new formulation of the problem and propose a carefully chosen approximation to work with the new problem. Also, the authors provide theoretical analysis for strongly convex and general convex settings. At the end of the paper, they provide experimental results for deep learning models.

**Summary Of The Review:**

This paper proposes a novel framework and considers the important problem. The theoretical results are obtained only for convex and strongly convex cases. Some assumptions are questionable and analysis can be generalized. Experiments are done only for deep learning models and cannot support the theory.

I think that novelty of the paper is significant, but the paper has some issues. Overall, the paper is marginally above the acceptance threshold.

---

> ### Author Response · Authors · 2021-11-17
> **To Reviewer s89S**
>
> Dear Reviewer s89S,
>
> We would like to thank you for your time spent reviewing our paper and for providing constructive comments. Please find our responses to your raised questions below:
>
> ----
>
> > “Table 1 is illustrative, but it can be useful to add more papers with analysis of FedAvg”.
>
> Thanks for the suggestion. In the revised draft, we have included the convergence results of [1] in Table 1, as suggested by the reviewer.
>
> Besides, it is worthwhile to note that our main contribution is the convergence rate of CFL, a new scenario that has not been studied by prior theory. Providing tighter rates for traditional FL setup is beyond the scope of this paper.
>
> [1] Woodworth B, Patel K K, Srebro N. Minibatch vs local sgd for heterogeneous distributed learning[J]. arXiv preprint arXiv:2006.04735, 2020. https://arxiv.org/pdf/2006.04735.pdf
>
> ----
>
> > The convergence rate of Yin et al. (2020b).
>
> The result of Yin et al. (2020b) we presented in Table 1 is exactly the results presented in the original paper. Compared to the convex and non-convex GD rates in Yin et al. (2020b) analyzed for centralized CL case, we use completely different proof techniques and provide tighter results, for mini-batch SGD on both CL and CFL, in terms of convex, strongly convex, non-convex functions.
> We can also recover the rates of GD by setting variance terms to zero, and matching the $\frac{1}{\epsilon}$ term in Yin et al. (2020b).
>
> ----
>
> > Assumption 1 is not well-described
>
> Sorry for the confusion. The previous Assumption 1 is only the corollary of $\mu$-convex and $L$-smooth, and we have replaced it with their original definition in the revised draft (blue colored). Please note that it will not change the proofs and our theoretical results.
>
> ----
>
> > Assumption 2 is limited.
>
> Thank you for pointing out an interesting improvement direction for future work.
> As the first theoretical work on modeling the challenging time-varying CFL scenarios, we follow the most common bounded noise assumption used in the FL community [2, 4, 5],  as our first step. We have commented on this in the revision and will leave the idea of using a relaxed assumption [3]---bounded noise at optimum---as our future work.
>
> [2] Xiang Li, Kaixuan Huang, Wenhao Yang, Shusen Wang, and Zhihua Zhang. On the convergence of fedavg on non-iid data. In International Conference on Learning Representations, 2020b. URL https://openreview.net/forum?id=HJxNAnVtDS.
> [3] Ahmed Khaled, Konstantin Mishchenko, and Peter Richtárik. Tighter theory for local sgd on identical and heterogeneous data. In International Conference on Artificial Intelligence and Statistics, pp. 4519–4529. PMLR, 2020.
> [4] Sai Praneeth Karimireddy, Satyen Kale, Mehryar Mohri, Sashank Reddi, Sebastian Stich, and Ananda Theertha Suresh. Scaffold: Stochastic controlled averaging for federated learning. In International Conference on Machine Learning, pp. 5132–5143. PMLR, 2020b.
> [5] Anastasia Koloskova, Nicolas Loizou, Sadra Boreiri, Martin Jaggi, and Sebastian Stich. A unified theory of decentralized sgd with changing topology and local updates. In International Conference on Machine Learning, pp. 5381–5393. PMLR, 2020.
>
> ----
>
> > About Assumption 4
>
> **The observations in Figure 1 are aligned with Assumption 4 and showcase that our bounded assumption is realistic in practice**: (1) it is sufficient to assume $R$ to be a relatively upper bound (arbitrarily large in our Assumption 4), and (2) the lesser the information loss, the higher the performance.
> More precisely, in Figure 1, we empirically investigate the effect of information loss on deep models and real datasets through the core-set method. This design choice comes from the fact of (1) intractable analysis of many considered function approximation methods (in Section 5), and (2) the size indeed is a good indicator of different approximation qualities (larger is better).
>
> We add a new Remark 3.3 to complement Assumption 4. This remark provides an analysis of the bounded information loss on the feasible Taylor-Extension-based approximation method.
> Together with the convex numerical results under the Noisy Quadratic Model in Appendix C.1,  we provide a smooth transition between theoretical analysis and experimental insights.
>
> ----
>
> > Theory and experiments
>
> We did include the convex experiments to verify our theoretical results for convex and strongly-convex functions, as shown in Appendix C.1.
> We completely agree with the reviewer and we have added the non-convex proof in Theorem 4.2 (blue colored) in the revised draft to strengthen the draft; the proof details are deferred in Appendix B.5. These theoretical results---verify the faster convergence of CFL than that of FedAvg---are aligned with our convex and non-convex deep learning experiments.
>
> ----

---

> > ### Comment · Reviewer_s89S · 2021-11-29
> > **Response**
> >
> > I thank authors for their response. They clarified and answered the majority of my questions. I believe this paper opens new directions in federated learning and it can be influential. I raised my score since many issues were resolved.

---

### Official Review · Reviewer_XVxz · 2021-11-02

**Correctness:** 3
**Technical Novelty And Significance:** 2
**Empirical Novelty And Significance:** 2
**Recommendation:** 3
**Confidence:** 3

**Main Review:**

**Limited novelty:** The technical novelty of this paper is relatively limited. Using a replay buffer to approximate historical objectives is a well-known approach in continual learning. Extending it to FL seems to be straightforward as long as the replay buffer remains local to each client. The theoretical analysis is somewhat interesting but it relies on strong assumptions such as convexity (more details below), which does not hold for deep neural networks. Furthermore, the result is a bit cumbersome and difficult to interpret, and the analysis technique and result appear to be quite similar to Yin et al. (2020b).

**Unrealistic assumptions:** It is evident that objectives related to deep learning models are not convex, so it would be good to analyze the convergence of non-convex objectives as well. Moreover, I feel that the assumption that the time drift is independent across both $t$ and $i$ (Remark 3.2) is a fairly strong assumption. Usually, the data varies gradually over time, so the time drift should also change gradually over time $t$, instead of being independent for each $t$. I further do not understand why $\delta_i$ needs to be zero-mean. Usually an upper bound of the gradient drift across clients is sufficient for the analysis. In general, considering gradient drifts as random variables is problematic, since you are talking about the full (i.e., non-stochastic) gradients in the definition of drifts. These gradients only depend on the model parameter $\omega$ and there should be no other random source (unless you consider the time evolution of data to be random, but even then it should be at least Markovian instead of fully independent).

**Interpretation of results:** The results are generally cumbersome. It would be worthwhile to see whether some of the terms can be simplified in the $\\mathcal{O}(\\cdot)$ notation. I do not understand how Section 4.2 is related to time-varying scenarios. In particular, it is not clear why setting $p_{t,i}=1$ would eliminate $R$. Intuitively it does not seem to make sense, since the information loss (upper bounded by $R$) should affect the convergence even if $p_{t,i}=1$. I also do not understand what "CFL methods accelerates the convergence by reducing the variance term" means in Remark 4.5. While I see that the result in Theorem 4.4 has less terms compared to the result in Theorem 4.2, it is unclear how those terms are related to the variance.

**Experiments:** When talking about generative methods, the paper mentions "Maintaining a core set for each client may become impractical when
learning scales to millions of clients". I'm not sure why the core set needs to be maintained collectively across different clients, and if this is the case in the core set method evaluated in the experiments, it may cause privacy issues. It seems sufficient for each client to separately select and manage its own core set. In other words, assuming that each client has a fixed amount of storage, whenever new data arrives, it has to delete some old data. The client can choose to delete the data samples that are less representative, and increase the weights of the remaining representative data samples. In this way, a core set would be constructed on each client alone, without involvement of other clients, so there will not be a scalability issue. In general, the use of replay buffer (e.g., in the core set method) for continual learning is quite standard.

Minor: The paper should have a conclusion section highlighting the main results and findings.

**Summary Of The Paper:**

The paper presents continual federated learning (CFL) to address the issue that data at clients in FL may vary over time. The main objective of CFL is to optimize the time-aggregated objective across multiple clients. While historical objective functions are usually not available, the paper considers a surrogate objective that approximates the original historical objective. A theoretical analysis is presented that takes into account the approximation error (referred to as information loss in the paper) and the time drift in local objective functions. Experiments were also conducted that show core set methods to perform well.

**Summary Of The Review:**

While the topic is interesting, the paper has limited novelty, unrealistic assumptions, and hardly interpretable results.

---

> ### Author Response · Authors · 2021-11-17
> **To Reviewer XVxz (1/2)**
>
> Dear Reviewer XVxz,
>
> We would like to thank you for your time spent reviewing our paper and for providing constructive comments. Please find our responses to your raised questions below:
>
> ----
>
> > Novelty and clarification.
>
> **We believe the reviewer might have misinterpreted key aspects of our submission.** As the first theory work towards time-evolving scenarios in FL, our novelty, or the significance of the work, lies in (1) the novel FL formulation with continual learning (CFL formulation), (2) the rigorous theoretical analysis of CFL framework from the perspective of gradient noise, and (3) interesting empirical insights derived from thorough evaluation designed for continual federated learning.
>
> To the best of our knowledge,
> 1. **This is the first work that formally defines the CFL**, where traditional FL formulation is insufficient to capture the complex client behaviors, and there DOES NOT exist prior theoretical results on the considered time-varying collaborative learning scenario. *Our work is a crucial starting point for the whole community to study the realistic continual federated learning setting.*
> 2. **We provide the first theoretical results for the time-varying CFL, which are novel and significant.** Note that,
> 	* Prior FL rates, developed on the conventional static FL scenario, are infeasible to compare with our new rates proposed for challenging time-varying CFL scenarios, due to the completely different FL setups.
> 	* Our rates under the CFL framework can be simplified to the traditional FL setup, but not the other way.
> 3. **We provide tighter convergence rates for CL.** The CL results in Yin et al. (2020b) is only a special centralized case of what we considered, and their results cannot be trivially generalized to a collaborative learning setup. Note that,
> 	* Recent research in the FL community has put a great effort to theoretically analyze the (time-invariant) client data heterogeneity. The techniques in Yin et al. (2020b) even cannot handle the basic client heterogeneity and are far behind analyzing the heterogeneity caused by time- and client- drift we considered in the submission.
> 	* We provide tighter CL rates for mini-batch SGD, which is *different* from the coarse GD rates in Yin et al. (2020b). For example, our CL rate measures the joint effect of stochastic noise and time-drift, which can better capture the convergence behavior of CL.
> 4. **We conduct extensive empirical evaluations to bridge the techniques in conventional centralized CL to our CFL, and provide some interesting insights.** We would like to clarify that
> 	* The experiments on the (convex) Noisy Quadratic Models were provided in the submission to verify the correctness of our theory in Appendix C.1. In the revised draft (blue colored), we also include the non-convex convergence rate in Theorem 4.2 to align with our main deep learning experiments.
> 	* Our contributions ARE NOT proposing the core-set (replay buffer) as the solution to CFL. Instead, as clearly demonstrated in Sec 5.2, the values of our empirical results provide some interesting numerical insights, by thoroughly examining some conventional CL methods under the scope of CFL. We believe filling such gaps is valuable to the FL community and has not been done yet.
>
> ----
>
> > The relationship between Sec 4.2 and time-varying scenarios, the vanishment of $R$ when $p_{t, i} = 1$, why do CFL methods accelerate the convergence by reducing the variance term.
>
> Thank you for pointing out the confusing part. We have improved the text with blue color in the revised draft, and we list the main takeaways for clarification here:
> * Section 4.2 examines the convergence rate of FedAvg under the proposed time-varying scenarios, where both CFL and FedAvg will suffer from time- and client- drift, and the difference only lies in the choice of $p$.
> * We have improved Remark 4.6 and highlighted an important conclusion that, setting $p_{t, i} = 1$ is equivalent to setting $R = \infty$ in Theorem 4.2 (detailed proof refers to Appendix B.4).  One intuitive check could be by considering $c_A = G^2 + \frac{D^2 R^2}{D^2 + R^2}$ and $R \to \infty$, we can derive $c_A = G^2 + D^2$ and thus the CFL results in Theorem 4.2 can be simplified to the results of FedAvg in Theorem 4.5.
> * The evidence to justify the variance reduction effect of CFL comes from the fact that the increase of $R$ also increases the variance terms like $c_A$: CFL provides different approximation methods to control the information loss $R$ while FedAvg is equivalent to $R \to \infty$.
>
> ----

---

> ### Author Response · Authors · 2021-11-17
> **To Reviewer XVxz (2/2)**
>
> ----
>
> > The independent time drifts $\xi_{t, i}$ and zero-mean random variable $\delta_{t, i}$
>
> The reviewer might have misinterpreted key aspects of our formulation, and we would like to first clarify that in our formulation,
> * We decompose the drift of local functions into client drift $\delta_{t, i}$ and time drift $\xi_{t, i}$.
> * We assume each client $i$ has its own fixed client drift $\delta_{t, i}$, which is invariant of time step $t$ and can be simplified to $\delta_{i}$.
> * We assume a fixed underlying client distribution for each client $i$, and the time drift is only the additive noise for the given client drift. More precisely, an independent time-drift will be sampled for each time step $t$ and client $i$ and added on top of the fixed client drift to capture the time-evolving nature of the clients’ local dataset.
>
> The zero-mean client drift $\delta_{t, i}$ in our formulation (i.e. Equation 4) is indeed stemmed from the gradient dissimilarity bound in modeling client drift (i.e. $E[\Vert \nabla f_i(\omega) \Vert ^2] \le G^2 + B^2 E[\Vert\nabla f(\omega)\Vert^2]$), a common assumption which is widely used in the most recent FL works [1, 2, 3]. The proof of such equivalence can be found in Lemma B.1 of Appendix B.
>
> We believe these assumptions are meaningful and valuable for many realistic CFL scenarios. The attempt on CFL---such as what we provide---is a necessary and important first step for the whole community to investigate more complex modeling of the client drift across time.
>
> The reviewer may also check [our response to Reviewer 2NMQ](
> https://openreview.net/forum?id=oxC2IBx8OuZ&noteId=AzUkyU7IfPY), titled `CFL extension to client-only-participate-once (stateless) FL setting` for more details.
>
> [1] Sai Praneeth Karimireddy, Satyen Kale, Mehryar Mohri, Sashank Reddi, Sebastian Stich, and Ananda Theertha Suresh. Scaffold: Stochastic controlled averaging for federated learning. In International Conference on Machine Learning, pp. 5132–5143. PMLR, 2020b.
> [2] Anastasia Koloskova, Nicolas Loizou, Sadra Boreiri, Martin Jaggi, and Sebastian Stich. A unified theory of decentralized sgd with changing topology and local updates. In International Conference on Machine Learning, pp. 5381–5393. PMLR, 2020.
> [3] Xiang Li, Kaixuan Huang, Wenhao Yang, Shusen Wang, and Zhihua Zhang. On the convergence of fedavg on non-iid data. In International Conference on Learning Representations, 2020b. URL https://openreview.net/forum?id=HJxNAnVtDS.
>
> ----
>
> > Simplifying some terms in $O(\cdot)$ notations.
>
> We include all the necessary higher-order terms in $O(\cdot)$ to identify the difference in rates. The simplification may cause the indistinguishable rates between CFL and FedAvg, as it cannot identify the impact of client drift, round drift, and information loss on the convergence rates.
> For example, the rate of strongly convex functions in Theorem 4.2 can be simplified to $O = \left( \frac{L c_1}{\mu} + \frac{c_2}{\mu \epsilon} + \frac{c_3}{\sqrt{\mu \epsilon}} \right)$, where $c_1$, $c_2$, and $c_3$ are variance terms and we cannot identify the difference.
>
> ----
>
> > Theory and experiments
>
> We did include the convex experiments to verify our theoretical results for convex and strongly-convex functions, as shown in Appendix C.1.
> We completely agree with the reviewer and we have added the non-convex proof in Theorem 4.2 (blue colored) in the revised draft to strengthen the draft; the proof details are deferred in Appendix B.5. These theoretical results---verify the faster convergence of CFL than that of FedAvg---are aligned with our convex and non-convex deep learning experiments.
>
> ----
>
> > Misunderstanding of core-set methods and the privacy concern
>
> We think it is definitely a misunderstanding: the core set will NEVER be maintained collectively across different clients, and each device will only maintain the core set locally.
> The motivation of mentioning the generative methods only lies in the resource efficiency concern: due to the low client participation in practice, maintaining the core set for each device may cause unnecessary storage burdens for e.g. millions/billions of inactive clients.
>
> ----

---

> ### Author Response · Authors · 2021-11-20
> **Kind reminder for our response**
>
> Dear Reviewer XVxz,
>
> We thank you again for providing timely feedback. As we believe we have addressed your main concerns in your review, we would therefore like to ask you if you could reconsider your score? We remain at your disposal for any further clarification which you might require.

---

> ### Comment · Reviewer_XVxz · 2021-11-20
> **Thank you for your response**
>
> Thanks for the authors' response. However, I'm not fully convinced with the explanations. The reasons are as follows.
>
> > Novelty
>
> Something that has not been done before does not justify it being novel. There needs to be a unique and non-obvious technical step to make it novel. I do not seem to see such a step from the response and not from the paper either.
>
> > Independence of $t$
>
> If the authors indeed mean invariant of time step $t$, then there should not be a subscript $t$ in $\delta$. The paper still says "independent", which I would naturally understand as $\delta_{t,i}$ being independent across different $t$. My concern was mostly related to the time-dependent drift $\xi_{t,i}$ though, where Remark 3.2 says that they are independent (I assume the independence is with respect to both $t$ and $i$). This is also seen in the proof of Lemma B.2 in the appendix, where independence of $\xi_{\tau,i}$ across different $\tau$ and zero-mean are needed to expand the last term of Equation (25) to obtain Equation (26). If $\\{\xi_{\tau,i}\\}$ are not independent across $\tau$, then there will be inner product terms after expanding the last term of Equation (25) which are generally non-zero. It also seems that the variables $A$ and $B$ are swapped in Equation (26) compared to the definition of upper bounds in Assumption 3.
>
> > $O(\cdot)$ notation
>
> The terms inside the $O(\cdot)$ notation could include the drift terms (i.e., do not consider them as constant). This will possibly show the effect of different drifts. However, the upper bounds due to Assumption 3 probably makes this impossible (also see below).
>
> > Time-varying (Sec. 4.2 etc.)
>
> I still do not see how the time-varying aspect is exactly captured. Assumption 3 expresses both $\delta$ and $\xi$ using upper bounds, which basically makes it not possible to capture how fast $\xi$ changes over time and how this change affects the convergence.
>
> > Variance terms like $c_A$
>
> This may be minor, but by variance, I was thinking about the SGD noise term (i.e., $\sigma^2$). The authors seem to mean something different here. It would be good if the authors can explain why they call these terms as variance.
>
> I appreciate the authors' effort of adding a non-convex result. However, the above issues remain. Also, Equation (9) expresses the convergence as an upper bound of the gradient norm, while the other results express the convergence in the number of rounds $T$. These two types or results are convertible to each other, but they should be expressed in a consistent way.

---

> > ### Author Response · Authors · 2021-11-22
> > **Follow-up response to reviewer Xvxz (1/2)**
> >
> > Dear Reviewer XVxz,
> >
> > Thanks for your feedback! We have fixed the typo in the swapped $A$ and $B$ in Equation 26 and rewrote Equation 9 in the form of the number of rounds $T$.
> >
> > To address your concerns about (1) the independence of $\xi_{t, i}$ and (2) the formulation of the time-varying aspect, we would like to reiterate the clarification of our formulation stated in the previous response, before explaining them in details:
> > * We decompose the drift of local functions into client drift $\delta_{t, i}$ and time drift $\xi_{t, i}$. Client drift is only in terms of client $i$, and time drift is in terms of time $t$ for each client $i$.
> > * We assume each client $i$ has its own fixed client drift $\delta_{t, i}$, which is **independent and invariant of time step $t$** and can be simplified to $\delta_{i}$.
> > * We assume a fixed underlying client distribution for each client $i$, and the time drift is only the additive noise for the given client drift. For each client $i$, **an independent time-drift** will be sampled for each time step $t$ and added on top of the fixed client drift to capture the time-evolving nature of the clients’ local dataset.
> >
> > As a summary, by extending the common assumption in the field [1, 2, 3, 4], given the gradient of the global objective function $\nabla f(x)$, we have the following modeling:
> > * The client drift is defined as $\nabla f_i(\omega) = \nabla f(\omega) + \delta_i$, where $\nabla f_i(\omega)$ is the gradient of a client.
> > * The time-varying scenario is captured by considering $\nabla f_i(\omega) = \nabla f_{t, i} (\omega) + \xi_{t, i}$, where $\nabla f_{t, i} (\omega)$ is the gradient of a client $i$ at round $t$, and $\xi_{t, i}$ measures the empirical noise caused by time-varying data.
> > * The local mini-batch SGD of each client $i$ at round $t$ is formulated as $\nabla f_{t, i, k} (\omega) = \nabla f_{t, i} (\omega) + \nu_{t, i, k}$, where $\nu_{t, i, k}$ indicates the stochastic noise of mini-batch SGD.
> >
> > Kindly note that we use $\delta_i, \xi_{t, i}, \nu_{t, i, k}$ to indicate the noise from different sources, and they can be bounded with the variance terms.
> >
> > ----
> >
> > > How the formulation captures the time-varying aspect, how the upper bounds in $\delta$ and $\xi$ measure the change speed and its impact on convergence
> >
> > We would like to clarify (together with the formulation above) that
> > * This formulation is inspired by the stochastic gradient noise in SGD. The empirical client-$i$ objective function, determined by the local dataset, will vary over time, while the client-$i$’s underlying data distribution (i.e. true objective function) will never change.
> > * The time drift **does not** refer to the difference between two empirical local objective functions at time $t$ and $t’$. Instead, the time drift **only** indicates the stochasticity of the empirical client-$i$ objective function compared to the true client-$i$ objective function (empirical objective function in expectation), which is **independent of time**.
> > * The time drift bound presented in previous draft comes from the definition $\nabla f_{t, i} (\omega) = \nabla f_{i} (\omega) + \xi_{t, i}$ and assumption $E[\Vert \xi_{t, i} \Vert^2] \le \hat{D}^2 + \hat{A}^2 E[\Vert \nabla f_i(\omega) \Vert^2] \le \hat{D}^2 + \hat{A}^2 (G^2 + B^2 E[\Vert \nabla f(\omega) \Vert^2])$. We can recover the assumption $E[\Vert \xi_{t, i} \Vert^2] \le D^2 + A^2 E[\Vert \nabla f(\omega) \Vert^2]$, by assigning $A^2 = \hat{A}^2 B^2$, $D^2 = \hat{D}^2 + \hat{A}^2 G^2$ (We have clarified it in the revised text). *The pair like (G, B) and (D, A) in the upper bounds of client and time drift cannot be exchanged*.

---

> > ### Author Response · Authors · 2021-11-22
> > **Follow-up response to reviewer Xvxz (2/2)**
> >
> > * Detailed characteristics of Assumption 3 on the bounded time drift.
> > 	* Recap the bound in Assumption 3, we assume $E[\Vert \xi_{t, i} \Vert^2] \le \hat{D}^2 + \hat{A}^2 E[\Vert \nabla f_i (\omega) \Vert^2]$. The bound on $E[\Vert \xi_{t, i} \Vert^2]$ depends on a constant $\hat{D}^2$, and a term related to the client gradient $\hat{A}^2 E[\Vert \nabla f_i (\omega) \Vert^2]$.
> > 	* The bound of time drift with constant $(\hat{D}, \hat{A})$ does not mean constant drift $\xi$. A larger $E[\Vert \xi_{t, i} \Vert^2]$ implies a more dramatic change of $\xi$.
> > 	* The $\hat{D}$ hinders the convergence by introducing $c_A$ and $c_B$ in Theorem 4.1 but has no effect on the $E[\Vert \omega_t - \omega^* \Vert^2]$, due to it being irrelevant to the gradients.
> > 		* As a side observation, the $\hat{D}$ only determines if the optimum of empirical local objective function will change over time or not. Considering client $i$ reaches the optimum $\omega_i^*$, i.e., $\nabla f_i(\omega_i^*) = 0$, we have $E[\Vert \nabla f_{t, i} (\omega_i^*) \Vert^2] \le \hat{D}^2$. When $\hat{D} = 0$, $f_{t, i} (\omega)$ and $f_{i} (\omega)$ share the same optimum $\omega_i^*$, as $\nabla f_{t, i}(\omega_i^*) = \nabla f_i(\omega_i^*) = 0$. When $\hat{D} > 0$, the optimum of $f_i(\omega)$ (i.e. $\omega_i^*$) is not the optimum weight of $f_{t,i}(\omega)$, as $\nabla f_{t, i} (\omega_i^*) > 0$.
> > 	* The impact of $\hat{A}$ will vanish when $E[\Vert \nabla f_i (\omega) \Vert^2] = 0$. This property implies that the $\hat{A}$ hinders the convergence by slowing down the main term ($E[\Vert \omega_0 - \omega^* \Vert^2]$ for convex functions, and $E[f_0] - E[f^*]$ for non-convex functions).
> >
> > [1] Sai Praneeth Karimireddy, Satyen Kale, Mehryar Mohri, Sashank Reddi, Sebastian Stich, and Ananda Theertha Suresh. Scaffold: Stochastic controlled averaging for federated learning. In International Conference on Machine Learning, pp. 5132–5143. PMLR, 2020b.
> > [2] Anastasia Koloskova, Nicolas Loizou, Sadra Boreiri, Martin Jaggi, and Sebastian Stich. A unified theory of decentralized sgd with changing topology and local updates. In International Conference on Machine Learning, pp. 5381–5393. PMLR, 2020.
> > [3] Xiang Li, Kaixuan Huang, Wenhao Yang, Shusen Wang, and Zhihua Zhang. On the convergence of fedavg on non-iid data. In International Conference on Learning Representations, 2020b. URL https://openreview.net/forum?id=HJxNAnVtDS.
> > [4] Ahmed Khaled, Konstantin Mishchenko, and Peter Richtárik. Tighter theory for local sgd on identical and heterogeneous data. In International Conference on Artificial Intelligence and Statistics, pp. 4519–4529. PMLR, 2020.

---

> > ### Comment · Reviewer_XVxz · 2021-11-22
> > **Thanks for the follow-up**
> >
> > Thanks for the follow-up response. I maintain my opinion that the time-independent and zero-mean $\xi_{t,i}$ is insufficient for capturing characteristics of continual learning. The client's dataset usually changes gradually over time, which means $\xi_{t,i}$ should not be independent across time nor zero-mean. As the authors also pointed out by "inspired by the stochastic gradient noise in SGD", the current notion of $\xi_{t,i}$ is indeed very similar to SGD noise. This also justifies my comment on lack of novelty. The only new contribution is the consideration of an approximate loss instead of the exact loss (more specifically, the gradient of the loss), which is a quite straightforward extension of what already exists.

---

> > > ### Author Response · Authors · 2021-11-22
> > > **Clarification**
> > >
> > > Thanks for your feedback.
> > >
> > > In this work, we aim to provide a general CFL formulation, as the first starting point for the community, to investigate the challenging time-varying FL, including the stateless cross-device FL where clients (out of millions of clients) have low participation frequency, or even join the learning only once.
> > >
> > > We do acknowledge the existence of a complex change pattern of local datasets for cross-silo FL settings, but this normally requires some scenario-dependent assumptions and is beyond the scope of this work. We would like to further point out that the dependent assumption on time drift may only introduce more variance terms on the rates, without changing our main results. Besides, it is hard to quantify the dependence of the gradient noise of two different parameters.

---

### Official Review · Reviewer_2NMQ · 2021-11-03

**Correctness:** 3
**Technical Novelty And Significance:** 3
**Empirical Novelty And Significance:** 3
**Recommendation:** 8
**Confidence:** 3

**Main Review:**

STRENGTHS:

- The paper introduces clearly the problem and contribution, while clearly stating differences with state of the art in FL and CL.
- The paper bridges between these two frameworks, and proposes a clear algorithm to learn from time evolving data.
- Analyses and guarantees are provided under full knowledge of $f_t$

IMPROVEMENTS:
- I would suggest discussing how to extend CFL for the setting when clients participate at most once during training. I saw the comment in the experimental section but it would be nice to have a remark on the theoretical section describing it.
- Time evolving patterns could have a common component across clients. It would be interesting to see experiments involving data partitions across time capturing this king of realistic trends. For example on Twitter data each user tends to have a drift (their own posting behavior) but there is also a time component (trending news, etc. )
- There is no analyses on the real algorithm. How does CFL with specific function approximation behave? what is $R$ for the considered estimation methods? Perhaps it is not tractable but at least a connection, or a remark on this aspect, could make the transition between theory and experiments more smooth.
-  Given the time evolving nature of data, it would be good to see different metrics (not only final accuracy) that represent how the algorithm is able to maintain accuracy as time-shifts occur.

Minor comments:
- I would suggest adding a subscript t in equation (2) to indicate that it is a time-evolving objective.


**Summary Of The Paper:**

This paper proposes to study time evolving heterogeneous data, and proposes Continual Federated Learning (CFL) to address this problem. Their analysis achieves this by introducing time-drift to capture data heterogeneity across time. Convergence results are presented followed by numerical results on time-varying and heterogeneous settings.


**Summary Of The Review:**

This is a paper that clearly defines a problem in Federated Learning and proposes to extend existing tools from continual learning to this setting. The problem is well motivated, and clearly formulated. Convergence guarantees are interesting but it would be better to have the real bounds for the practical algorithms used, bounds that account for the error estimation of $f_t$. This would allow to see how practical the algorithm is in real settings. Experiments are consistent with claims and show the potential of this framework.

My score is based on above comments, but I think several of these can be addressed during the rebuttal.

---

> ### Author Response · Authors · 2021-11-17
> **To Reviewer 2NMQ**
>
> Dear Reviewer 2NMQ,
>
> We would like to thank you for your time spent reviewing our paper and for providing constructive comments. Please find our responses to your raised questions below:
>
> ----
> > CFL extension to client-only-participate-once (stateless) FL setting.
>
> **Our CFL formulation can capture the mentioned setting, and a discussion has been included as a theoretical remark in the revised draft (Remark 4.4 with blue colors).**
>
> We explain the ingredient we use to formulate the mentioned setting below:
> * In our proposed gradient noise model, we decompose the drift of local function into client drift $\delta_{t, i}$ and time drift $\xi_{t, i}$.
> * We assume each client $i$ has its own fixed client drift $\delta_{t, i}$, which is invariant of time step $t$. $\delta_{t, i}$ can be simplified to $\delta_{i}$.
> * We assume a fixed underlying client distribution for each client $i$, and the time drift is only the additive noise for the given client drift. More precisely, an independent time-drift will be sampled for each time step $t$ and added on top of the fixed client drift to capture the time-evolving nature of the clients’ local dataset.
> * Our CFL formulation can cover the mentioned client-only-participate-once setting, by additionally assuming the independence between $\delta_{t, i}$. This assumption is equivalent to assuming stateless clients, a generalized version of the client-only-participate-once setting.
>
> A theoretical remark regarding this setting has been added in the revised draft (Remark 4.4), where we derive a tighter bound for the stateless scenario. Compared with general cases (i.e. cases with stateful clients), the variance terms in the stateless scenario will be changed from $G^2 + \frac{D^2}{t}$ to $\frac{G^2 + D^2}{t}$, allowing a straightforward improvement.
>
> We also report the results of training with stateless clients in Table 6, where we exclude the core-set method as it is infeasible for the stateless scenario.
>
> ----
>
> > Time evolving patterns could have a common component across clients. It would be interesting to see experiments involving data partitions across time capturing this kind of realistic trend...
>
> **The requested experiments were included in Table 5.** In Table 5, we assume a common time-evolving pattern across clients, where the data will be partially and continually removed and generated on each device over time. The illustration of such a pattern can be found in Figure 3 of Appendix C.2.1.
> We agree with the reviewer that it is an interesting direction and we would like to investigate other types of time-evolving patterns in our future work.
>
> ----
>
> > The behavior of CFL with different function approximation methods.
>
> **Due to the intractable analysis of many considered function approximation methods (in Section 5), Figure 1 in the original submission empirically justifies the correlation between information loss $R$ and learning performance, through the core-set method**. In Figure 1, we consider training deep networks on different core set sizes and the size indicates different approximation qualities (larger is better).
> The observations in Figure 1 are aligned with our theoretical assumptions/results: *the lesser the information loss, the higher the performance.*
>
> **We add a new Remark 3.3 to complement Assumption 4, as suggested by the reviewer.**
> This remark provides a tight analysis of the bounded information loss on the feasible Taylor-Extension-based approximation method.
> Together with the provided convex experimental results under the Noisy Quadratic Model in Appendix C.1,  we achieve a smooth transition between theoretical analysis and experimental insights.
>
> ----
>
> > Other metrics to capture the behavior of the algorithm with time-shifts.
>
> **Various types of learning curves at different learning phases can be found in Appendix C.2.3**. For example, Figure 6 provides different views of learning and explicitly investigates the robustness when time-shifts occur: CFL has much fewer spikes than FedAvg and thus a stabilized optimization procedure.
>
> ----
>
> > subscript t in equation (2)
>
> We have updated the text in the revised draft to better indicate the time-evolving objective.
>
> ----

---

### Author Response · Authors · 2021-11-17
**Reply to all reviewers (1/2)**

We thank all reviewers for their time and their valuable feedback. We addressed comments from each of the reviewers separately below each of the reviews. In this thread, we also summarized our updates on the draft and clarified some joint concerns of the reviewers.

Please find the summary of our revision below (the updated part is highlighted in blue color):
* We added corrections/clarifications as suggested by each reviewer.
* We added the convergence rate of CFL for non-convex objectives in Theorem 4.2 and included the analysis details in Appendix B.5, as requested by Reviewer XVxz, Reviewer s89S, and Reviewer 87pq. In addition, we want to clarify that we did include convex numerical experiments, which verify our theoretical results on convex cases, in the original submission.
* We improved Remark 4.6 to better clarify the relationship between CFL methods and FedAvg, under the challenging time-varying client heterogeneous data scenarios, as pointed out by Reviewer XVxz.
* We added a new Remark 3.3 to complement Assumption 4 and addressed the comment of Reviewer s89S.
* We added a new Remark 4.4 to show the feasibility of extending the CFL framework to the stateless FL scenario, as suggested by Reviewer 2NMQ.
* We added the convergence results of [7] in Table 1, as suggested by Reviewer s89S.

We would like to emphasize and clarify the contribution and novelty of our work, as acknowledged by the positive review of Reviewer 2NMQ and Reviewer s89S:
1. **This is the first work that formally defines the CFL**, and our work is a crucial starting point for the whole community to study the realistic continual federated learning setting. More importantly, the traditional FL formulation is only a special case of our CFL formulation, not the other way. CFL considers the time-varying local (client) objective functions, captured by time- and client- drift, while FL only considers the fixed client objective functions.
2. **The proposed CFL framework is novel**, covering different approximation techniques for diverse time-varying FL scenarios.
	* In Section 3.1, we introduce the definition of information loss (Definition 3.1) to measure the approximation quality. This definition, together with Assumption 4, Remark 3.3, leads to the novel theoretical results for CFL in Section 4.
	* We introduce three types of approximation techniques in Section 5.1 and elaborate its details in Appendix C.2.2. We further empirically justify its practical preciseness and insights using 2.5 pages of numerical results.
3. **The gradient noise model, the basic assumption of our theoretical analysis, is standard and widely used** [1, 2, 3, 4, 5, 6]. Please note that most theoretical works on FL/decentralized SGD rely on this assumption, e.g., [4, 5, 6]; a more complete list can be found in our related work section.
4. **Our way of modeling client and time drift is interesting and valuable**. We would like to clarify that in our formulation,
	* We decompose the drift of local functions into client drift $\delta_{t, i}$ and time drift $\xi_{t, i}$.
	* We assume each client $i$ has its own fixed client drift $\delta_{t, i}$, which is invariant of time step $t$ and can be simplified to $\delta_{i}$.
	* We assume a fixed underlying client distribution for each client $i$, and the time drift is only the additive noise for the given client drift. More precisely, an independent time-drift will be sampled for each time step $t$ and client $i$ and added on top of the fixed client drift to capture the time-evolving nature of the clients’ local dataset.

---

### Author Response · Authors · 2021-11-17
**Reply to all reviewers (2/2)**

5. **We provide the first theoretical results for the time-varying CFL, which are novel and significant.** Note that,
	* Prior FL rates, developed on the conventional static FL scenario, are infeasible to compare with our new rates analyzed for challenging time-varying CFL scenarios, due to the completely different FL setups.
	* Our rates under the CFL framework can be simplified to the traditional FL setup and traditional CL setup, but not the other way.
	* Providing a tighter rate for traditional FL setup is beyond the scope of this work.
	* Compared to the rate of Yin et al. 2020b under the traditional CL setup for GD, we can provide tighter rates for CL on mini-batch SGD.
6. **We conduct extensive empirical evaluations to bridge the techniques in conventional centralized CL to our CFL, and provide some interesting insights.** We would like to clarify that
	* The experiments on the (convex) Noisy Quadratic Models were provided in the submission to verify the correctness of our theory in Appendix C.1. In the revised draft, we also included the non-convex convergence rate in Theorem 4.2 to align with our deep learning experiments.
	* Our contributions ARE NOT proposing the core-set (replay buffer) as the solution to CFL. Instead, as clearly demonstrated in Sec 5.2, the values of our empirical results provide some interesting practical insights, by thoroughly examining some conventional CL methods under the scope of CFL. We believe filling such gaps is valuable to the FL community and has not been done yet.

[1] Sievert S. Improving the convergence of SGD through adaptive batch sizes[J]. arXiv preprint arXiv:1910.08222, 2019.
[2] Shamir O, Zhang T. Stochastic gradient descent for non-smooth optimization: Convergence results and optimal averaging schemes[C]//International conference on machine learning. PMLR, 2013: 71-79.
[3] Gower R M, Loizou N, Qian X, et al. SGD: General analysis and improved rates[C]//International Conference on Machine Learning. PMLR, 2019: 5200-5209.
[4] Xiang Li, Kaixuan Huang, Wenhao Yang, Shusen Wang, and Zhihua Zhang. On the convergence of fedavg on non-iid data. In International Conference on Learning Representations, 2020b. URL https://openreview.net/forum?id=HJxNAnVtDS.
[5] Ahmed Khaled, Konstantin Mishchenko, and Peter Richtárik. Tighter theory for local sgd on identical and heterogeneous data. In International Conference on Artificial Intelligence and Statistics, pp. 4519–4529. PMLR, 2020.
[6] Sai Praneeth Karimireddy, Satyen Kale, Mehryar Mohri, Sashank Reddi, Sebastian Stich, and Ananda Theertha Suresh. Scaffold: Stochastic controlled averaging for federated learning. In International Conference on Machine Learning, pp. 5132–5143. PMLR, 2020b.
[7] Woodworth B, Patel K K, Srebro N. Minibatch vs local sgd for heterogeneous distributed learning[J]. arXiv preprint arXiv:2006.04735, 2020. https://arxiv.org/pdf/2006.04735.pdf

---

### Decision · Program_Chairs · 2022-01-20

**Decision:**

Reject

**Comment:**

This paper proposes “Continual Federated Learning (CFL)” to study time evolving heterogeneous data. To do this the authors introduce time-drift to capture data heterogeneity across time. The authors also present some preliminary convergence results. Finally, the authors carryout numerical experiments in time-varying and heterogeneous settings. The reviewers identified the following strengths: (1) combining FL and CL is interesting, (2) the development of a new algorithm and providing some initial analysis is a good step. They also identified weaknesses as follows: (1) limited technical novelty as the use of replay buffer is quite standard, (2) cumbersome and not easy to interpret results, (3) lack of time evolving patterns with a common component (4) lack of different metrics that demonstrate how the algorithm is able to maintain accuracy as time-shifts occur, (5) lack of questionable assumptions. The reviewers had a very bimodal view advocating acceptance with a score of 8 and 2 advocating a rejection and neither group changed their opinion. Although the authors thorough responses did alleviate the concerns IMO. My own reading of the paper is that this is an interesting paper working on an emerging area. However, I must agree with some of the reviewers that the final conclusions are not easy to interpret, and the assumptions are not fully motivated. After this is carried out, I think the novelty of the paper can also become much clear. Therefore, I cannot strongly advocate acceptance of the paper in its currently state given the scores. However, I very strongly encourage the authors to submit to a future ML venue after addressing the remaining comments of the reviewers. I would also like to commend the authors for a very strong rebuttal sorry the final decision couldn’t be more favorable given the borderline ratings and the aforementioned issues.